# A Case for Vanilla SWD: New Perspectives on Informative Slices, Sliced-Wasserstein Distances, and Learning Rates

**Huy Tran**                                    *huy.tran@vanderbilt.edu*
*Department of Computer Science*
*Vanderbilt University*

**Yikun Bai**                                    *yikun.bai@vanderbilt.edu*
*Department of Computer Science*
*Vanderbilt University*

**Ashkan Shahbazi**                          *ashkan.shahbazi@vanderbilt.edu*
*Department of Computer Science*
*Vanderbilt University*

**John R. Hershey**                              *johnhershey@google.com*
*Google Research*

**David Hyde**                                  *david.hyde.1@vanderbilt.edu*
*Department of Computer Science*
*Vanderbilt University*

**Soheil Kolouri**                            *soheil.kolouri@vanderbilt.edu*
*Department of Computer Science*
*Vanderbilt University*

**Reviewed on OpenReview:** *https:// openreview. net/ forum? id= li8D5pxczd*

## Abstract

The practical applications of Wasserstein distances (WDs) are constrained by their sample and computational complexities. Sliced-Wasserstein distances (SWDs) provide a workaround by projecting distributions onto one-dimensional subspaces, leveraging the more efficient, closed-form WDs for 1D distributions. However, in high dimensions, most random projections become uninformative due to the concentration of measure phenomenon. Although several SWD variants have been proposed to focus on *informative* slices, they often introduce additional complexity, numerical instability, and compromise desirable theoretical (metric) properties of SWD. Amid the growing literature that focuses on directly modifying the slicing distribution, we revisit the standard, "vanilla" Sliced-Wasserstein through an effective-subspace model and a rescaling view of slice informativeness. We show that, with an effective-subspace-aligned notion of *slice informativeness*, reweighting all individual slices simplifies in expectation to a single global scaling factor relating ambient-space SWD to effective-subspace SWD. For GD/SGD-style first-order optimization, the same factor appears as a step-size calibration effect. We perform extensive experiments across various machine learning tasks showing that vanilla SWD, when properly calibrated, can often match or surpass the performance of more complex variants while retaining its simplicity and metric structure.

# 1 Introduction

Optimal transport (OT) theory (Villani et al., 2009; Peyré et al., 2019) provides a principled way to compare data distributions by finding an optimal transportation plan that minimizes the expected cost of moving mass between them, leading to the popular Wasserstein distance (WD) central to many learning applications (Khamis et al., 2024). However, the computational complexity of OT solvers poses a significant bottleneck when calculating the WD. In cases of discrete measures or sample-based scenarios, which are common in machine learning, the problem typically reduces to linear programming with time complexity $\mathcal{O}(N^3 \log N)$, space complexity $\mathcal{O}(N^2)$, and sample complexity $\mathcal{O}(N^{-\frac{1}{d}})$, where $N$ is the number of support points and $d$ the data dimensionality. These unfavorable scaling properties, particularly the curse of dimensionality in sample complexity, make WD impractical for many real-world applications. To address these challenges, several approaches have been proposed, including entropic regularized OT (Cuturi, 2013), smooth OT (Blondel et al., 2018; Manole et al., 2024), and sliced OT (Bonneel et al., 2015).

In particular, the Sliced-Wasserstein distance (SWD) (Rabin et al., 2012; Bonneel et al., 2015) projects high-dimensional distributions onto 1D subspaces and aggregates the closed-form OT solutions in these subspaces. This method is particularly attractive because 1D Wasserstein distances can be computed efficiently with a time complexity of $\mathcal{O}(N \log N)$ and a space complexity of $\mathcal{O}(N)$ for discrete measures. Additionally, SWD provides a metric between probability distributions that retains many desirable properties of the Wasserstein distance (WD), such as being statistically and topologically equivalent to WD, while being more computationally tractable (Nadjahi et al., 2020). Notably, with a dimension-free sample complexity of $\mathcal{O}(N^{-\frac{1}{2}})$, SWD avoids the curse of dimensionality. However, a key drawback of SWD is its projection complexity, which requires exponentially more slices as the data dimensionality increases.

The projection complexity of SWD has motivated several lines of work that aim to enhance the effectiveness of the slicing approach, especially in addressing variance reduction (Nguyen & Ho, 2023), approximation error reduction (Nguyen et al., 2023), and slicing complexity (Kolouri et al., 2019; Deshpande et al., 2019; Nguyen et al., 2020; 2024a; Nguyen & Ho, 2024; Nguyen et al., 2024b). This is particularly relevant in high-dimensional machine learning settings, where data often has supports in low-dimensional subspaces. These SWD variants are data-driven, focusing on identifying the most informative slices for capturing distributional differences in the data. For instance, Max-SW (Deshpande et al., 2019) and DSW (Nguyen et al., 2020) seek to find slices/projections that maximize the differences between the data distributions. GSW (Kolouri et al., 2019), ASW (Chen et al., 2020), and TSW (Tran et al., 2025a) extend SWD by allowing 'non-linear' projections to capture complex data structures. EBSW (Nguyen & Ho, 2024) designs an energy-based slicing distribution that is parameter-free and has the density proportional to an energy function of the projected 1D distance. MSW (Nguyen et al., 2024a) imposes a first-order Markov structure to avoid redundant, independent projections. More recently, RPSW (Nguyen et al., 2024b) proposes using the normalized differences between random samples from the two distributions to ensure that the projections are sampled from the subspace in which the data resides. BOSW (Acharya & Hyde, 2025) uses Bayesian optimization to select informative slices in a sample-efficient manner. These methods improve the performance of SW in various downstream tasks and have significantly expanded the tools at our disposal for both researchers and practitioners alike. Nonetheless, these elegant extensions also come with increased computational cost, numerical instability, complicated design choices, and often lose the metricity of the SWD.

In this paper, we argue that the standard SWD, once properly calibrated, can often match or surpass the performance of more complex variants in many learning tasks while retaining its simplicity and theoretical guarantees. Our key insight is that when $d$-dimensional data have $k$-dimensional supports, where $k \ll d$, almost all random slices $\theta \sim \mathcal{U}(\mathbb{S}^{d-1})$ can be decomposed into an *informative* component $\theta_D \in \mathbb{R}^k$ within the data subspace and its orthogonal complement $\theta_D^\perp \in \mathbb{R}^{d-k}$. This implies most slices still carry relevant information for distinguishing distributions, proportional to $\|\theta_D\|$. By appropriately scaling the distance per slice, we obtain better gradients for learning. In expectation, we show that, with our defined notion of *informativeness*, scaling for all slices simplifies to scaling the SWD by a single scalar factor under the effective-subspace model studied in the paper. In practice, for gradient-based learning, we show empirically that the resulting change is reflected mainly through standard learning-rate calibration. This allows the classical SWD to adapt to the data's intrinsic dimensionality without explicitly limiting the computation

to the subspace. We provide theoretical justification and empirical evidence for this rescaling perspective on SWD in high-dimensional settings. Our insights clarify why the standard SWD performs strongly in distribution-based learning tasks and open up new directions for investigating other task-specific data assumptions and corresponding notions of informative slices.

By revisiting the standard SWD with these insights, we elucidate the "performance gap" between the original formulation and recent variants in the existing literature. We emphasize that our work does not diminish the valuable contributions of these variants, which have greatly advanced our understanding of SWD. Instead, we offer a complementary perspective that highlights the potential of the standard SWD when properly integrated into learning tasks. In the same spirit, the related body of specialized methods that respects the data geometry (Rabin et al., 2011; Bonet et al., 2022; Martin et al., 2023; Quellmalz et al., 2023; Bonet et al., 2024a; Tran et al., 2024; 2025b) remains valuable when the manifold constraint on the data is readily known.

In common ML settings where data is (nearly) supported on a $k$-dimensional subspace embedded in a $d$-dimensional space, our findings can be summarized as follows:

- We use the $\phi$-weighting formulation as a unifying language for different SWD variants. Within this framework, our specific contribution is to show that under the effective-subspace model, an appropriate notion of *slice informativeness* yields a Subspace Sliced-Wasserstein variant that differs from the standard SWD only by a scalar factor in expectation.

- Our findings reduce the problem of *non-informative slices* to the learning-rate search for the classic SWD, a process that is already standard in ML workflows. In other words, this calibration can make the slices effectively informative.

- We perform a comprehensive learning-rate sweep across a range of experiments, including gradient flow (on 3 classic toy datasets, MNIST images, CelebA images), color transfer (3 sets of images), and deep generative modeling on the FFHQ dataset (unconditional generation and unpaired translation with SW). We show that the classic SWD, with appropriate hyperparameters, performs competitively with more advanced methods in these settings.

**Notations.** Throughout the paper, we let $\mathbb{R}^d$ denote a $d$-dimensional inner product space, and we denote the unit hypersphere in this space by $\mathbb{S}^{d-1} = \{\theta \in \mathbb{R}^d : \|\theta\|_2 = 1\}$. Additionally, we denote by $\mathcal{P}(\mathbb{R}^d)$ the set of probability measures on $\mathbb{R}^d$ endowed with the $\sigma$-algebra of Borel sets, and by $\mathcal{P}_p(\mathbb{R}^d) \subset \mathcal{P}(\mathbb{R}^d)$ the subset of those measures with finite $p$-th moments. For a measurable function $f : \mathbb{R}^d \to \mathbb{R}$ defined by $f(x) = \theta^\top x$ such that $\theta \in \mathbb{S}^{d-1}$, we denote the pushforward of a measure $\mu \in \mathcal{P}(\mathbb{R}^d)$ through $f$ as $f_\# \mu$. More generally, for any $w \in \mathbb{R}^d$, we write $w_\# \mu$ for the pushforward of $\mu$ under the map $x \mapsto w^\top x$. In particular, when $w \in \mathbb{S}^{d-1}$ this coincides with the standard 1D projection onto a unit direction.

## 2 Background: The Sliced-Wasserstein Distance

Let $\mu \in \mathcal{P}_p(\mathbb{R}^d)$ and $\nu \in \mathcal{P}_p(\mathbb{R}^d)$ be two probability measures of interest.

**Wasserstein distance (WD).** The $p$-WD between $\mu$ and $\nu$ is:

$$W_p^p(\mu, \nu) = \inf_{\pi \in \Pi(\mu,\nu)} \int_{\mathbb{R}^d \times \mathbb{R}^d} \|x - y\|_p^p \, d\pi(x, y), \tag{1}$$

with $\Pi(\mu, \nu) = \{\pi \in \mathcal{P}_p(\mathbb{R}^d \times \mathbb{R}^d) : \pi(A \times \mathbb{R}^d) = \mu(A), \quad \pi(\mathbb{R}^d \times A) = \nu(A)\}$ for all measurable sets $A \subset \mathbb{R}^d$. In one dimension ($d = 1$), the p-WD admits the following closed-form solution:

$$W_p^p(\mu, \nu) = \int_0^1 |F_\mu^{-1}(z) - F_\nu^{-1}(z)|^p \, dz, \tag{2}$$

where $F_\mu, F_\nu$ are the cumulative distribution functions (CDF) of $\mu$ and $\nu$, respectively, and $F_\mu^{-1}(z) := \inf\{t \in \mathbb{R} : F_\mu(t) \geq z\}$ (and similarly for $F_\nu^{-1}$). For empirical measures, Equation 2 becomes a Monte Carlo sum that

can be calculated by averaging $|x_{(i)} - y_{(i)}|^p$ between sorted samples. In general, this translates to a highly favorable time complexity of $\mathcal{O}(N \log N)$ and gives rise to the following Sliced-Wasserstein distance.

**Sliced-Wasserstein distance (SWD).** The SWD between $\mu$ and $\nu$ is defined as:

$$SW_p(\mu, \nu; \sigma) := \left( \mathbb{E}_{\theta \sim \sigma} \left[ W_p^p(\theta_\# \mu, \theta_\# \nu) \right] \right)^{\frac{1}{p}} \tag{3}$$

where $\sigma \in \mathcal{P}(\mathbb{S}^{d-1})$ is the reference measure for slicing vector $\theta$. In the default setting, $\sigma$ is the uniform distribution, denoted as $\sigma = \mathcal{U}(\mathbb{S}^{d-1})$, and we use $SW_p(\mu, \nu)$ to denote $SW_p(\mu, \nu; \sigma)$ for simplicity. The intractable expectation in Equation 3 admits a Monte Carlo estimator:

$$SW_p(\mu, \nu; \sum_{l=1}^{L} \frac{1}{L} \delta_{\theta_l}) = \left( \frac{1}{L} \sum_{l=1}^{L} W_p^p \left( (\theta_l)_\# \mu, (\theta_l)_\# \nu \right) \right)^{\frac{1}{p}}, \tag{4}$$

where $\{\theta_l\}_{l=1}^{L} \overset{\text{i.i.d.}}{\sim} \sigma$. The Monte Carlo estimator's error decreases as $\frac{1}{\sqrt{L}}$, where $L$ is the number of slices. The main issue becomes how much one can simulate (for large $d$), which proves to be challenging since most slices are known to be non-informative for data supported in low dimensions. As a result, $SW_p(\mu, \nu; \sum_{l=1}^{L} \frac{1}{L} \delta_{\theta_l})$ often underestimates the distance between $\mu$ and $\nu$ in practice. Moreover, $L$ should be sufficiently large compared to $d$, which is undesirable since the time complexity of SW scales linearly with $L$.

## 3 Motivating Related Work

Given this context of SWD, we pause to highlight two sets of works that particularly motivate the present manuscript:

**Subspace-constrained OT.** Recent works propose computing OT in lower-dimensional subspaces (Paty & Cuturi, 2019; Bonet et al., 2021b; Muzellec & Cuturi, 2019) to improve both efficiency and robustness for high-dimensional data. **1) Subspace Detours** (Bonet et al., 2021b; Muzellec & Cuturi, 2019) constrain transport plans to be optimal when projected onto a subspace. This allows efficient extension of low-dimensional transport solutions to the full space. **2) Subspace Robust** (Paty & Cuturi, 2019) considers the worst-case transport cost over all low-dimensional projections. This can be computed by minimizing $S_k^2(\mu, \nu) = \min_{\pi \in \Pi(\mu,\nu)} \sum_{l=1}^{k} \lambda_l(V_\pi)$ where $V_\pi := \int (x-y)(x-y)^T d\pi(x,y)$ is the $2^{\text{nd}}$ order displacement matrix for a coupling $\pi$, and $\lambda_l(V_\pi)$ its $l$-th largest eigenvalue. A related statistical perspective is developed by Niles-Weed & Rigollet (2022), who study Wasserstein estimation under a spiked transport model and analyze a max-subspace sliced construction under a similar low-dimensional-structure assumption.

**Gaussian Sliced-Wasserstein.** Earlier works ((Sudakov, 1978; Diaconis & Freedman, 1984; Reeves, 2017)) establish several central limit theorems showing that under mild conditions, low-dimensional projections of high-dimensional data converge to Gaussians. Nadjahi et al. (2021) leverages this concentration of measure phenomenon and shows that the sliced-Wasserstein distance with Gaussian projection vectors, defined by $\widetilde{SW}_p^p(\mu, \nu) := \int_{\mathbb{R}^d} W_p^p(\theta_\# \mu, \theta_\# \nu) \, d\gamma_d(\theta)$ with $\gamma_d = \mathcal{N}(0, \frac{1}{d} I_d)$, is proportional to the classical SWD: $\widetilde{SW}_p^p(\mu, \nu) = C_{d,p} SW_p^p(\mu, \nu; \mathcal{U}(\mathbb{S}^{d-1}))$, where $C_{d,p} = \left( \frac{2}{d} \right)^{p/2} \frac{\Gamma\left(\frac{d}{2} + \frac{p}{2}\right)}{\Gamma\left(\frac{d}{2}\right)}$, and propose an efficient approximation of the SWD without simulation.

## 4 A Subspace Perspective on Sliced-Wasserstein Distances

Many machine learning problems involve high-dimensional data that has a low-dimensional structure. Formally, this phenomenon, known as the *manifold hypothesis*, states that for a dataset $X \subset \mathbb{R}^d$, there exists a $k$-dimensional manifold $\mathcal{M}$ where $k \ll d$ such that $X$ approximately lies on $\mathcal{M}$ (Fefferman et al., 2016). For instance, rigorous dimensionality estimation methods applied to common datasets like MS-COCO (Lin et al., 2014) and ImageNet (Deng et al., 2009) suggest $k < 50$ (Pope et al., 2021), despite their ambient dimension $d$ being orders of magnitude larger. While these manifolds are generally nonlinear, they admit local linear approximations via their tangent spaces. Moreover, in practice, data features typically have strong linear

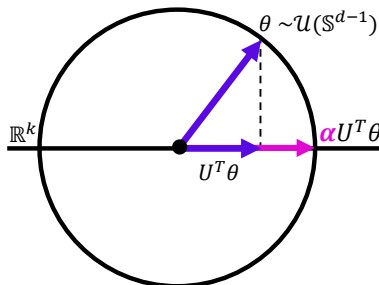

**Figure 1:** An illustration for rescaling the 1D Wasserstein based on slice *informativeness*. Here, the notion of informativeness can be defined as the slice alignment with the (principal) data subspace.

correlations, allowing techniques like Principal Component Analysis (PCA) to identify a principal subspace that captures most of the data variance.

**Motivating challenge.** This subspace approximation is particularly relevant in the context of SWD. A standard concentration phenomenon on the sphere implies that when slices $\theta$ are sampled uniformly from $\mathbb{S}^{d-1}$, the probability that a random slice is nearly orthogonal to any fixed direction increases exponentially with dimension; see, e.g., Vershynin (2018, Theorem 5.1.3). This observation also appears in the SW literature (Kolouri et al., 2019). Specifically, for a unit vector $x_0$ representing a principal direction in the data subspace,

$$\Pr\left(|\langle\theta, x_0\rangle| \leq \epsilon\right) > 1 - e^{-d\epsilon^2}, \quad \theta \sim \mathcal{U}(\mathbb{S}^{d-1}). \tag{5}$$

This concentration of measure phenomenon implies that as dimensionality $d$ grows, most random slices become nearly orthogonal to the principal directions of the data subspace. Consequently, the corresponding 1D Wasserstein distances contribute minimally to the SWD. This effect, which we refer to as *slice non-informativeness*, limits the effectiveness of SWD in high-dimensional spaces.

**Current approaches: Designing the slicing distribution.** Sampling-based methods seek to define a non-uniform slicing distribution that focuses on *discriminative* directions. Optimization-free methods (Nguyen & Ho, 2024; Nguyen et al., 2024b) are objectively faster but do not yield true metrics. Other methods (Nguyen et al., 2020; 2024a) yield proper metrics but are more computationally expensive due to the optimization involved. In the limit, the Max variants use discrete slicing distributions that require global optimality to be metrics, which is generally intractable in practice. Empirically, without careful hyperparameter tuning, the different variants face numerical instability in the larger learning rate regimes, likely because of the overemphasis on directions with large projected distances.

**A rescaling perspective: Rescaling 1D Wasserstein distances.** These challenges in directly redefining the slicing distribution motivate us to take a second look at the conventional wisdom of sampling informative slices. We propose an alternative formulation that reweights each 1D Wasserstein based on the *informativeness* of the corresponding slice/projecting direction (see Figure 1 for illustration). By defining the notion of an *informative slice* based on its alignment with the effective data subspace, we show under Assumption 4.1 that it is possible to reweight all slices by *a global constant* on the SWD. This maintains the efficiency and theoretical properties of the classical Sliced-Wasserstein distance; in other words, by simply rescaling by a global constant, one can make SWD often perform comparably to the aforementioned variants of SWD that go to great lengths to design slicing distributions to accelerate SWD estimation[1]. The implications of this finding for using SWD in gradient-based learning will be discussed in subsequent sections.

To formalize this approach, we introduce the following assumption and definitions:

**Assumption 4.1** (Effective Subspace Structure). *Let $\mu^d, \nu^d \in \mathcal{P}_p(\mathbb{R}^d)$ be probability measures. We say $(\mu^d, \nu^d)$ has k-dimensional effective structure if:*

---

[1]We remark that in the world of Bayesian optimization (BO), an analogous discovery has recently been made. Hvarfner et al. (2024) found that a simple global rescaling of the Gaussian prior lengthscale in vanilla BO allows that algorithm to perform well in high dimensions, despite the conventional wisdom that specialized variants of BO for higher dimensions are needed.

*1. There exists a semi-orthogonal matrix $U \in \mathbb{R}^{d \times k}$ (i.e., $U^T U = I_k$) such that*

$$supp(\mu^d), supp(\nu^d) \subset V_k := col\text{-}span(U).$$

*2. $k$ is minimal, meaning that there does not exist any $U' \in \mathbb{R}^{d \times k'}$ with $k' < k$ s.t (1) holds.*

*We refer to $V_k$ as the **effective subspace** (ES) of $\mu^d, \nu^d$, and $k$ as their **effective dimensionality** (ED).*

*Approximate-subspace regime.* The exact statements below assume a common $k$-dimensional support. Outside this regime, Appendix A.10 gives coarse residual-energy bounds in terms of the mass outside a candidate subspace, together with a compact numerical illustration. These results indicate that the exact equivalence deteriorates continuously as the orthogonal residual energy increases, but we do not claim a sharp extension beyond the exact effective-subspace setting. This low-dimensional-structure viewpoint is also related to the spiked transport setting studied by Niles-Weed & Rigollet (2022).

All results below remain valid if $supp(\mu^d)$ and $supp(\nu^d)$ lie in a common affine subspace $m + V_k$ for some $m \in \mathbb{R}^d$, since both $W_p$ and $SW_p$ are invariant under a common translation of $\mu^d$ and $\nu^d$. For simplicity we state Assumption 4.1 in the linear case.

**Informative slices.** To unify the diverse slicing strategies in the literature, we introduce the concept of a slice informativeness function $\phi : \mathbb{S}^{d-1} \to \mathbb{R}_+$ that quantifies the relevance of a projection direction $\theta \in \mathbb{S}^{d-1}$ (e.g., by assigning a non-negative value) for distinguishing distributions. Existing variants typically define $\phi$ as a functional of the input measure $\mu$ and $\nu$. For instance, Max-SW (Deshpande et al., 2019), Markovian SW Nguyen et al. (2024a), and EBSW (Nguyen & Ho, 2024) implicitly use

$$\phi_{\mu,\nu}(\theta) = W_p^p(\theta_\# \mu, \theta_\# \nu), \tag{6}$$

to measure the informativeness of $\theta$. On the other hand, RPSW (Nguyen et al., 2024b) implicitly uses

$$\phi_{\mu,\nu}(\theta; \gamma_\kappa) = \mathbb{E}_{(X,Y) \sim \mu \times \nu}[\gamma_\kappa(\theta; P_{\mathbb{S}^{d-1}}(X - Y))], \tag{7}$$

where $\gamma_\kappa$ is a location-scale distribution (e.g., vMF (Fisher, 1953; Mardia & Jupp, 2009)) and $P_{\mathbb{S}^{d-1}}$ is the projection onto $\mathbb{S}^{d-1}$.

While these data-dependent definitions are expressive, they inherently require evaluating 1D distances or expectations for slices that may effectively be discarded, incurring unnecessary computational overhead (Nguyen et al., 2024a). Furthermore, coupling the slicing distribution to the input measures introduces complex dependencies that complicate the verification of metric properties, such as the triangle inequality (Nguyen & Ho, 2024). To address this limitation, we propose a notion of informativeness grounded in Assumption 4.1 that decouples slice selection from transport cost.

**Definition 4.2.** *Let $V_k = col(U)$, where $U \in \mathbb{R}^{d \times k}$ has orthonormal columns (i.e., $U^\top U = I_k$). We define the effective subspace (ES)-aligned informativeness function $\phi_U : \mathbb{S}^{d-1} \to [0, 1]$ by*

$$\phi_U(\theta) = \|U^\top \theta\|_2. \tag{8}$$

Geometrically, $\phi_U(\theta)$ measures the magnitude of the projection of $\theta$ onto the effective data subspace $V_k$. A value of $\phi_U(\theta) \approx 1$ indicates that the slice captures maximum variation within the data support while $\phi_U(\theta) \approx 0$ implies orthogonality to the data. The basic linear-algebraic properties of $\phi_U$ are standard and are collected in Appendix A.4 for completeness.

### 4.1 The $\phi$-weighting formulation

Starting from Equation 3, we propose a general formulation for reweighting slice contributions:

$$\widetilde{SW}_p(\mu, \nu; \sigma, \rho) = \left( \int_{\mathbb{S}^{d-1}} \underbrace{\rho(\phi(\theta)) W_p^p(\theta_\# \mu, \theta_\# \nu)}_{\text{Reweighted contribution}} d\sigma(\theta) \right)^{\frac{1}{p}}, \tag{9}$$

where $\rho : [0,1] \to \mathbb{R}_+$ is a $\phi$-**weighting function** that rescales the contribution of each slice based on a general informativeness function $\phi$. This formulation preserves metricity under mild conditions (See Proposition A.5 in the Appendix).

**Example 4.3.** *If the goal were to reweight all slices to be treated as informative[2], an appropriate choice for $\rho$ could be the multiplicative inverse of $\phi(\theta)$ (more informative slices are scaled to be smaller, and vice versa). That is,*

$$\rho(\phi(\theta)) = \begin{cases} \dfrac{1}{\phi(\theta)^p}, & \text{if } \phi(\theta) > 0, \\ 0, & \text{if } \phi(\theta) = 0. \end{cases} \tag{10}$$

**Remark 4.4.** *Equation 9 notably does not rely on Assumption 4.1. By defining the appropriate $\rho(\cdot)$ and $\phi(\cdot)$, the $\phi$-weighting formulation can be seen as a unifying formulation that recovers different SW variants.*

- *We set $\rho \equiv 1$ and obtain the classical Sliced-Wasserstein distance.*

- *We set $\phi_{\mu,\nu}(\theta) = W_p^p(\theta_{\#}\mu, \theta_{\#}\nu)$ and $\rho \equiv 1$, and take $\sigma = \delta_{\theta^\star}$ where $\theta^\star \in \arg\max_{\theta \in \mathbb{S}^{d-1}} \phi_{\mu,\nu}(\theta)$, and recover Max-SW (Deshpande et al., 2019) (when a maximizer exists).*

- *We set $\phi_{\mu,\nu}(\theta) = W_p^p(\theta_{\#}\mu, \theta_{\#}\nu)$, $\rho(r) = \dfrac{f(r)}{\int_{\mathbb{S}^{d-1}} f(W_p^p(\theta_{\#}\mu, \theta_{\#}\nu))\, d\sigma(\theta)}$, $\sigma = \mathcal{U}(\mathbb{S}^{d-1})$ where $f : [0, \infty) \to (0, \infty)$ is an increasing energy function (e.g., $f(x) = e^x$), and recover EBSW (Nguyen & Ho, 2024).*

- *We set $\phi_{\mu,\nu}(\theta) = \mathbb{E}_{(X,Y) \sim \mu \times \nu}[\gamma_\kappa(\theta; P_{\mathbb{S}^{d-1}}(X - Y))]$, $\rho(r) = \dfrac{r}{\int_{\mathbb{S}^{d-1}} \phi_{\mu,\nu}(\theta)\, d\sigma(\theta)}$, $\sigma = \mathcal{U}(\mathbb{S}^{d-1})$, where $\gamma_\kappa$ is a location-scale distribution with parameter $\kappa$, and recover RPSW (Nguyen et al., 2024b).*

### 4.2 Misaligned random projections are implicitly downweighted by a scalar

Under Assumption 4.1, we will show that the 1D Wasserstein corresponding to each random projection is weighted by a scalar related to the (ES-aligned) informativeness of that projection.

**The case for 1D effective subspaces.** Let $V_1 = \text{span}(u)$ where $u \in \mathbb{S}^{d-1}$, and suppose $\text{supp}(\mu^d), \text{supp}(\nu^d) \subset V_1$. Given $\theta \in \mathbb{S}^{d-1}$, we can decompose it uniquely as $\theta = \theta_{V_1} + \theta_{V_1^{\perp}}$, where $\theta_{V_1} = (u^{\top}\theta)u$ and $\theta_{V_1^{\perp}} \perp V_1$. For any $x \in V_1$, we have $x = (x^{\top}u)u$, and $\theta^{\top}x$ can thus be decomposed as:

$$\theta^{\top}x = (\theta_{V_1} + \theta_{V_1^{\perp}})^{\top}x = \theta_{V_1}^{\top}x = (u^{\top}\theta)(u^{\top}x). \tag{11}$$

This implies that for any slice $\theta$, the projected distributions $\theta_{\#}\mu^d$ and $\theta_{\#}\nu^d$ are equivalent (up to scaling) to the distributions obtained by projecting $\mu^d$ and $\nu^d$ onto $u$. Specifically:

$$W_p^p(\theta_{\#}\mu^d, \theta_{\#}\nu^d) = |u^{\top}\theta|^p W_p^p(u_{\#}\mu^d, u_{\#}\nu^d). \tag{12}$$

Thus, misaligned slices (projections) $\theta$, with $|u^T\theta| < 1$, implicitly are downweighted: $W_p^p(\theta_{\#}\mu^d, \theta_{\#}\nu^d) < W_p^p(u_{\#}\mu^d, u_{\#}\nu^d)$.

**Generalizing to higher-dimensional effective subspaces.** We extend the idea from one dimension to a $k$-dimensional subspace $V_k$ and investigate how the reweighting function $\rho(\phi_U(\theta)) = \|U^{\top}\theta\|^{-p}$ adjusts the contributions of slices in higher dimensions.

**Proposition 4.5.** *Under Assumption 4.1, let $\mu^k = (U^{\top})_{\#}\mu^d$ and $\nu^k = (U^{\top})_{\#}\nu^d$ be the pushforward measures in $\mathbb{R}^k$. Then, for any $\theta^d \in \mathbb{S}^{d-1}$, we have that:*

---

[2]Conceptually, we reiterate that SWD often underestimates the distance between two distributions because when slices are not aligned with the data subspace, they are implicitly underweighted based on their misalignment (as will be introduced in Section 4.2), even though those slices are in general informative (they contain some component aligned with the data subspace, when they are not totally orthogonal to it; see Figure 1). To rectify this inherent trait of vanilla SWD, one could consider a desirable reweighting to be one that treats all slices as equally informative. Subsequent sections will further motivate this choice.

$$W_p^p(\theta_\#^d \mu^d, \theta_\#^d \nu^d) = W_p^p((U^\top \theta^d)_\# \mu^k, (U^\top \theta^d)_\# \nu^k) = \|U^\top \theta^d\|^p W_p^p(\theta_\#^k \mu^k, \theta_\#^k \nu^k). \tag{13}$$

*where $\theta^k = \frac{U^\top \theta^d}{\|U^\top \theta^d\|}$ with convention $\theta^k = 0_k$ if $\|U^\top \theta^d\| = 0$.*

*Furthermore, we have that:*

$$SW_p^p\left(\mu^k, \nu^k; \frac{1}{L}\sum_{l=1}^{L}\delta_{\theta_l^k}\right) = \widetilde{SW}_p^p\left(\mu^d, \nu^d; \frac{1}{L}\sum_{l=1}^{L}\delta_{\theta_l^d}, \rho\right) \tag{14}$$

$$SW_p^p\left(\mu^k, \nu^k\right) = \widetilde{SW}_p^p\left(\mu^d, \nu^d; \mathcal{U}(\mathbb{S}^{d-1}), \rho\right) \tag{15}$$

*We adopt the convention $\frac{1}{0}\cdot 0 = 0$ in Equation 14 if $\|U^\top \theta_l^d\| = 0$.*

The proof is in Appendix A.5.

**Implicit downweighting.** Under the conditions of Proposition 4.5, each slice contribution is implicitly downweighted by $\|U^T \theta^d\|^p$. That is, for any $\theta^d \in \mathbb{S}^{d-1}$, we have that $W_p^p(\theta_\#^d \mu^d, \theta_\#^d \nu^d) \leq W_p^p(\mu^k, \nu^k)$. Moreover, the downweighting is maximal if $\theta^d \perp \mathrm{span}(U)$ and vanishes if $\theta^d \in \mathrm{span}(U) \cap \mathbb{S}^{d-1}$.

**Rescaling to equalize informativeness.** Assumption 4.1 gives rise to the fact that each one-dimensional Wasserstein distance $W_p^p(\theta_\#^d \mu^d, \theta_\#^d \nu^d)$ is implicitly downweighted by $\|U^\top \theta^d\|^p$. This observation naturally fits into the proposed $\phi$-weighting formulation, as there is an implicit scaling factor associated with each slice. To counteract it and make all slices equally (ES-aligned) informative, we use the reciprocal weighting function (Equation 10) to compensate for the implicit downweighting of misaligned slices. Then, we have that

$$\tilde{W}_p^p(\theta_\#^d \mu^d, \theta_\#^d \nu^d) = \begin{cases} W_p^p(\theta_\#^k \mu^k, \theta_\#^k \nu^k), & \text{if } \phi_U(\theta^d) > 0, \\ 0, & \text{if } \phi_U(\theta^d) = 0. \end{cases} \tag{16}$$

*where $\theta^k = \frac{U^\top \theta^d}{\|U^\top \theta^d\|}$.*

### 4.3 Subspace Sliced-Wasserstein is rescaled Sliced-Wasserstein

In this section, we will show that the generalized notion of informative slices (as defined in Section 4.2) becomes particularly advantageous for equalizing slice informativeness.

Starting from Equation 13, we integrate both sides over $\theta^d \in \mathbb{S}^{d-1}$ with respect to the uniform measure $\sigma(\theta^d)$ and obtain

$$SW_p^p(\mu^d, \nu^d) = \int_{\mathbb{S}^{d-1}} W_p^p(\theta_\#^d \mu^d, \theta_\#^d \nu^d)\, d\sigma(\theta^d) = \int_{\mathbb{S}^{d-1}} \|U^\top \theta^d\|^p W_p^p\left(\theta_\#^k \mu^k, \theta_\#^k \nu^k\right)\, d\sigma(\theta^d). \tag{17}$$

Note that $\theta^k$ depends on $\theta^d$, and the distribution of $\theta^k$ induced by $\theta^d \sim \sigma$ is uniform over $\mathbb{S}^{k-1}$. We introduce the change of variables from $\theta^d$ to $\theta^k$ and express the integral in terms of $\theta^k$:

$$SW_p^p(\mu^d, \nu^d) = \int_{\mathbb{S}^{k-1}} dT_\# \sigma(\theta^k) \times W_p^p\left(\theta_\#^k \mu^k, \theta_\#^k \nu^k\right) \times \left(\int_{\theta^d : \frac{U^\top \theta^d}{\|U^\top \theta^d\|} = \theta^k} \|U^\top \theta^d\|^p\, d\sigma(\theta^d|\theta^k)\right), \tag{18}$$

where $\sigma(\cdot|\theta^k)$ is the conditional distribution of $\theta^d$, and $T : \theta \mapsto \frac{U^\top \theta}{\|U^\top \theta\|}$ is the mapping from $\theta^d$ to $\theta^k$.

The inner integral over $\theta^d$ can be evaluated as a scaling factor $C_{d,k}$ dependent on $\sigma, \theta^k, U$. When $\sigma = \mathcal{U}(\mathbb{S}^{d-1})$, $C_{d,k}$ is invariant for all $\theta^k$.

Substituting back into Equation 18, and letting $\sigma_k = T_{\#}\sigma = \mathcal{U}(\mathbb{S}^{k-1})$ denote the distribution of $\theta^k$, we obtain

$$SW_p^p(\mu^d, \nu^d) = C_{d,k} \int_{\mathbb{S}^{k-1}} W_p^p\left(\theta_{\#}^k \mu^k, \theta_{\#}^k \nu^k\right) d\sigma_k(\theta^k). \tag{19}$$

Since $\sigma_k(\theta^k)$ integrates to 1 over $\mathbb{S}^{k-1}$, and $W_p^p\left(\theta_{\#}^k \mu^k, \theta_{\#}^k \nu^k\right)$ is integrated over all $\theta^k$, we can express the right-hand side as $C_{d,k} \cdot SW_p^p(\mu^k, \nu^k; \sigma_k)$. Intuitively speaking, this means the *loss of information*[3] is due to an implicit constant factor on $SW_p^p(\mu^d, \nu^d)$, which we denote as the **Effective Subspace Scaling Factor** (ESSF). Thus, rescaling the one-dimensional Wasserstein for all slices via Equation 16 is mathematically equivalent to simply multiplying the SWD by the reciprocal of the ESSF. We proceed further to make this connection explicit by the following theorem.

**Theorem 4.6** (Effective Subspace Scaling Factor). *Let $\mu^d, \nu^d \in \mathcal{P}(\mathbb{R}^d)$ satisfy Assumption 4.1, and define $\mu^k = (U^{\top})_{\#}\mu^d$ and $\nu^k = (U^{\top})_{\#}\nu^d$. Then we have that*

$$SW_p^p(\mu^d, \nu^d) = \frac{C_k}{C_d} \cdot SW_p^p(\mu^k, \nu^k), \tag{20}$$

*where $C_d = 2^{p/2} \frac{\Gamma\left(\frac{d}{2} + \frac{p}{2}\right)}{\Gamma\left(\frac{d}{2}\right)}$ and $C_k$ is defined analogously, with $\Gamma$ denoting the Gamma function.*

Informally, for fixed $k$ and large $d$, the factor $\frac{C_k}{C_d}$ scales like $(k/d)^{p/2}$. Thus, if the ambient dimension doubles while the effective dimension stays fixed, the matched GD/SGD step size is expected to increase by roughly $2^{p/2}$. When $k < d$, assuming $\|U^{\top}\theta_l^d\| \neq 0$ is reasonable since $\mathcal{U}(\mathbb{S}^{d-1})(\{\theta \in \mathbb{S}^{d-1} : U^{\top}\theta = 0\}) = 0$.

The proof is in Appendix A.5.

While Theorem 4.6 establishes the exact scaling law in expectation, we now define an empirical estimator $\widehat{ESSF}(L)$ and bound its convergence to show the property holds for the Monte Carlo estimates used in practical settings (finite numbers of slices).

**Proposition 4.7.** *Let $\mu^d, \nu^d \in \mathcal{P}(\mathbb{R}^d)$ satisfy Assumption 4.1. Consider the empirical estimator $\widehat{ESSF}(L)$ defined as:*

$$\widehat{ESSF}(L) = \frac{1}{L} \sum_{l=1}^{L} \|U^{\top}\theta_l^d\|^p, \tag{21}$$

*where $\{\theta_l^d\}_{l=1}^{L} \overset{i.i.d.}{\sim} \mathcal{U}(\mathbb{S}^{d-1})$. We have that:*

1. *$\mathbb{E}[\widehat{ESSF}(L)] = \frac{C_k}{C_d}$ and $Var(\widehat{ESSF}(L)) = \mathcal{O}\left(\frac{1}{L}\right)$.*

2. *Let*
$$\epsilon_L = \left| SW_p^p\left(\mu^d, \nu^d; \frac{1}{L}\sum_{l=1}^{L}\delta_{\theta_l^d}\right) - \widehat{ESSF}(L) \cdot SW_p^p\left(\mu^k, \nu^k; \frac{1}{L}\sum_{l=1}^{L}\delta_{\theta_l^k}\right) \right|.$$
*Then $\epsilon_L \xrightarrow{a.s.} 0$ as $L \to \infty$.*

3. *There exists a constant $K > 0$ depending only on $\mu^d$ and $\nu^d$ such that for any $\delta > 0$, we have*
$$\mathbb{P}(\epsilon_L < \delta) \geq 1 - \exp\left(-\frac{\delta^2 L}{K^2}\right).$$

The proof of this proposition is in Appendix A.7.

In Section 5.1, we provide empirical results showing how the variance of $\widehat{ESSF}(L)$ changes with $L$.

---

[3]By loss of information, we mean the fact that vanilla SWD reduces the weight of a slice that is not aligned with the data subspace; aggregating over all the slices, this leads to the conclusion computing the SWD in the ambient dimension $d$ gives a value that is in general less than computing the SWD in the subspace dimension $k$ (note that $C_{d,k} \leq 1$).

### 4.4 Informative slicing via learning rate search in first-order optimization

While Assumption 4.1 is reasonable as an effective model for real-world data, in gradient-based learning, the SWD objective is evaluated on empirical measures supported on finite minibatches. At each iteration, the source and target supports contain at most $2B$ points and therefore lie in a low-dimensional affine span.

**Remark 4.8.** *Let $\{x_i\}_{i=1}^{2B} \subset \mathbb{R}^d$ be a minibatch of $2B$ samples ($B$ from source, $B$ from target). Let $X = [x_1, \ldots, x_{2B}] \in \mathbb{R}^{d \times 2B}$ be the corresponding data matrix. Then the samples lie in an affine subspace of dimension at most $k \le \min\{2B - 1, d\}$; after centering, they lie in a linear subspace with the same dimension bound.*

The relevance of this observation is that, in first-order optimization such as stochastic gradient descent (SGD), an SWD objective is differentiated with respect to the locations of these empirical support points. Consequently, the gradients driving SGD depend on how the projected one-dimensional Wasserstein distances vary along directions contained in the effective span of the current empirical measures. Under Assumption 4.1, this induces a systematic scaling of per-slice gradients by the alignment factor $\|U^\top \theta\|^p$, which we make explicit below.

**Proposition 4.9.** *For discrete distributions $\hat{\mu}_d = \sum_{i=1}^{n} q_i^1 \delta_{x_i}$ and $\hat{\nu}_d = \sum_{j=1}^{m} q_j^2 \delta_{y_j}$ satisfying Assumption 4.1, let $\hat{\mu}_k = (U^\top)_\# \hat{\mu}_d$ and $\hat{\nu}_k = (U^\top)_\# \hat{\nu}_d$.*

*Then for any $\theta \in \mathbb{S}^{d-1}$ we have:*

$$\nabla_x W_p^p(\theta_\# \hat{\mu}_d, \theta_\# \hat{\nu}_d) = \|U^\top \theta\|^p \nabla_x W_p^p(\theta_\#^k \hat{\mu}_k, \theta_\#^k \hat{\nu}_k), \tag{22}$$

*where $\theta^k = U^\top \theta / \|U^\top \theta\|$. Furthermore, define the empirical gradient error for each $x_i$ as*

$$\epsilon_L(x_i) := \left\| \nabla_{x_i} SW_p^p \left( \hat{\mu}_d, \hat{\nu}_d; \frac{1}{L} \sum_{l=1}^{L} \delta_{\theta_l^d} \right) - \widehat{ESSF}(L) \cdot \nabla_{x_i} SW_p^p \left( \hat{\mu}_k, \hat{\nu}_k; \frac{1}{L} \sum_{l=1}^{L} \delta_{\theta_l^k} \right) \right\|. \tag{23}$$

*Then the following statements hold:*

1. *$\epsilon_L(x_i) \xrightarrow{\mathbb{P}} 0$ as $L \to \infty$.*

2. *$\mathbb{P}(\|\epsilon_L(x_i)\| \le \epsilon) \ge 1 - 2e^{-\epsilon^2 L/(pq_i^1 K)^2}$, where $K = \max_{x_i, y_j} \|x_i - y_j\|^{p-1} < \infty$.*

We refer readers to the Appendix A.8 for the detailed discussion and proofs.

In particular, note that Equation 22 looks like a scalar times the gradient of the objective function (when using SWD as an objective). In SGD, one updates an iterate by the negative of a scalar—the learning rate—times the gradient of the objective. We can absorb the scaling factor $\|U^\top \theta\|^p$ into the learning rate. As in Equation 23, after aggregating across slices this corresponds to absorbing the global effective-subspace scaling factor into the learning rate once. Thus, *it is mathematically equivalent to rescale the slices by the ESSF or to search for an appropriate learning rate*; standard learning-rate search is therefore sufficient to recover effective optimization behavior. This equivalence is exact for first-order methods whose update is a scalar multiple of the gradient, such as GD and SGD. For Adam-type methods, the same global gradient rescaling also changes the adaptive second-moment normalization, so we treat that setting as beyond the scope of the present theorem.

We emphasize that we do not propose a new learning-rate selection strategy. The choice of how to search over learning rates (e.g., grid or random search (Bergstra & Bengio, 2012), Bayesian optimization (Snoek et al., 2012), bandit-based methods (Li et al., 2018), or standard schedules (Smith, 2017; Loshchilov & Hutter, 2017)) is orthogonal to our contribution and follows standard practice in machine learning. Our central observation is simply that, once learning rate is accounted for, *the apparent performance gap between classical SWD algorithms and more complex variants largely disappears across a broad range of SWD-based learning tasks.* Moreover, since hyperparameter search is already standard in practical workflows, users of SWD in gradient-based optimization pipelines often effectively get informative slices for free.

# 5 Experiments

Our theoretical analysis (particularly Theorem 4.6 and Proposition 4.9) predicts that the primary consequence of high ambient dimensionality is an (approximately) scalar shrinkage of SWD gradient magnitude, up to a residual term that vanishes in probability as $L \to \infty$. Since (S)GD-style first-order updates take the form $x \leftarrow x - \eta g$ where $\eta > 0$ is the learning rate, multiplying $g$ by a scalar is operationally equivalent to rescaling the learning rate $\eta$. Consequently, benchmarks that compare methods at fixed or narrowly tuned learning rates may be insufficient. We therefore design our experiments to reveal the one-dimensional performance basin along $\eta$, i.e., the range of learning rates yielding stable effective convergence. If the degradation of standard SWD is primarily a scaling issue and not a loss of information, we hypothesize that its good basins persist but shift toward larger learning rates as $d/k$ increases. We test this hypothesis in two stages: first on synthetic data where the effective dimension $k$ is controlled, and second on real-world generative tasks where $k$ is unknown and dynamic.

We sweep learning rates for all methods using the same grid; for non-learning-rate hyperparameters we use the default settings from the official implementations. Unless stated otherwise, we fix $L = 50$ slices in learning experiments to match standard SWD practice and to ensure a consistent computational budget across methods. While increasing $L$ reduces Monte Carlo variability in the sliced estimator and its gradient (Propositions 4.7 and 4.9), the population-level scaling effect identified in Theorem 4.6 persists independently of $L$. As shown in Figure 3, the concentration of $\widehat{\mathrm{ESSF}}(L)$ is already visible in our tested $(d, k)$ regimes, and larger $L$ primarily smooths the curves without altering the basin-shift behavior. Accordingly, we report two complementary empirical views: the basin plots serve as the theory-diagnostic view of the predicted shift along the learning-rate axis, while the representative qualitative results in the main text and the quantitative tables reported later in the appendix summarize single training configurations selected from the shared grid under the same official defaults for the remaining hyperparameters. We evaluate on representative SW-based learning tasks that are standard in the sliced-OT literature and are commonly used to claim gains over classical SW. We do not claim that SWD is sufficient for every OT application; settings with known geometric constraints or tasks that depend on a small set of discriminative directions may still benefit from specialized slicing schemes. Appendix Table 1 gives a compact illustration of one such departure from the exact effective-subspace regime.

Further detailed numerical results and additional visualizations are in the Appendix.

## 5.1 Numerical Validation of Main Results

We first validate the two quantitative predictions that drive the rest of the paper: (i) under Assumption 4.1, increasing the ambient dimension $d$ primarily rescales $SW_p^p$ by a constant factor depending only on $(d, k, p)$ (Theorem 4.6); and (ii) in the finite-slice regime, this factor is well-approximated by the empirical estimator $\widehat{\mathrm{ESSF}}(L)$ with concentration as $L$ increases (Proposition 4.7). These two checks isolate the "scalar shrinkage" effect from unrelated modeling choices and from downstream optimization dynamics.

**Verifying Theorem 4.6 for** $p = 1, 2$. We consider two $k$-dimensional isotropic Gaussians embedded in $\mathbb{R}^d$ ($d \geq k$), so the effective subspace dimension is controlled by construction. Theorem 4.6 predicts $SW_p^p(\mu^d, \nu^d) = \frac{C_k}{C_d} SW_p^p(\mu^k, \nu^k)$, hence the ratio $\widehat{C} = \frac{\widehat{SW}_p^p(\mu^d, \nu^d)}{\widehat{SW}_p^p(\mu^k, \nu^k)}$ should depend only on $(d, k, p)$ and match $\frac{C_k}{C_d}$ (up to Monte Carlo error). In particular, for fixed $k$ the ratio should decrease as $d$ increases, and for fixed $d$ it should increase with $k$. We generate 500 samples from each distribution and fix the number of slices to $L = 1000$ to make Monte Carlo error negligible relative to the predicted scaling. We vary $d$ and $k$ as follows: **a)** fix $k = 2$, vary $p \in \{1, 2\}$ and $d \in \{10, 30, 50, 80, 100, 300, 500, 800, 1000\}$; **b)** fix $d = 1000$, vary $p \in \{1, 2\}$ and $k \in \{10, 30, 50, 80, 100, 300, 500, 800, 1000\}$ (averaged over 10 runs). Figure 2 shows that $\widehat{C}$ closely tracks the theoretical $\frac{C_k}{C_d}$ in both regimes, supporting the interpretation that the high-$d$ degradation of classical SWD is captured by a constant downweighting of random slices, not by a change in the underlying 1D transport along the effective subspace.

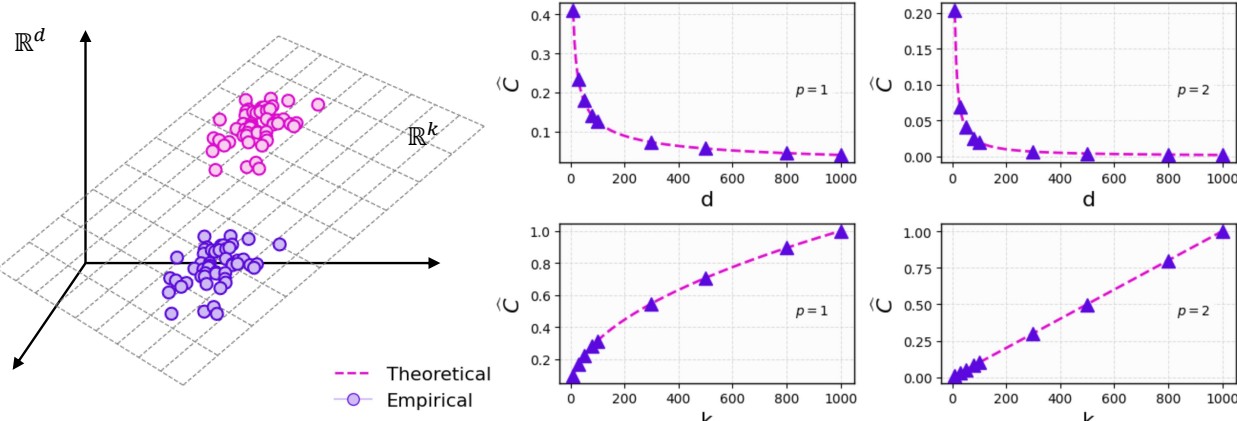

**Figure 2: Left**: Illustration of two embedded Gaussians. **Top row**: Empirical $\widehat{C}$ for varying $d$ with $k = 2$ and $p = 1, 2$. **Bottom row**: Empirical $\widehat{C}$ for varying $k$ at $d = 1000$ with $p = 1, 2$.

**Verifying Proposition 4.7.** Proposition 4.7 states that $\widehat{\mathrm{ESSF}}(L) = \frac{1}{L} \sum_{l=1}^{L} \|U^\top \theta_l^d\|^p$ is an unbiased estimator of $\frac{C_k}{C_d}$ with $\mathrm{Var}(\widehat{\mathrm{ESSF}}(L)) = \mathcal{O}(1/L)$ and exponential concentration. We evaluate $\widehat{\mathrm{ESSF}}(L)$ over 1000 trials for $L \in \{10, 50, 100, 500, 1000, 5000, 10000\}$, using $d \in \{100, 500, 1000\}$ and $k \in \{2, 10, 50\}$ (with $p = 1$). Figure 3 confirms both aspects: the empirical mean matches the theoretical constant and the dispersion shrinks as $L$ increases. This supports treating the loss of informativeness in finite-slice SWD as a stable scalar effect that can be compensated for in optimization (Section 5.2 and Figure 4).

## 5.2 Gradient Flow

**Classic synthetic datasets under ambient embeddings.** We generate 300 target particles from three classic 2D datasets (Swiss roll, 8 Gaussians, Knot) and initialize 300 source particles from a 2D isotropic Gaussian. To isolate the effect predicted by Theorem 4.6 and Proposition 4.9, we embed both source and target into $\mathbb{R}^d$ for $d \in \{2, 50, 100\}$ by padding with zeros and applying a random $d$-dimensional rotation. This procedure increases the ambient dimension while preserving the intrinsic geometry ($k = 2$), so any change in optimization behavior is attributable to the ambient-dimension scaling of the SWD gradient magnitude. We run 10,000 iterations of vanilla gradient descent, reporting results over 3 runs. Learning rates are swept over $\{1, 3, 5, 8\} \times 10^{\{-6, -5, -4, -3, -2, -1, 0, 1, 2\}}$. Consistent with a scalar shrinkage effect, the set of learning rates yielding stable progress for classical SWD does not disappear as $d$ increases; instead it translates toward larger $\eta$, as expected from maintaining an approximately constant effective step size $\eta \cdot \widehat{ESSF}(L)$.

**MNIST and CelebA particle flows.** We further examine this behavior on image-derived empirical measures where the effective dimension $k$ is unknown and may evolve over optimization. For MNIST, we sample 50 images from digit 0 (source) and 50 from digit 1 (target), running gradient flow for 200,000 iterations with learning rates in $\{1, 5\} \times 10^{\{-3, -2, -1, 0, 1, 2, 3\}}$. For CelebA, we initialize particles from Gaussian noise and optimize toward 50 target face images for 200,000 iterations, sweeping learning rates up to 3200. Figure 4(a,b) shows the same qualitative signature: classical SWD exhibits a nontrivial basin of effective learning rates, but its basin is centered at substantially larger $\eta$ than many subspace-aware variants. This matches the interpretation that the dominant gap at default hyperparameters comes from step-size calibration (through gradient scaling) and not from an absence of informative directions.

## 5.3 Color Transfer

We follow the color-transfer setup introduced in Bonet et al. (2024b) and later used in Nguyen et al. (2024a); Nguyen & Ho (2024), with different hyperparameters. Our experiments are performed over 3 image sets (see Figure 5). The optimization uses 50,000 iterations. To reduce computational complexity, we optionally apply K-means clustering with 3,000 clusters, reducing the colorspace into an empirical measure with $N = 3,000$

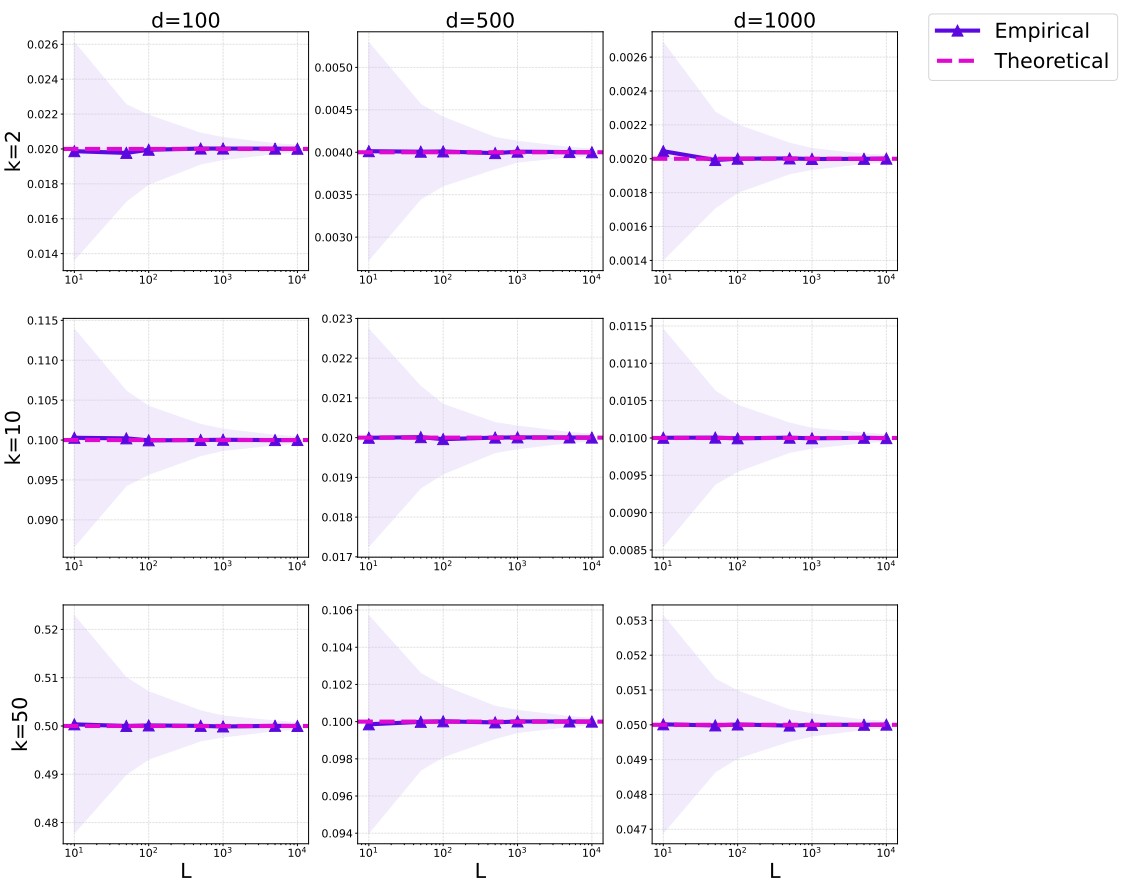

**Figure 3:** Empirical $\widehat{\mathrm{ESSF}}(L)$ for varying $d, k$ over 1000 runs and with $p = 1$. The dotted line depicts the theoretical value.

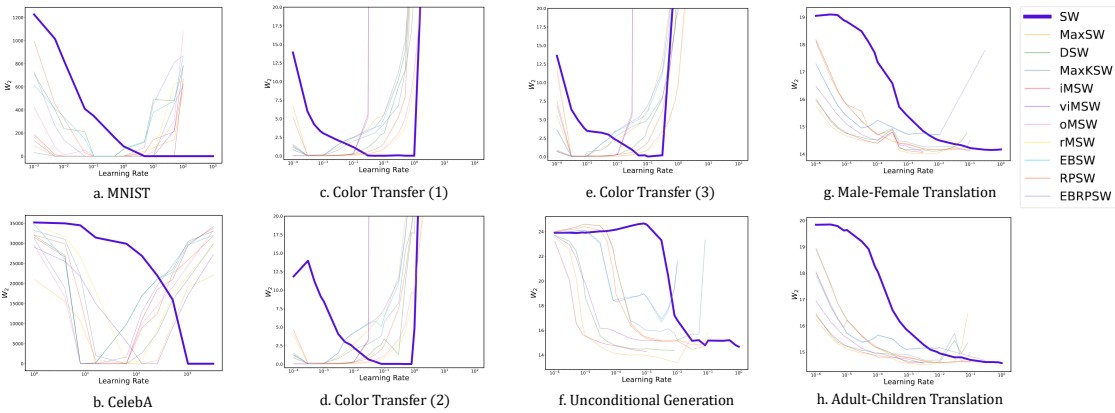

**Figure 4:** Optimal basin plots for gradient flow (a,b), color transfer (c,d,e), and generative modeling (f,g,h).

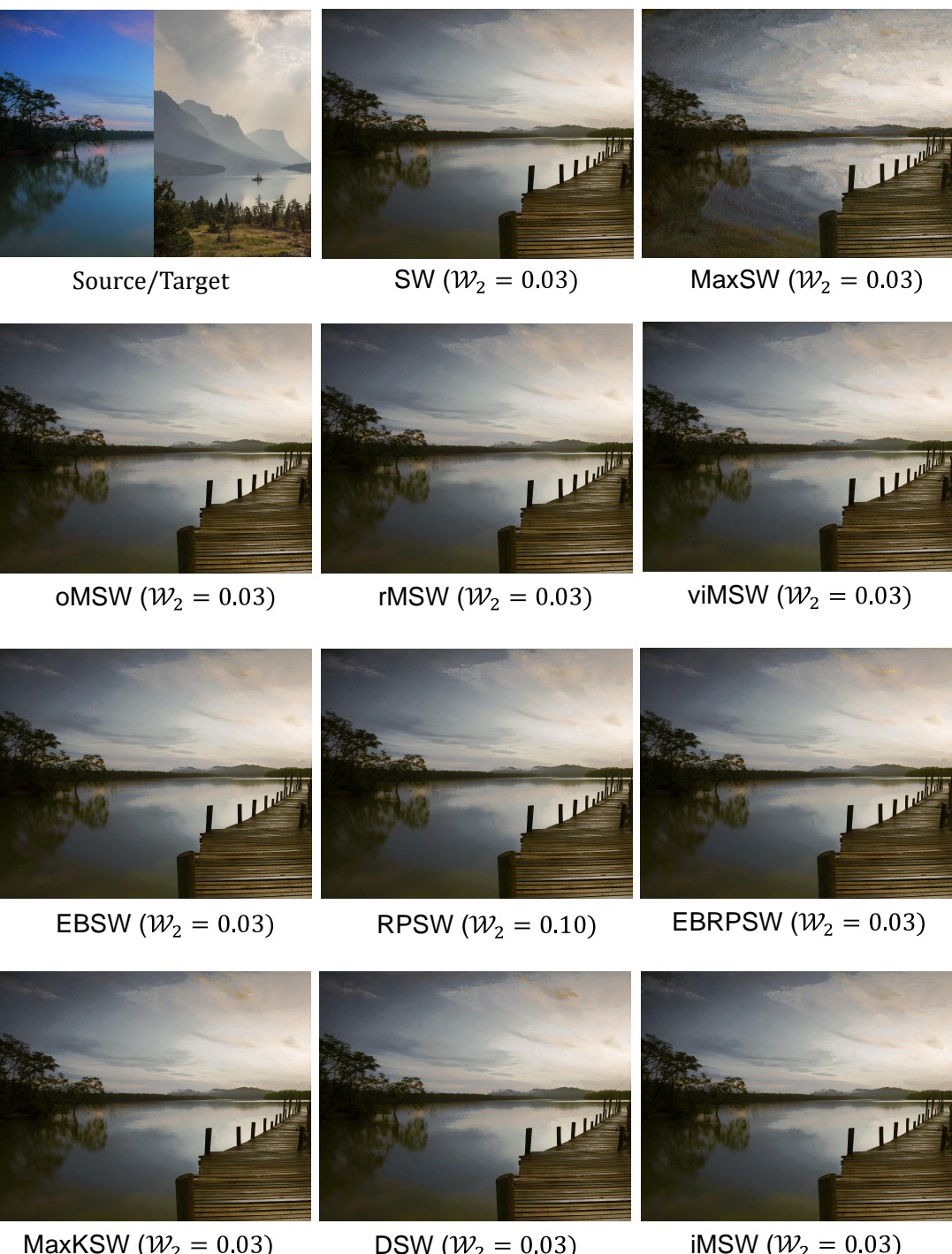

**Figure 5:** Color transfer visualization. Appropriate learning rates are chosen for each method; performing that hyperparameter calibration, all methods are able to perform similarly on the color transfer task.

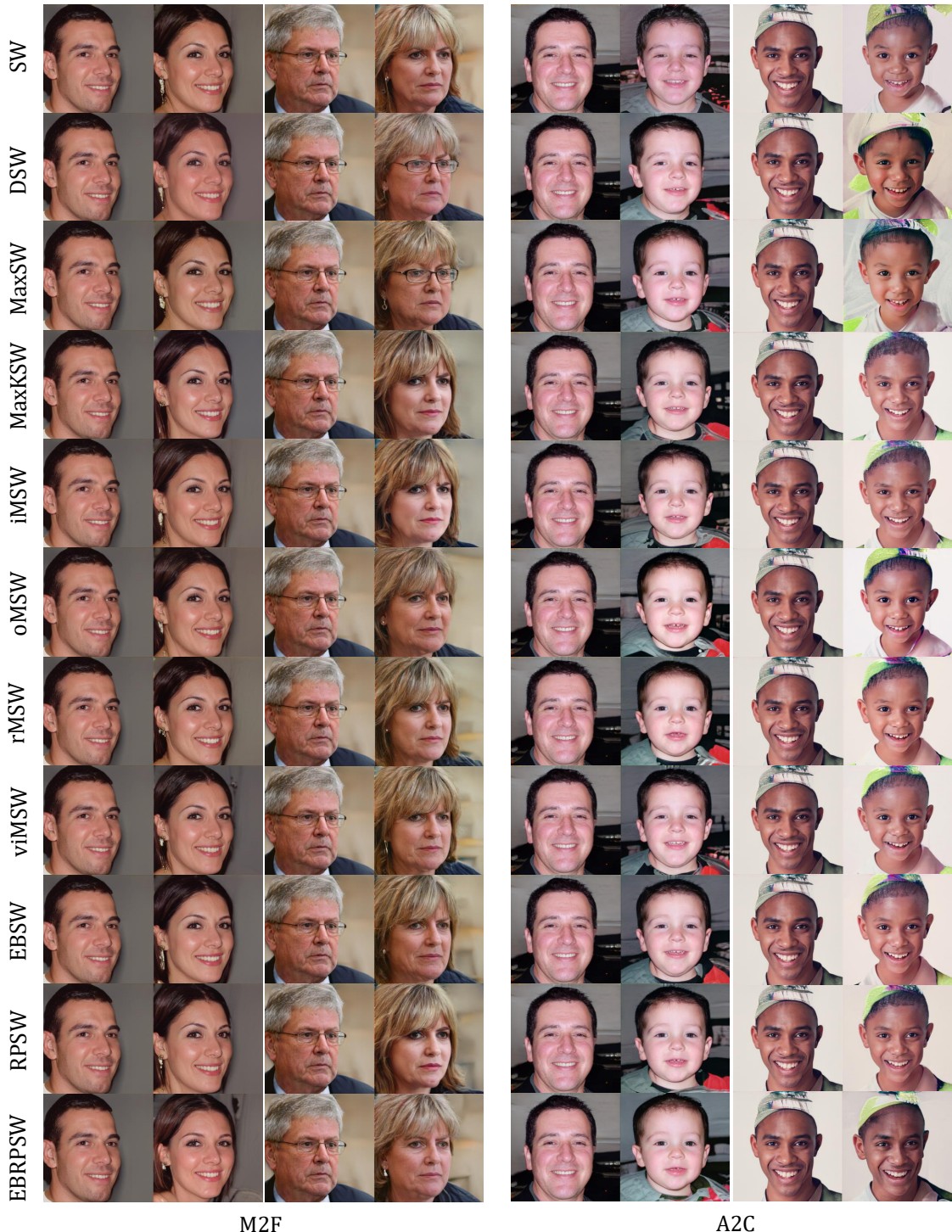

**Figure 6:** Deep generative modeling example, where samples generated using different SWD variants. The left four columns show male-to-female (M2F) examples, while the right four columns show adult-to-child (A2C) examples. We use appropriate learning rates for each method (guided by the results shown in Figure 4). Because of this, all methods perform similarly, demonstrating that tuning learning rates and using classical SWD can be effective as using customized slicing distributions and more complex variants of SWD.

particles. **Learning rates:** $\{1, 3, 5, 8\} \times 10^{\{-4,-3,-2,-1,0,1\}}, 100$. Figure 4(c,d,e) makes the role of $\eta$ explicit: classical SWD attains competitive final objectives, but its effective basin is shifted relative to several variants, and some variants exhibit narrower stability ranges at larger $\eta$. Selecting $\eta$ from the basin therefore separates optimization effects (step-size and stability) from the slicing mechanism itself; with this calibration, the qualitative transfers in Figure 5 are consistent across methods.

### 5.4 Deep Generative Modeling

There exist various generative modeling setups with Sliced-Wasserstein distances (Kolouri et al., 2018; Deshpande et al., 2018; Wu et al., 2019; Liutkus et al., 2019; Nguyen et al., 2024b). We restrict our setup to the latent space ($d = 512$) of an autoencoder (Pidhorskyi et al., 2020) pretrained on the $1024 \times 1024$ FFHQ dataset (Karras et al., 2019). **Learning rates:** $\{1, 3, 5, 8\} \times 10^{\{-6,-5,-4,-3,-2,-1\}}, 1$. In this high-dimensional latent setting, Proposition 4.9 predicts that classical SWD gradients can be substantially smaller in magnitude (via $\widehat{ESSF}(L)$), so recovering comparable optimization dynamics requires larger $\eta$.

We evaluate SWD variants on both unconditional generation and unpaired image-to-image translation tasks. For generation, we follow Deshpande et al. (2018)'s SWG setup using a generator $G_\phi(\cdot)$ to transform $z \in \mathbb{R}^8$ to latents $X \in \mathbb{R}^{512}$. For translation, we modify this to use a residual generator transforming source domain $X$ to target domain $Y$ latents. Following Rombach et al. (2022); Korotin et al. (2023), we operate in an autoencoder's latent space to sidestep the known dimensionality challenges of pixel-space SWG (Deshpande et al., 2018; Nadjahi et al., 2021). We train for 10,000 iterations using vanilla gradient descent with batch size 2048. Figure 4(f,g,h) supports the scaling interpretation directly: classical SWD has a clear basin of effective learning rates, but its basin is shifted toward larger $\eta$ in the $d = 512$ regime. When $\eta$ is chosen within this basin, classical SWD attains objectives comparable to the variants, and the corresponding samples in Figure 6 are qualitatively similar, which is consistent with the view that much of the apparent advantage of specialized slicing shows up as an implicit step-size change.

**Summary.** Our experiments show that random slicing does not lack informative directions. Instead, it induces an approximately scalar rescaling of the SWD value and its first-order signal, through factors of the form $\|U^\top \theta\|^p$ that aggregate into $\widehat{ESSF}(L)$ (with a residual that vanishes as $L$ grows). The Gaussian embedding study validates the implied scaling law at the level of distances and shows that $\widehat{ESSF}(L)$ concentrates around its theoretical limit as $L$ increases. The learning experiments then show that the set of learning rates yielding stable effective optimization for classical SWD persists across tasks, but it shifts toward larger $\eta$ as $d/k$ increases, which is exactly what one expects if the dominant failure mode is step-size miscalibration via gradient shrinkage. Once $\eta$ is chosen within this basin, classical SWD attains comparable objectives and comparable qualitative outputs to methods that explicitly bias slices toward the data subspace, supporting our hypothesis that much of the apparent high-dimensional gap is primarily a rescaling effect and not a loss of information.

## 6 Conclusion

In this paper, we revisit the classical, "vanilla" Sliced-Wasserstein distance and rethink the dominant approach of modifying the slicing distribution to target *informative* directions. We instead view informativeness through a $\phi$-weighting formulation that rescales the one-dimensional Wasserstein contributions. Under an effective subspace model, defining informativeness via alignment with the data subspace yields a Subspace SWD variant that is equivalent to standard SWD up to a single scalar factor. In first-order optimization, the same phenomenon appears at the gradient level: the loss of slice informativeness manifests primarily as an approximately scalar shrinkage of gradient magnitude (up to a finite-slice residual), which is operationally equivalent to a learning-rate rescaling. This provides a direct explanation for why vanilla SWD can match or surpass more complex slicing schemes once standard learning-rate calibration is performed, without additional implementation or loss of metric structure. Our experiments across representative SWD-based learning tasks support this view by showing that the "performance gap" is often a shift in the learning-rate basin, not the disappearance of good solutions. While SWD is not expected to be sufficient for every OT application, especially in settings with known geometric constraints or tasks that rely on a small set of discriminative

directions, our results clarify when and why vanilla SWD remains a strong baseline. Future work can use the same framework to study alternative assumptions on data structure and corresponding choices of the rescaling function $\rho$ and informativeness $\phi$.

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

# A Proofs and Additional Theoretical Results

## A.1 Notation

- $\mathbb{R}^d$: d-dimensional Euclidean space, where $d$ is a positive integer.

- $\mathbb{S}^{d-1} := \{x \in \mathbb{R}^d : \|x\| = 1\}$: unit sphere defined in $\mathbb{R}^d$.

- $\mathcal{P}(\mathbb{R}^d)$: set of all probability measures defined on $\mathbb{R}^d$.

- $\mathcal{P}_p(\mathbb{R}^d)$: set of probability measures whose $p$-th moment is finite, where $p \geq 1$.

- $\mathbb{V}_{k,d}$: set of all $d \times k$ orthogonal matrices, i.e.

$$\mathbb{V}_{k,d} := \{U \in \mathbb{R}^{d \times k} : U^\top U = I_k\}.$$

  Note, $\mathbb{S}^{d-1} = \mathbb{V}_{1,d}$.

- $U = [U[:,1], U[:,2], \ldots U[:,k]] \in \mathbb{V}_{k,d}$: an orthogonal matrix. For each $i \in [1:k]$, $U[:,i] \in \mathbb{R}^d$ is the $i$-th column of $U$.

  Note that $U$ induces a linear function from $\mathbb{R}^d$ to $\mathbb{R}^k$, i.e. $x \mapsto U^\top x$. With abuse of notation, we do not distinguish the matrix $U$ and the corresponding linear mapping.

- $\mathrm{Span}(U)$: The linear subspace spanned by $U$, i.e.

$$\mathrm{Span}(U) := \mathrm{Span}(\{U[:,1], U[:,2], \ldots U[:,k]\}) = \left\{\sum_{i=1}^{k} \alpha_i U[:,i] : \alpha_i \in \mathbb{R}\right\}.$$

- $V_k \subset \mathbb{R}^d$: a $k$-dimensional subspace, where $k$ is a positive integer with $k \leq d$. Note, by classical linear algebra theory, we have

$$V_k = \mathrm{Span}(U)$$

  for some $U \in \mathbb{V}_{d,k}$. **Note, given $V_k$, $U$ is not uniquely determined.**

- $V_k^\perp$: perpendicular complement of $V_k$, which is a subspace of dimension $d - k$.

- $\mu^d, \mu, \nu^d, \nu \in \mathcal{P}(\mathbb{R}^d)$: probability measures in $d$-dimensional space.

- $\mathcal{L}^d$: Lebesgue measure in $\mathbb{R}^d$.

- $C_0(\mathbb{R}^d)$: set of all continuous functions defined on $\mathbb{R}^d$ which vanish at infinity.

- $f_\mu = \frac{d\mu^d}{d\mathcal{L}^d}$: density of $\mu$, that is, for all test functions $\phi \in C_0(\mathbb{R}^d)$:

$$\int_{\mathbb{R}^d} \phi(x) d\mu^d(x) = \int_{\mathbb{R}^d} f_\mu(x) \phi(x) dx.$$

- $X \sim \mu$: A random variable/vector $X$ following distribution $\mu$. We say $X$ is a **realization** of $\mu$.

- $\mathbb{E}[X] := \mathbb{E}[\mu]$, where $X \sim \mu$: expected value of $X$, i.e.

$$\mathbb{E}_\mu[X] = \int_{\mathbb{R}^d} x d\mu(x).$$

- $m_k(\mu)$: $k$-th moment of measure $\mu$. That is, given realization $X \sim \mu$, $m_k(\mu)$ is defined by

$$m_k(\mu) := \mathbb{E}[X^k]$$

- $Var(X) := \mathbb{E}[(X - \mathbb{E}(X))^\top (X - \mathbb{E}(X))]$: the covariance matrix of $X$ (or the measure $\mu$).

- $T_\# \mu$, where $T : \mathbb{R}^d \to \mathbb{R}^d$ is a function: push-forward measure $\mu$ under mapping $T$. That is, for all Borel sets $A \subset \mathbb{R}^d$, we have

$$T_\# \mu(A) = \mu(T^{-1}(A)).$$

  Equivalently speaking, suppose $X \sim \mu$ is a realization of $\mu$; then, $T(X) \sim T_\# \mu$.

- $\mathcal{N}(e, \Sigma)$: Gaussian distribution, where $e \in \mathbb{R}^d$ is the expected value, $\Sigma \in \mathbb{R}^{d \times d}$ is the covariance matrix.

- $0_d$: $d \times 1$ vector where each entry is 0. Similarly, we define $1_d$.

- $I_d$: $d \times d$ identity matrix.

- $\mathcal{U}(\mathbb{S}^{d-1})$: Uniform distribution defined on $\mathbb{S}^{d-1}$.

- $\theta^d \sim \mathcal{U}(\mathbb{S}^{d-1})$: a $d$-dimensional random vector. We say $\theta^d$ is a **realization** of $\mathcal{U}(\mathbb{S}^{d-1})$.

- $\theta, \theta^d, \theta^g$: a $d$-dimensional vector.

- $\theta^k$: a $k$-dimensional vector.

- $P_{V_k} := P_U$, where $V_k = \mathrm{Span}(U)$: the projection mapping from $\mathbb{R}^d$ into subspace $V_k$, i.e.

$$P_{V_k}(x) := P_U(x) = UU^\top x, \forall x \in \mathbb{R}^d.$$

  Note, in this case: the mapping $U : \mathbb{R}^d \to \mathbb{R}^k$ with $x \mapsto U^\top x$ is the corresponding parameterization function of projection $P_U$.

- $\Gamma(\mu, \nu)$: set of joint measures whose marginals are $\mu, \nu$ respectively:

$$\Gamma(\mu, \nu) := \{ \gamma \in \mathcal{P}((\mathbb{R}^d)^2) : (\pi_1)_\# \gamma = \mu, (\pi_2)_\# \gamma = \nu \},$$

  where $\pi_1 : (x, y) \mapsto x, \pi_2 : (x, y) \mapsto y$ are canonical projection mappings.

- $W_p^p(\mu, \nu)$: Wasserstein problem between $\mu$ and $\nu$:

$$W_p^p(\mu, \nu) := \inf_{\gamma \in \Gamma(\mu, \nu)} \int_{(\mathbb{R}^d)^2} \|x - y\|^p d\gamma(x, y)$$

- $SW(\mu, \nu; \sigma)$, where $\sigma \in \mathcal{P}(\mathbb{S}^{d-1})$: Sliced Wasserstein problem between $\mu$ and $\nu$ with respect to reference measure $\sigma$:

$$SW_p^p(\mu, \nu; \sigma) := \int_{\mathbb{S}^{d-1}} W_p^p(\theta_\# \mu, \theta_\# \nu) d\sigma(\theta)$$

- $\phi_U : \mathbb{S}^{d-1} \to \mathbb{R}_+$: ES-informative aligned mapping. A measurable mapping which describes the information of the projected $\theta$ on the space spanned by $U$.

- $\widetilde{SW}_p^p(\mu, \nu; \sigma, \rho)$: rescaled sliced-Wasserstein objective:

$$\widetilde{SW}_p^p(\mu, \nu; \sigma, \rho) := \int_{\mathbb{S}^{d-1}} \rho(\phi_U(\theta)) W_p^p(\theta_\# \mu, \theta_\# \nu) d\sigma(\theta)$$

  where $\rho : \mathbb{R}_+ \to \mathbb{R}_+$ is a rescaling function. In this paper, we set $\rho$ as the following decreasing function:

$$\rho(x) = \frac{1}{x^p}$$

  When $x = 0$, we adopt the convention $\rho(x) = 0$.

**Remark A.1.** *In this paper, we adopt the following convention.*

*We do not distinguish the scalar/vector/matrix and the corresponding induced linear mapping. For example, $\theta \in \mathbb{R}^d$, induces the mapping*

$$\mathbb{R}^d \ni x \mapsto \theta^\top x \in \mathbb{R}.$$

- *When $\theta$ is a random vector, we refer to it as a "random projection mapping" in both the main text and the appendix. We adopt the same convention for the scalar notation $\alpha$ and the matrix notation $U$.*

- *We use $\theta_\# \mu$ to denote the push-forward measure induced by mapping $x \mapsto \theta^\top x$. Similarly, $(\theta \times \theta)_\# \gamma$ denotes the push-forward measure of joint measure $\gamma \in \mathcal{P}((\mathbb{R}^d)^2)$ induced by mapping $(x, x') \mapsto (\theta^\top x, \theta^\top x')$. The same convention is adopted for $\alpha, U$.*

**Remark A.2.** *For simplicity, in notation $SW(\mu, \nu; \sigma)$, we may relax the restriction that $\sigma$ is a probability measure. We allow $\sigma$ to be a finite positive measure in the main text and appendix.*

## A.2 Wasserstein distances in $\mathbb{R}^d$ and $\mathbb{R}^k$

In the present manuscript, we assume the probability measures $\mu^d, \nu^d \in \mathcal{P}_p(\mathbb{R}^d)$ are supported in a lower dimensional subspace. We refer to Assumption 4.1 for details.

Let $P_U$ denote the projection mapping from $\mathbb{R}^d$ to $V_k$:

$$P_U(x) = UU^\top x, \forall x \in \mathbb{R}^d, \tag{24}$$

Then, the corresponding lower-dimensional parameterization mapping is defined as:

$$x \mapsto U^\top x, \forall x \in \mathbb{R}^d. \tag{25}$$

By classical linear algebra theory, it is straightforward to verify the following:

**Proposition A.3.** *[Basic properties of linear projection] Let $P_U, U$ be defined above, then we have:*

(1) *For each $\theta \in \mathbb{R}^d$, $\theta$ can be uniquely decomposed into $V_k, V_k^\perp$, i.e. $\theta = \theta_{V_k} + \theta_{V_k^\perp}$, where $\theta_{V_k} = P_U(\theta) \in V_k, \theta_{V_k^\perp} \in V_k^\perp$.*

(2) *For all $x \in V_k$, $P_U(x) = x$.*

(3) *If we restrict $U$ to the subspace $V_k$, denoted as $U \mid_{V_k}$, then $U \mid_{V_k} : V_k \to \mathbb{R}^k$ is a bijection. The inverse is given by*

$$\left(U \mid_{V_k}\right)^{-1}(y) = Uy, \quad \forall y \in \mathbb{R}^k.$$

*In addition, $\|U^\top x\| = \|x\|$ for all $x \in V_k$.*

*Proof.* It follows directly from the definitions of $P_U, U$. $\qquad \square$

Let $\mu^k = (U^\top)_\# \mu^d$, $\nu^k = (U^\top)_\# \nu^d$, the above proposition directly induces the following relation between the Wasserstein distance between $\mu^d, \nu^d$ and the Wasserstein distance between $\mu^k, \nu^k$.

**Proposition A.4.** *Under Assumption 4.1, we have the following:*

(1) *$\mu^d$ can be recovered by the inverse of $U \mid_{V_k}$, i.e.*

$$\mu^d = U_\# \mu^k.$$

*(2) The mapping*

$$\Gamma(\mu^d, \nu^d) \ni \gamma^d \mapsto \gamma^k := (U^\top \times U^\top)_\# \gamma^d \in \Gamma(\mu^k, \nu^k), \tag{26}$$

*is a well-defined bijection, where $U^\top \times U^\top$ is defined as*

$$\mathbb{R}^d \times \mathbb{R}^d \ni (x, x') \mapsto (U^\top x, U^\top x') \in \mathbb{R}^k \times \mathbb{R}^k. \tag{27}$$

*(3) The Wasserstein distance is preserved via the lower-dimensional parameterization:*

$$W_p^p(\mu^d, \nu^d) = W_p^p((P_U)_\# \mu^d, (P_U)_\# \nu^d) = W_p^p(\mu^k, \nu^k) \tag{28}$$

*Proof.* Let $X \sim \mu^d$ be a realization.

(1) We have $U^\top X \sim \mu^k$ since $\mu^k = (U^\top)_\# \mu^d$. In addition, by Assumption 4.1, we have $X = UU^\top X$, thus $UU^\top X \sim \mu^d$. That is $U_\# \mu^k = \mu^d$.

(2) Pick $\gamma^d \in \Gamma(\mu^d, \nu^d)$, we have

$$(\pi_1)_\# (U^\top \times U^\top)_\# \gamma^d = (U^\top)_\#((\pi_1)_\# \gamma^d) = (U^\top)_\# \mu^d = \mu^k$$

Similarly, $(\pi_2)_\# (U^\top \times U^\top)_\# \gamma^d = \nu^k$. Thus the mapping defined in Equation 26 is well-defined. Moreover, from statement (1), we have

$$\Gamma(\mu^k, \nu^k) \ni \gamma^k \mapsto (U \times U)_\# \gamma^k \in \Gamma(\mu^d, \nu^d) \tag{29}$$

is well-defined.

Next, we show that this mapping is the inverse of Equation 26.

Let $(X, Y) \sim \gamma^d$ be a realization. Since $\operatorname{supp}(\mu^d), \operatorname{supp}(\nu^d) \subset V_k$, we have $(X, Y) = (UU^\top X, UU^\top Y)$ almost surely. Hence

$$(U \times U)_\# (U^\top \times U^\top)_\# \gamma^d = \gamma^d.$$

Conversely, if $(X', Y') \sim \gamma^k$, then $U^\top U = I_k$ implies

$$(U^\top \times U^\top)_\# (U \times U)_\# \gamma^k = \gamma^k.$$

Thus, the mapping in Equation 29 is the inverse of the mapping in Equation 26, and Equation 26 is a bijection.

(3) By Proposition A.3 (2), for each $x \in \operatorname{supp}(\mu^d) \subset V_k$, we have $P_U(x) = x$, thus $(P_U)_\# \mu^d = \mu^d$. Similarly, $(P_U)_\# \nu^d = \nu^d$. Thus we obtain the first equality:

$$W_p^p(\mu^d, \nu^d) = W_p^p((P_U)_\# \mu^d, (P_U)_\# \nu^d).$$

For the second equality, we first pick $\gamma^d \in \Gamma(\mu^d, \nu^d)$ and let $\gamma^k = (U^\top \times U^\top)_\# \gamma^d$. By statement (2), we have $\gamma^k \in \Gamma(\mu^k, \nu^k)$.

$$\int_{(\mathbb{R}^d)^2} \|x - y\|^p d\gamma^d(x, y)$$

$$= \int_{(\mathbb{R}^d)^2} \|U^\top x - U^\top y\|^p \, d\gamma^d(x, y)$$

$$= \int_{(\mathbb{R}^k)^2} \|x' - y'\|^p \, d(U^\top \times U^\top)_\# \gamma^d(x', y')$$

$$= \int_{(\mathbb{R}^k)^2} \|x' - y'\|^p d\gamma^k(x', y')$$

where the first equality follows from Proposition A.3 (3), the second equality follows from the definition of push-forward measure, the third equality holds from statement (2).

Combining the above equality with statement (2), we obtain

$$
\begin{aligned}
W_p^p(\mu^d, \nu^d) &= \inf_{\gamma^d \in \Gamma(\mu^d, \nu^d)} \int_{\mathbb{R}^d} \|x - y\|^p d\gamma^d(x, y) \\
&= \inf_{\gamma^k \in \Gamma(\mu^k, \nu^k)} \int_{\mathbb{R}^k} \|x' - y'\|^p d\gamma^k(x', y') \\
&= W_p^p(\mu^k, \nu^k)
\end{aligned}
$$

$\square$

**Proposition A.5.** *Let $\sigma = \mathcal{U}(\mathbb{S}^{d-1})$ and let $\phi : \mathbb{S}^{d-1} \to \mathbb{R}_+$ and $\rho : \mathbb{R}_+ \to \mathbb{R}_{\geq 0}$ be measurable and define $m$ on $\mathbb{S}^{d-1}$ by*

$$
dm(\theta) = (\rho \circ \phi)(\theta) d\sigma(\theta). \tag{30}
$$

*If $0 < m(\mathbb{S}^{d-1}) < \infty$ and $(\rho \circ \phi)(\theta) > 0$ for $\sigma$-a.e. $\theta$, then $\widetilde{SW}_p(\cdot, \cdot; \sigma, \rho)$ is a metric on $\mathcal{P}_p(\mathbb{R}^d)$.*

*Proof.* Non-negativity and symmetry are immediate. For the triangle inequality, we note that for each $\theta \in \mathbb{S}^{d-1}$,

$$
W_p(\theta_\# \mu, \theta_\# \xi) \leq W_p(\theta_\# \mu, \theta_\# \nu) + W_p(\theta_\# \nu, \theta_\# \xi), \tag{31}
$$

and applying Minkowski's inequality to $\theta \mapsto W_p(\theta_\# \mu, \theta_\# \nu)$ in $L^p(\mathbb{S}^{d-1}, m)$ yields the result.

For the identity of indiscernibles, we have that $\widetilde{SW}_p(\mu, \nu) = 0$ implies

$$
W_p(\theta_\# \mu, \theta_\# \nu) = 0 \tag{32}
$$

for $m$-a.e. $\theta$.

Since $\rho(\phi(\theta)) > 0$ for $\sigma$-a.e. $\theta$ and $dm = (\rho \circ \phi)d\sigma$, the measures $m$ and $\sigma$ are mutually absolutely continuous. Hence the above identity also holds for $\sigma$-a.e. $\theta$, and Cramér–Wold implies $\mu = \nu$.

$\square$

**Remark A.6.** *For the ES-aligned choice $\phi_U(\theta) = \|U^\top \theta\|_2$ and $\rho(x) = x^{-p}$ (with $\rho(0) = 0$), the condition $m(\mathbb{S}^{d-1}) < \infty$ requires $p < k$. Indeed, for $\theta \sim \sigma$ we have $\|U^\top \theta\|_2^2 \sim \mathrm{Beta}(k/2, (d-k)/2)$, and the negative moment exists iff $\frac{p}{2} < \frac{k}{2}$. This does not affect Theorem 4.6 or Proposition 4.9, which relies only on the scaling relationship in expectation.*

### A.3 Background: Relationship between the Gaussian and Spherical Uniform Distribution

In this section, we introduce basic properties of multivariate Gaussian and the relation between Gaussian and spherical uniform distribution.

First we consider 1D space $\mathbb{R}$, choose $e \in \mathbb{R}$ and $\sigma > 0$, the Gaussian distribution, denoted as $\mathcal{N}(e, \sigma^2)$, is the probability measure whose density is defined by

$$
f(x) := \frac{1}{\sqrt{2\pi\sigma^2}} e^{-\frac{(x-e)^2}{2\sigma^2}},
$$

where $e, \sigma^2$ are the expected value and variance of $X$ respectively.

When $e = 0, \sigma^2 = 1$, the induced measure is called standard (1D) Gaussian distribution, whose density is given by

$$f(x) := \frac{1}{\sqrt{2\pi}} e^{-\frac{x^2}{2}} \tag{33}$$

In space $\mathbb{R}^d$, the above density function can be generalized as:

$$f(x) := \frac{1}{(2\pi)^{d/2}} e^{-\frac{\|x\|^2}{2}} \tag{34}$$

and the induced distribution is called $d$**-dimensional Standard Gaussian distribution**.

Given $e \in \mathbb{R}^d$ and positive definite $d \times d$ matrix, $\Sigma = AA^T$ where $A \in \mathbb{R}^{d \times k}$, the Gaussian distribution is denoted as $\mathcal{N}(e, \Sigma)$, can be defined by the following well-known proposition:

**Proposition A.7** (Definition of Gaussian distribution). *Let $X \sim \mathcal{N}(e, \Sigma)$ be a realization, then the following are equivalent:*

- $\mathcal{N}(e, \Sigma)$ *is Gaussian distribution, with expected value $e$ and covariance matrix $\Sigma$.*

- $X = AG + e$, *where $G \sim \mathcal{N}(0, I_d)$, whose density is defined by Equation 34.*

- $\forall \theta \in \mathbb{R}^d$, $\theta^\top X$ *is a 1D Gaussian variable:*

$$\theta^\top X \sim \mathcal{N}(\theta^\top e, (\theta^\top A)^\top (\theta^\top A)).$$

From the proposition, it is straightforward to verify the following:

**Proposition A.8** (Basic properties of Gaussian distribution). *Suppose $X \sim \mathcal{N}(e, \Sigma)$, then we have:*

*(1) If $rank(\Sigma) = d$, then $\mathcal{N}(e, \Sigma)$ admits the density function:*

$$f(x) = \frac{1}{(2\pi)^{d/2} det(\Sigma)^{1/2}} e^{-\frac{(x-e)^T \Sigma^{-1}(x-e)}{2}}$$

*(2) Choose $B \in \mathbb{R}^{d \times k}, \beta \in \mathbb{R}^k$, and let $T_{B,e,\beta}(x) := B(x - e) + \beta$, then we have*

$$B(X - e) + \beta \sim (T_{B,e,\beta})_\# \mathcal{N}(e, \Sigma) = \mathcal{N}(\beta, B^\top \Sigma B).$$

*(3) Suppose $Z \sim \mathcal{N}(0, I_d)$, then the absolute p-th power of $Z$ is given by*

$$\mathbb{E}[\|Z\|^p] = 2^{p/2} \frac{\Gamma(\frac{p+d}{2})}{\Gamma(d/2)}.$$

*(4) Suppose $Z \sim \mathcal{N}(0, I_d)$, then $r = \|Z\|, \theta = \frac{Z}{\|Z\|}$ are independent.*

At the end of this section, we introduce the following relation between the Gaussian distribution and the spherical uniform distribution.

**Proposition A.9.** *We define the following function $f$ with*

$$\mathbb{R}^d \setminus \{0\} \ni x \mapsto f(x) = \frac{x}{\|x\|}.$$

*Suppose $\Sigma = AA^\top$ is a full rank positive-semi-definite matrix, then we have*

$$f_\# \mathcal{N}(0_d, \Sigma) = \mathcal{U}(\mathbb{S}^{d-1}).$$

*Proof.* Let $X \sim \mathcal{N}(0_d, \Sigma)$ be a realization of the $d$-dimensional Gaussian, $\Theta = f(X) = \frac{X}{\|X\|}$. Note that $\Theta$ is well defined $\mathcal{N}(0_d, \Sigma)$-a.s.

**Step 1**. Suppose $\Sigma = I_d$, it is equivalent to the following:

Suppose $X_1, \ldots X_d \overset{\text{i.i.d.}}{\sim} \mathcal{N}(0, 1)$ and $\Theta = [\frac{X_1}{\sqrt{\sum_{i=1}^d X_i^2}}, \ldots, \frac{X_d}{\sqrt{\sum_{i=1}^d X_i^2}}]^T$, then $\Theta \sim \text{Unif}(\mathbb{S}^{d-1})$. It is a standard result in probability theory. In particular, choose test function $\phi \in C_0(\mathbb{S}^{d-1})$, we have:

$$
\begin{aligned}
\mathbb{E}[\phi(\Theta)] &= \int_{\mathbb{R}^d} \phi(\frac{x}{\|x\|}) f_X(x) dx \\
&= \frac{1}{(2\pi)^{d/2}} \int_{\mathbb{R}^d} \phi\left(\frac{x}{\|x\|}\right) e^{-\frac{\|x\|^2}{2}} dx \\
&= \frac{1}{(2\pi)^{d/2}} \int_{\mathbb{S}^{d-1}} \int_{\mathbb{R}_+} \phi(\theta) e^{-r^2/2} r^{d-1} d\theta dr \qquad r, \theta \text{ are spherical coordinates} \\
&= \int_{\mathbb{S}^{d-1}} \phi(\theta) d\theta \cdot \underbrace{\frac{1}{(2\pi)^{d/2}} \int_{\mathbb{R}_+} e^{-r^2/2} r^{d-1} dr}_{1/\|\mathbb{S}^{d-1}\|}
\end{aligned}
$$

Thus, $\Theta \sim \text{Unif}(\mathbb{S}^{d-1})$.

**Step 2**. Suppose $\Sigma = \text{diag}(\sigma_1, \ldots \sigma_d)$ where $\sigma_1, \ldots \sigma_d > 0$, we have

$$
\Theta = \frac{X}{\|X\|} = \frac{\Sigma^{-1/2} X}{\|\Sigma^{-1/2} X\|},
$$

where $\Sigma^{-1/2} X \sim \mathcal{N}(0, I_d)$. Thus, by step 1, we have $\Theta \sim \mathcal{U}(\mathbb{S}^{d-1})$.

**Step 3**. We consider the general positive definite $\Sigma$. We have $\Sigma = U \Lambda U^\top$ where $U \in \mathbb{V}_{d,d}$ is orthonormal matrix.

We have

$$
U^\top \Theta = \frac{U^\top X}{\|X\|} = \frac{U^\top X}{\|U^\top X\|}
$$

Since $U^\top X \sim \mathcal{N}(0, \Lambda)$ and $\Lambda$ is a positive diagonal matrix, then from step 2, we have $U^\top \Theta \sim \mathcal{U}(\mathbb{S}^{d-1})$. Thus, $\Theta = U(U^\top \Theta) \sim \mathcal{U}(\mathbb{S}^{d-1})$.

$\square$

**Remark A.10.** *Note that the above statement (especially the statement in Step 1) is a well-known result, and that is why the isotropic Gaussian distribution is called a "rotationally invariant distribution." We do not claim this proposition or its proof as contributions of this article; we include the proof for completeness.*

### A.4 Basic properties of the ES-aligned informativeness map

**Remark A.11.** *The next lemma is a basic linear-algebra fact. We do not view it as a contribution; we record it only for completeness because it is used to interpret the ES-aligned informativeness map.*

**Lemma A.12.** *Let $U \in \mathbb{R}^{d \times k}$ satisfy $U^\top U = I_k$ and define $\phi_U(\theta) = \|U^\top \theta\|_2$ for $\theta \in \mathbb{S}^{d-1}$. Then: **(a)** $0 \le \phi_U(\theta) \le 1$ for all $\theta \in \mathbb{S}^{d-1}$; **(b)** $\phi_U(\theta) = 1$ iff $\theta \in \text{span}(U) \cap \mathbb{S}^{d-1}$ and $\phi_U(\theta) = 0$ iff $\theta \perp \text{span}(U)$; and **(c)** for any orthogonal matrix $Q \in \mathbb{R}^{k \times k}$, $\phi_{UQ}(\theta) = \phi_U(\theta)$.*

### A.5 Relationship between the SWD in $\mathbb{R}^d$ and $\mathbb{R}^k$

In this section, we discuss the proof of Theorem 4.6. We first introduce some intermediate results in the following subsection.

### A.5.1 Relationship between $SW_p^p(\mu^d, \nu^d; \mathcal{U}(\mathbb{S}^{d-1}))$ and $SW_p^p(\mu^d, \nu^d; \mathcal{N}(0, I_d))$

The main result in this section is the following proposition

**Proposition A.13.** *Choose $\mu, \nu \in \mathcal{P}(\mathbb{R}^d)$, we have*

$$2^{p/2} \frac{\Gamma(\frac{p+d}{2})}{\Gamma(d/2)} SW_p^p(\mu, \nu; \mathcal{U}(\mathbb{S}^{d-1})) = SW_p^p(\mu, \nu; \mathcal{N}(0, I_d)) \tag{35}$$

**Remark A.14.** *The scaling identity underlying Proposition A.13 is classical; see, for example, Peyré et al. (2019, Remark 2.19). If we replace $\mathcal{N}(0, I_d)$ by $\mathcal{N}(0, \frac{1}{d}I_d)$, the corresponding conclusion has been proved by (Nadjahi et al., 2021, Proposition 1). Our proof below uses the same Gaussian-spherical decomposition and a related change-of-variables argument. We therefore do not claim this statement or its proof as part of the contribution in this paper; we include it only because it is directly used in the later equivalence argument.*

To prove the above statement, first it is straightforward to verify the following:

**Lemma A.15.** *Given $\alpha \in \mathbb{R}$, with abuse of notations, we let $\alpha_{\#}\mu$ denote the pushforward measure of $\mu$ under mapping $x \mapsto \alpha x$, then we have*

$$|\alpha|^p W_p^p(\mu, \nu) = W_p^p(\alpha_{\#}\mu, \alpha_{\#}\nu) \tag{36}$$

*Proof.* If $\alpha = 0$, then both sides are zero, and we've done.

If $\alpha \neq 0$, it is straightforward to verify the following is a well-defined bijection:

$$\Gamma(\mu, \nu) \ni \gamma \mapsto (\alpha \times \alpha)_{\#}\gamma \in \Gamma(\alpha_{\#}\mu, \alpha_{\#}\nu) \tag{37}$$

where $(\alpha \times \alpha)$ denotes the mapping

$$(\mathbb{R}^d)^2 \ni (x, x') \mapsto (\alpha x, \alpha x') \in (\mathbb{R}^d)^2.$$

Pick $\gamma \in \Gamma(\mu, \nu)$, we have

$$|\alpha|^p \int_{(\mathbb{R}^d)^2} |x - y|^p d\gamma(x, y)$$

$$= \int_{(\mathbb{R}^d)^2} |\alpha x - \alpha y|^p d\gamma$$

$$= \int_{\mathbb{R}^2} |x - y|^p d(\alpha \times \alpha)_{\#}\gamma(x, y)$$

Take the infimum for both sides over $\Gamma(\mu, \nu)$, combine it with the fact that Equation 37 is a bijection. We obtain Equation 36. $\qquad\square$

Now we introduce the proof of Proposition A.13.

*Proof.* Suppose $\theta^g \sim \mathcal{N}(0, I_d)$ and let $\theta = \frac{\theta^g}{\|\theta^g\|}$, we have $\theta \sim \mathcal{U}(\mathbb{S}^{d-1})$ by Proposition A.9. Then we have:

$$SW_p^p(\mu, \nu; \mathcal{N}(0, I_d))$$

$$= \mathbb{E}_{\theta^g \sim \mathcal{N}(0, I_d)}[W_p^p(\theta_{\#}^g \mu, \theta_{\#}^g \nu)]$$

$$= \mathbb{E}_{\theta^g \sim \mathcal{N}(0, I_d)}[\|\theta^g\|^p W_p^p(\theta_{\#}\mu, \theta_{\#}\nu)] \qquad \text{by Lemma A.15}$$

$$= \mathbb{E}_{\theta^g \sim \mathcal{N}(0, I_d)}[\|\theta^g\|^p] \cdot \mathbb{E}_{\theta \sim \mathcal{U}(\mathbb{S}^{d-1})}[W_p^p(\theta_{\#}\mu, \theta_{\#}\nu)] \qquad \text{by Proposition A.8 (4)}$$

$$= 2^{p/2} \frac{\Gamma(\frac{p+d}{2})}{\Gamma(d/2)} \cdot SW_p^p(\mu, \nu; \mathcal{U}(\mathbb{S}^{d-1})) \qquad \text{by Proposition A.8 (3).}$$

$$\square$$

## A.6 Proof of Proposition A.4

We adapt notations $V_k, U$ in previous subsection.

**Lemma A.16.** *Suppose $\mu^d, \nu^d$ satisfy Assumption 4.1, pick $\theta^d \in \mathbb{R}^d$ and let $\hat{\theta}^k = U^\top \theta^d$ then we have:*

$$\theta_\# \mu^d = \hat{\theta}^k_\# \mu^k, \theta_\# \nu^d = \hat{\theta}^k_\# \nu^k.$$

*Proof.* For each $x \in \mathrm{Span}(U) = V_k$, we have

$$
\begin{aligned}
\theta^\top x &= P_U(\theta)^\top x + (\theta - P_U(\theta))^\top x \\
&= P_U(\theta)^\top x + 0 && \text{Since } \theta - P_U(\theta) \in V_k^\perp \\
&= (UU^\top \theta)^\top x \\
&= (U^\top \theta)^\top (U^\top x)
\end{aligned}
$$

Thus,

$$\theta^d_\# \mu^d \;=\; (U^\top \theta^d)_\# \mu^k \;=\; \hat{\theta}^k_\# \mu^k$$

Similarly, we have $\theta^d_\# \nu^d = \hat{\theta}^k_\# \nu^k$ and we complete the proof. $\qquad\square$

**Lemma A.17.** *Suppose $\theta^d_1, \ldots, \theta^d_L \overset{\text{i.i.d.}}{\sim} \mathcal{U}(\mathbb{S}^{d-1})$ and let $\theta^k_l = \frac{U^\top \theta^d_l}{\|U^\top \theta^d_l\|}$ for all $l \in [1{:}L]$. Then $\theta^k_1, \ldots, \theta^k_L \overset{\text{i.i.d.}}{\sim} \mathcal{U}(\mathbb{S}^{k-1})$.*

*Proof.* First, since $k < d$, we have

$$\mathcal{U}(\theta^d \in \mathbb{S}^{d-1} : U^\top \theta^d = 0_k) = 0.$$

Thus, with probability 1, each $\theta^k_l$ is well-defined.

By Proposition A.9, with probability 1, we can realize $\theta^d_1, \ldots, \theta^d_L$ as follows:

Suppose $X_1, \ldots, X_L \overset{\text{i.i.d.}}{\sim} \mathcal{N}(0, I_d)$ and $\theta^d_l = \frac{X_l}{\|X_l\|}$.

Then

$$\theta^k_l = \frac{U^\top \theta^d_l}{\|U^\top \theta^d_l\|} = \frac{U^\top X_l / \|X_l\|}{\|U^\top X_l / \|X_l\|\|} = \frac{U^\top X_l}{\|U^\top X_l\|}.$$

Since $U^\top X_l \sim \mathcal{N}(0, I_k)$, we have $\theta^k_l \sim \mathcal{U}(\mathbb{S}^{k-1})$.

Furthermore, since $X_1, \ldots, X_L$ are independent, $\theta^k_1, \ldots, \theta^k_L$ are independent. Thus, $\theta^k_1, \ldots, \theta^k_L \overset{\text{i.i.d.}}{\sim} \mathcal{U}(\mathbb{S}^{k-1})$.
$\qquad\square$

Now we discuss the proof of Proposition 4.5.

*Proof of Proposition 4.5.* Pick $\theta^d \in \mathbb{S}^{d-1}$.

We have

$$
\begin{aligned}
W_p^p(\theta^d_\# \mu^d, \theta^d_\# \nu^d) &= W_p^p((U^\top \theta^d)_\# \mu^k, (U^\top \theta^d)_\# \nu^k) && \text{By Lemma A.16} \\
&= \|U^\top \theta^d\|^p W_p^p(\theta^k_\# \mu^k, \theta^k_\# \nu^k) && \text{By Lemma A.15}
\end{aligned}
$$

Thus we prove Equation 13.

Now, we pick $\theta_1^d, \ldots, \theta_L^d \in \mathbb{S}^{d-1}$, and thus we have:

$$
\begin{aligned}
&SW_p^p(\mu^k, \nu^k; \frac{1}{L}\sum_{l=1}^L \delta_{\theta_l^k}) \\
&= \frac{1}{L}\sum_{l=1}^L W_p^p((\theta_l^k)_\#\mu^k, (\theta_l^k)_\#\nu^k) \\
&= \frac{1}{L}\sum_{l=1}^L \frac{1}{\|U^\top\theta_l^d\|^p} W_p^p((U^\top\theta_l^d)_\#\mu^k, (U^\top\theta_l^d)_\#\nu^k) && \text{By convention } 0\cdot\frac{1}{0}=0 \\
&= \frac{1}{L}\sum_{l=1}^L \frac{1}{\|U^\top\theta_l^d\|^p} W_p^p((\theta_l^d)_\#\mu^d, (\theta_l^d)_\#\nu^d) && \text{By Equation 13} \\
&= \widetilde{SW}_p^p\left(\mu^d, \nu^d; \frac{1}{L}\sum_{l=1}^L \delta_{\theta_l^d}, \rho\right)
\end{aligned}
$$

And we prove Equation 14.

Similarly, we obtain the last equation,

$$
\begin{aligned}
\widetilde{SW}_p^p(\mu^d, \nu^d; \mathcal{U}(\mathbb{S}^{d-1}), \rho) &= \mathbb{E}_{\theta^d\sim\mathcal{U}(\mathbb{S}^{d-1})}\left[\frac{1}{\|U^\top\theta^d\|^p}W_p^p((\theta^d)_\#\mu^d, (\theta^d)_\#\nu^d)\right] \\
&= \mathbb{E}_{\theta^d\sim\mathcal{U}(\mathbb{S}^{d-1})}\left[W_p^p((\theta^k)_\#\mu^k, (\theta^k)_\#\nu^k)\right] && \text{By Equation 13} \\
&= \mathbb{E}_{\theta^k\sim\mathcal{U}(\mathbb{S}^{k-1})}[W_p^p((\theta^k)_\#\mu^k, (\theta^k)_\#\nu^k)] && \text{By Lemma A.17} \\
&= SW_p^p(\mu^k, \nu^k)
\end{aligned}
$$

$\square$

### A.6.1 Proof of Theorem 4.6

In this section, we first discuss the relation between $SW_p^p(\mu^d, \nu^d; \mathcal{N}(0, I_d))$ and $SW_p^p(\mu^k, \nu^k; \mathcal{N}(0, I_k))$ under Assumption 4.1. Next, we present the proof of Theorem 4.6.

Based on the above lemma, we can derive the following relation between $SW_p^p(\mu^d, \nu^d; \mathcal{N}(0, I_d))$ and $SW_p^p(\mu^k, \nu^k; \mathcal{N}(0, I_k))$.

**Lemma A.18.** *Under Assumption 4.1, we have*

$$SW_p^p(\mu^d, \nu^d; \mathcal{N}(0, I_d)) = SW_p^p(\mu^k, \nu^k; \mathcal{N}(0, I_k)) \tag{38}$$

*Proof.* Suppose $\theta^d \sim \mathcal{N}(0, I_d)$ and let $\theta^k = U^\top\theta^d$. Then by Proposition A.8 (1), we have $\theta^k \sim \mathcal{N}(0, U^\top I_d U) = \mathcal{N}(0, I_k)$. Therefore,

$$
\begin{aligned}
&SW_p^p(\mu^d, \nu^d; \mathcal{N}(0, I_d)) \\
&= \mathbb{E}_{\theta^d\sim\mathcal{N}(0,I_d)}[W_p^p(\theta_\#^d\mu^d, \theta_\#^d\nu^d)] \\
&= \mathbb{E}_{\theta^d\sim\mathcal{N}(0,I_d)}[W_p^p(\theta_\#^k\mu^k, \theta_\#^k\nu^k)] && \text{By Lemma A.16, where } \theta^k = U^\top\theta^d \\
&= \mathbb{E}_{\theta^k\sim\mathcal{N}(0,I_k)}[W_p^p(\theta_\#^k\mu^k, \theta_\#^k\nu^k)] \\
&= SW_p^p(\mu^k, \nu^k; \mathcal{N}(0, I_k))
\end{aligned}
$$

and we complete the proof. $\square$

Combining the above lemma and Proposition A.13, we can prove Theorem 4.6.

*Proof of Theorem 4.6.* For the first equality, we have

$$
\begin{aligned}
&SW_p^p(\mu^d, \nu^d; \mathcal{U}(\mathbb{S}^{d-1})) \\
&= \frac{1}{C_d} SW_p^p(\mu^d, \nu^d; \mathcal{N}(0_d, I_d)) && \text{By Proposition A.13} && (39) \\
&= \frac{1}{C_d} SW_p^p(\mu^k, \nu^k; \mathcal{N}(0_k, I_k)) && \text{By Lemma A.18} \\
&= \frac{C_k}{C_d} SW_p^p(\mu^k, \nu^k; \mathcal{U}(\mathbb{S}^{k-1})) && \text{By Proposition A.13} && (40)
\end{aligned}
$$

where $C_d = 2^{p/2} \frac{\Gamma\left(\frac{d}{2} + \frac{p}{2}\right)}{\Gamma\left(\frac{d}{2}\right)}$ and $C_k$ is defined similarly.

$\square$

## A.7    Proof of Proposition 4.7

We first introduce the following lemma:

**Lemma A.19.** *Let $I_{d\times k}$ denote the matrix $\begin{bmatrix} I_{k\times k} \\ 0_{(d-k)\times k} \end{bmatrix}$, and suppose $\theta^d \sim \mathcal{U}(\mathbb{S}^{d-1})$, then $\|U^\top \theta^d\|$ and $\|I_{d\times k}^\top \theta^d\|$ have the same distribution.*

*Proof.* Since $U$ has orthonormal columns, there exist orthogonal matrices $V_1 \in \mathbb{R}^{d\times d}$ and $V_2 \in \mathbb{R}^{k\times k}$ such that $U = V_1 I_{d\times k} V_2$.

Then we have

$$
\|U^\top \theta^d\| = \|V_2^\top I_{d\times k}^\top V_1^\top \theta^d\| = \|I_{d\times k}^\top V_1^\top \theta^d\|
$$

Since $\theta^d \sim \mathcal{U}(\mathbb{S}^{d-1})$, then $V_1^\top \theta^d \sim \mathcal{U}(\mathbb{S}^{d-1})$.

Thus, $I_{d\times k}^\top \theta^d, I_{d\times k}^\top V_1^\top \theta^d$ have the same distribution. Thus $\|I_{d\times k}^\top \theta^d\|, \|I_{d\times k}^\top V_1^\top \theta^d\| = \|U^\top \theta^d\|$ have the same distribution. $\square$

Based on this, we can prove the statement (1) in Proposition 4.7.

*Proof of Proposition 4.7 (1).* By the above lemma, it is sufficient to consider $U = I_{d\times k}$.

Let $\theta^{d,g} \sim \mathcal{N}(0, I_d)$, and let $\theta^{d,g}[i], i \in [1:d]$ denote each component of $\theta^{d,g}$. Thus $\theta^{d,g}[1], \dots \theta^{d,g}[d] \overset{i.i.d}{\sim} \mathcal{N}(0,1)$. We can redefine $\theta^d$ as $\theta^d = \frac{\theta^{d,g}}{\|\theta^{d,g}\|}$, thus,

$$
\begin{aligned}
\|U^\top \theta^d\|^2 &= \frac{\|U^\top \theta^{d,g}\|^2}{\|\theta^{d,g}\|^2} \\
&= \frac{\sum_{i=1}^{k} \theta^{d,g}[i]^2}{\sum_{i=1}^{d} \theta^{d,g}[i]^2} \sim \text{Beta}(\frac{k}{2}, \frac{d-k}{2})
\end{aligned}
$$

Thus, we have

$$
\mathbb{E}[\|U^\top \theta^d\|^p] = \mathbb{E}[(\|U^\top \theta^d\|^2)^{p/2}] = \frac{\Gamma(k/2 + p/2)\Gamma(d/2)}{\Gamma(k/2)\Gamma(d/2 + p/2)} = \frac{C_k}{C_d}.
$$

Note, $\|U^\top \theta_1^d\|, \dots \|U^\top \theta_L^d\|$ are i.i.d. random variables, thus, we have

$$
\mathbb{E}[\widehat{ESSF(L)}] = \frac{1}{L}\mathbb{E}[\sum_{l=1}^{L} \|U^\top \theta_l^d\|^p] = \frac{C_k}{C_d}.
$$

Similarly,

$$\mathrm{Var}[\widehat{ESSF}(L)] = \frac{1}{L}\mathrm{Var}[\|U^\top \theta_l^d\|^p]$$

where $\mathrm{Var}[\|U^\top \theta_l^d\|^p] > 0$, is the variance of the $p/2-$th power of a $\mathrm{Beta}(k/2, (d-k)/2)$ variable, which is a constant only depends on $(d, k, p)$.

$\square$

**Proposition A.20.** *For fixed $d$ and $p > 0$, the function $k \mapsto \frac{C_k}{C_d}$ is monotone increasing in $k$.*

*Proof.* Since $C_d$ is constant with respect to $k$, it is enough to show that

$$g(x) := \frac{\Gamma\left(\frac{x}{2} + \frac{p}{2}\right)}{\Gamma\left(\frac{x}{2}\right)}$$

is increasing for $x > 0$. Differentiating $\log g(x)$ gives

$$\frac{d}{dx}\log g(x) = \frac{1}{2}\psi\left(\frac{x}{2} + \frac{p}{2}\right) - \frac{1}{2}\psi\left(\frac{x}{2}\right),$$

where $\psi$ is the digamma function. Since $\psi$ is increasing on $(0, \infty)$, the right-hand side is nonnegative. Hence $g$ is increasing, and so is $k \mapsto \frac{C_k}{C_d}$. $\square$

*Proof of Proposition 4.7(2)–(3).* Write $Z_l := \|U^\top \theta_l^d\|^p \in [0, 1]$ and $\bar{Z} := \frac{1}{L}\sum_{l=1}^L Z_l = \widehat{ESSF}(L)$. From Proposition 4.5 and the discrete gradient formula, each per-sample gradient satisfies

$$\nabla_{x_i} W_p^p(\theta_{\#}\hat{\mu}^d, \theta_{\#}\hat{\nu}^d) = Z\,\nabla_{x_i}W_p^p(\theta_{\#}^k\hat{\mu}_k, \theta_{\#}^k\hat{\nu}_k),$$

with $\|\nabla_{x_i}W_p^p(\theta_{\#}^k\hat{\mu}_k, \theta_{\#}^k\hat{\nu}_k)\| \le K$ for some $K < \infty$ depending only on the (compact) supports. Hence

$$\|\epsilon_L(x_i)\| \le \frac{pq_i^1}{L}\sum_{l=1}^L |Z_l - \bar{Z}| \left\|\nabla_{x_i}W_p^p(\theta_{l\#}^k\hat{\mu}_k, \theta_{l\#}^k\hat{\nu}_k)\right\| \le pq_i^1 K \frac{1}{L}\sum_{l=1}^L |Z_l - \bar{Z}|.$$

Using $|Z_l - \bar{Z}| \le |Z_l - \mathbb{E}Z| + |\bar{Z} - \mathbb{E}Z|$ and averaging, we obtain

$$\|\epsilon_L(x_i)\| \le 2pq_i^1 K\,|\bar{Z} - \mathbb{E}Z|.$$

Since $Z_l \in [0, 1]$ are i.i.d., Hoeffding's inequality gives for any $\delta > 0$,

$$\mathbb{P}\left(|\bar{Z} - \mathbb{E}Z| \ge \delta\right) \le 2\exp\left(-2\delta^2 L\right).$$

This yields (2) by the strong law of large numbers (a.s. convergence), and (3) by setting $\delta \mapsto \delta/(2pq_i^1 K)$ above.

$\square$

### A.7.1 Proof of Proposition 4.7

### A.8 Special case: Learning rate bound for the SWD Gradient Flow problem

In this section, we consider the following sliced gradient flow problem Bonet et al. (2021a):

$$\mu_{t+1} \leftarrow \arg\min_{\mu \in \mathcal{P}_2(\mathbb{R}^k)} \frac{1}{2\tau} SW_2^2(\mu, \mu_t) + F(\mu)$$

$$s.t.\mu_0 = \mu^k$$

$$\text{where } F(\mu) := SW_2^2(\mu, \nu^k), \text{for some } \nu^k, \tau > 0$$

In the discrete setting, $\mu^k = \sum_{i=1}^n q_i^1 \delta_{x_i}, \nu^k = \sum_{j=1}^m q_j^2 \delta_{y_j}$. Furthermore, we assume that the pmf of $\mu_t$ is fixed. Then the above problem can be transferred to the following:

$$X^{t+1} \leftarrow X^t - h_t \odot \nabla_X SW_2^2(\mu_t, \nu^k), \text{ where } \mu_t = \sum_{i=1}^n q_i^1 \delta_{x_i^t}, X^t = [x_1, \ldots, x_n] \tag{41}$$

where $\odot$ denote the element-wise product operator, and $h_t \in \mathbb{R}_+^n$.

We will discuss how to select the appropriate learning rate $h_t$.

**Gradient and Hessian of Sliced Wasserstein distance.** First, we discuss the gradient and Hessian matrix of the function $X \mapsto SW_2^2(\mu, \nu^k)$:

Pick $\theta \in \mathbb{S}^{d-1}$ and suppose that $\gamma_\theta$ is an optimal transportation plan for $W_2^2(\theta_{\#}\mu, \theta_{\#}\nu^k)$.

Then by Bonneel & Coeurjolly (2019), we have:

$$\nabla_{x_i} W_2^2(\theta_{\#}\mu, \theta_{\#}\nu^k) = 2\theta\theta^\top (q_i^1 x_i - \sum_{j=1}^m y_j \gamma_{i,j}^\theta), \forall x_i$$

Note, when $W_2^2(\theta_{\#}\mu, \theta_{\#}\nu^k)$ is induced by a Monge mapping, the above formulation can be simplified to $q_i^1 \theta\theta^\top (x_i - T(x_i))$.

Thus the Hessian matrix is

$$\left[\frac{\partial^2 W_2^2(\theta_{\#}\mu, \theta_{\#}\nu^k)}{\partial x_i[l]\partial x_i[l']}\right]_{l,l' \in [1:d]} = 2q_i^1 \theta\theta^\top.$$

Therefore, the gradient for mapping $X \mapsto SW_2^2(\mu, \nu^k)$ with respect to each $x_i$ is given by:

$$g(x_i) := \nabla_{x_i} SW_2^2(\mu, \nu^k) = 2 \int_{\mathbb{S}^{d-1}} \theta\theta^\top (q_i^1 x_i - \sum_{i=1}^m y_j \gamma_{i,j}^\theta) d\mathcal{U}(\mathbb{S}^{d-1})(\theta)$$

$$\approx \frac{2}{N} \sum_{l=1}^L \theta_l \theta_l^\top (q_i^1 x_i - \sum_{j=1}^m y_j \gamma_{i,j}^{\theta_l})$$

where the second line is the Monte carlo approximation.

Similarly, the Hession matrix and the Monte carlo approximation are given by

$$H(x_i) := H_{x_i}(SW_2^2(\mu, \nu^k)) = 2q_i^1 \int \theta\theta^\top d\mathcal{U}(\mathbb{S}^{d-1})(\theta) = 2q_i^1 \frac{1}{k} I_k$$

$$\approx \frac{2q_i^1}{N} \sum_{i=1}^N \theta\theta^\top$$

By classical machine learning theory, the optimal learning rate for $x_i$, is given by

$$(h_t)_i = \frac{g(x_i)^\top g(x_i)}{g(x_i)^\top H g(x_i)} = \frac{k}{2q_i^1}, \forall i \in [1:n] \tag{42}$$

**Remark A.21.** *We consider a simplified case to intuitively understand the above learning rate. Suppose* $\mu^k = q_i^1 \delta_{x_i}$ *and* $\nu^k = q_i^1 \delta_{y_j}$ *(relaxing the assumption that* $\mu^k$ *and* $\nu^k$ *are probability measures). Then, we have:*

$$
\begin{aligned}
&SW_2^2(\mu^d, \nu^d) \\
&= q_i^1 \mathbb{E}_{\theta \sim \mathcal{U}(\mathbb{S}^{d-1})}[(\theta^\top x_i - \theta^\top y_j)^2] \\
&= q_i^1 \mathbb{E}_{\theta \sim \mathcal{U}(\mathbb{S}^{d-1})}[\|\theta\theta^\top x_i - \theta\theta^\top y_j\|^2] \\
&= q_i^1 (x_i - y_j)^\top \mathbb{E}[\theta\theta^\top](x_i - y_j) \\
&= \frac{1}{k} q_i^1 \|x_i - y_j\|_2^2 \\
&= \frac{1}{k} W_2^2(\mu, \nu).
\end{aligned}
$$

*Thus the gradient with respect to* $x_i$ *becomes*

$$
g(x_i) = 2\frac{q_i^1}{k}(x_i - y_j).
$$

*Letting* $t = 0$, *we plug the learning rate from Equation 42 and the gradient into Equation 41, obtaining:*

$$
x_i^{t+1} \leftarrow x_i^t - (y_j - x_i) = y_j.
$$

*Intuitively, the learning rate* $(h_t)_i$ *for* $x_i$ *is chosen such that the (negative) gradient becomes the displacement given by the classical OT transportation plan, i.e.,*

$$
-g(x_i) \approx y_j - x_i.
$$

*That is, when* $\theta$ *is sufficiently large (i.e.,* $\frac{1}{L}\sum_{i=1}^{L} \theta\theta^\top \approx \frac{1}{k}I_k$*),* $\mu_t^k$ *will converge to* $\nu^k$ *in one step.*

## A.9 Proof of Proposition 4.9

Pick $x_i$ from $\{x_1, \ldots, x_n\}$. Note, based on Assumption 4.1, $x_i = UU^\top x_i = Ux_i^k, \forall i \in [1:n]$. Thus, we have

$$
\begin{aligned}
&\nabla_{x_i} SW_p^p(\hat{\mu}, \hat{\nu}; \frac{1}{L}\sum_{l=1}^{L}\delta_{\theta_l}) \\
&= U\nabla_{x_i^k} SW_p^p(\hat{\mu}, \hat{\nu}; \frac{1}{L}\sum_{l=1}^{L}\delta_{\theta_l}) \\
&= U\nabla_{x_i^k} \sum_{l=1}^{L}\frac{1}{L}\sum_{i,j}(q_i^1\|U^\top\theta_l\|(\theta_l^k)^\top(x_i^k - \frac{1}{q_i^1}y_j^k\gamma_{i,j}^{\theta_l}))^p \\
&= q_i^1 p U \frac{1}{L}\sum_{l=1}^{L}\|U^\top\theta_l\|(\theta_l^k)^\top(x_i^k - \frac{1}{q_i^1}y_j^k\gamma_{i,j}^{\theta_l})^{p-1}.
\end{aligned}
$$

Similarly,

$$
\begin{aligned}
&\nabla_{x_i} SW_p^p(\hat{\mu}^k, \hat{\nu}^k; \frac{1}{L}\sum_{l=1}^{L}\delta_{\theta_l^k}) \\
&= q_i^1 p U \frac{1}{L}\sum_{l=1}^{L}(\theta_l^k)^\top(x_i^k - \frac{1}{q_i^1}y_j^k\gamma_{i,j}^{\theta_l})^{p-1}
\end{aligned}
\tag{43}
$$

where $\gamma^{\theta_l}$ is the optimal transportation plan for 1D problem $W_p^p((\theta_l)_{\#}\hat{\mu}, (\theta_l)_{\#}\hat{\nu}) = W_p^p((\theta_l^k)_{\#}\hat{\mu}^k, (\theta_l^k)_{\#}\hat{\nu}^k)$.

Thus,

$$
\epsilon_L(x_i) = \nabla_{x_i} SW_p^p(\hat{\mu}^d, \hat{\nu}^d; \sum_{l=1}^L \delta_{\theta_l^d}) - \widehat{ESSF}(L) \cdot \nabla_{x_i} SW_p^p(\hat{\mu}_k, \hat{\nu}_k; \sum_{l=1}^L \delta_{\theta_l^k})
$$

$$
= \frac{pq_i^1}{L} U \sum_{l=1}^L (\|U^\top \theta_l\|^p - \frac{1}{L}\sum_{l'=1}^L \|U^\top \theta_{l'}\|^p) \underbrace{\theta_l^k ((\theta_l^k)^\top (x_i^k - \frac{1}{q_i^1}\sum_{j=1}^m y_j^k \gamma_{i,j}^{\theta_l}))^{p-1}}_{A(\theta_l^k)}.
$$

where $A(\theta_l^k)$ is a vector function from $\mathbb{S}^{k-1}$ to $\mathbb{R}^k$. By Cauchy–Schwarz inequality, and the fact $\|\theta_l^k\| = 1$, we have

$$
\|A(\theta_l^k)\| \le \max_{x_i, y_j} \|x_i^k - y_j^k\|^{p-1} = \max_{x_i, y_j} \|x_i - y_j\|^{p-1}
$$

Then we have:

$$
\|\epsilon_L(x_i)\| = pq_i^1 \left| \underbrace{\sum_{l=1}^L \frac{1}{L}\left(\|U^\top \theta_l\|^p - \frac{1}{L}\sum_{l'=1}^L \|U^\top \theta_{l'}\|^p\right)}_{B_L} \right| \|U A(\theta_l^k)\|
$$

$$
= pq_i^1 \|A(\theta_l^k)\| |B_L|
$$

$$
\le pq_i^1 K |B_L|.
$$

By law of large number, with probability 1, $B_L \to 0$, thus $\|(\epsilon_L)\| \to 0$, that is $\epsilon_L \to 0_d$.

It remains to bound the convergence rate of $\|\epsilon_L\|$.

By Hoeffding inequality and the fact $\|U^\top \theta_l\|^2 \in [0, 1]$, we have

$$
\mathbb{P}(|B_L| \ge \epsilon) \le 2e^{-\epsilon^2 L}.
$$

Replacing $\epsilon$ by $\epsilon/(pq_i^1 K)$, we obtain:

$$
\mathbb{P}(\|\epsilon_L(x_i)\| \le \epsilon) \ge 1 - 2e^{-\epsilon^2 L/(pq_i^1 K)^2}.
$$

and we complete the proof.

### A.10 Discussion when Assumption 4.1 is not satisfied

In this section, we briefly discuss the context when Assumption 4.1 is not satisfied. In particular, we aim to show the following:

**Proposition A.22.** *Let $U, V_k$ be defined in Assumption 4.1, choose $\mu^d, \nu^d \in \mathcal{P}_p(\mathbb{R}^d)$, and let $\mu^k, \nu^k$ be defined by $(U^\top)_{\#}\mu^d, (U^\top)_{\#}\nu^d$, we claim the following:*

$$
W_2^2(\mu^k, \nu^k) \le W_2^2(\mu^d, \nu^d) \le W_2^2(\mu^k, \nu^k) + 2(m_2(U_{\#}^\perp \mu^d) + m_2(U_{\#}^\perp \nu^d)) \tag{44}
$$

*where $m_2(U_{\#}^\perp \mu^d)$ denotes the second moment of the measure $U_{\#}^\perp \mu^d$.*

*Proof.* For each pair $(x, y) \in (\mathbb{R}^d)^2$, we have

$$
\begin{aligned}
&\|P_U(x) - P_U(y)\|^2 \\
&\le \|x - y\|^2 &&\text{By definition of projection} &&(45)\\
&= \|P_U(x) - P_U(y)\|^2 + \|P_{U^\perp}(x) - P_{U^\perp}(y)\|^2 &&\text{Pythagorean theorem}\\
&\le \|P_U(x) - P_U(y)\|^2 + 2\|(U^\perp)^\top x\|^2 + 2\|(U^\perp)^\top y\|^2 &&&&(46)
\end{aligned}
$$

From Equation 45, we have

$$W_2^2((P_U)_\# \mu^d, (P_U)_\# \nu^d) \leq W_2^2(\mu^d, \nu^d).$$

Combining it with Proposition A.4, we have:

$$W_2^2(\mu^k, \nu^k) = W_2^2((P_U)_\# \mu^d, (P_U)_\# \nu^d) \leq W_2^2(\mu^d, \nu^d),$$

and we prove the first inequality in Equation 44.

Similarly, let $\gamma \in \Gamma(\mu, \nu)$ be the optimal transportation plan for $W_2^2((P_U)_\# \mu^d, (P_U)_\# \nu^d)$. From Equation 46, we have:

$$
\begin{aligned}
& W_2^2(\mu^d, \nu^d) \\
& \leq \mathbb{E}_{(X,Y) \sim \gamma}[\|X - Y\|^2] \\
& \leq \mathbb{E}_{(X,Y) \sim \gamma}[\|P_U(X) - P_U(Y)\|^2 + 2\|(U^\perp)^\top X\|^2 + 2\|(U^\perp)^\top Y\|^2] \qquad \text{By Equation 46} \\
& = W_2^2((P_U)_\# \mu, (P_U)_\# \nu) + 2(m_2((U^\perp)_\# \mu^d) + m_2((U^\perp)_\# \nu^d))
\end{aligned}
$$

Thus, we prove the second inequality of Equation 44. $\qquad \square$

**Proposition A.23.** *Based on the same notations of Proposition A.22, we have:*

$$\frac{k}{d} SW_2^2(\mu^k, \nu^k) \leq SW_2^2(\mu^d, \nu^d) \leq \frac{k}{d} SW_2^2(\mu^k, \nu^k) + 2\frac{d-k}{d}(m_2(U_\#^\perp \mu^d) + m_2(U_\#^\perp \nu^d)) \qquad (47)$$

*Proof.* Pick $\theta \in \mathbb{S}^{d-1}$ and $x, y \in \mathbb{R}^d$. We have:

$$
\begin{aligned}
& \|\theta^\top P_U(x) - \theta^\top P_U(y)\|^2 \\
& \leq \|\theta^\top x - \theta^\top y\|^2 \qquad\qquad\qquad\qquad\qquad\qquad\qquad\qquad\qquad\qquad\qquad (48) \\
& = \|P_U(\theta)^\top P_U(x) - P_U(\theta)^\top P_U(y)\|^2 + \|P_{U^\perp}(\theta)^\top P_{U^\perp}(x) - P_{U^\perp}(\theta)^\top P_{U^\perp}(y)\|^2 \\
& \leq \|\theta^\top P_U(x) - \theta^\top P_U(y)\|^2 + \|(U^\perp)^\top \theta\|^2 \|(U^\perp)^\top x - (U^\perp)^\top y\|^2 \qquad \text{Cauchy–Schwarz inequality} \\
& \leq \|\theta^\top P_U(x) - \theta^\top P_U(y)\|^2 + 2\|(U^\perp)^\top \theta\|^2 (\|(U^\perp)^\top x\|^2 + \|(U^\perp)^\top y\|^2) \qquad\qquad\qquad (49)
\end{aligned}
$$

Choose $\theta \in \mathbb{S}^{d-1}$. From Proposition A.22, we have

$$W_2^2(\theta_\# (P_U)_\# \mu^d, \theta_\# (P_U)_\# \nu^d) \leq W_2^2(\theta_\# \mu^d, \theta_\# \nu^d)$$

Take expected value with respect to $\theta$, we have

$$SW_2^2((P_U)_\# \mu^d, (P_U)_\# \nu^d) \leq SW_2^2(\mu^d, \nu^d)$$

Combining this with Theorem 4.6, we prove the first inequality in Equation 47.

Similarly, from Equation 49, we have

$$W_2^2(\theta_\# \mu^d, \theta_\# \nu^d) \leq W_2^2(\theta_\# (P_U)_\# \mu^d, \theta_\# (P_U)_\# \nu^d) + 2\|(U^\perp)^\top \theta\|^2 (m_2(U_\#^\perp \mu^d) + m_2(U_\#^\perp \nu^d)).$$

Take expected value with respect to $\theta$, we obtain:

$$
\begin{aligned}
& SW_2^2(\mu^d, \nu^d) \\
& \leq SW_2^2((P_U)_\# \mu^d, (P_U)_\# \nu^d) + 2\mathbb{E}_\theta[\|U^\perp \theta\|^2](m_2(U_\#^\perp \mu^d) + m_2(U_\#^\perp \nu^d)) \\
& = \frac{k}{d} SW_2^2(\mu^k, \nu^k) + 2\frac{d-k}{d}(m_2(U_\#^\perp \mu^d) + m_2(U_\#^\perp \nu^d))
\end{aligned}
$$

there the last equality holds from Theorem 4.6 and the fact $\|U^\perp \theta\|^2 \sim \text{Beta}(\frac{d-k}{2}, \frac{k}{2})$.

$\qquad \square$

### A.10.1 A compact numerical illustration away from the exact subspace regime

To complement the bounds above, Table 1 varies the orthogonal residual energy in a synthetic Gaussian example with fixed $(d, k) = (50, 2)$. We add isotropic noise in $V_k^\perp$ with standard deviation $\sigma$ and a matched orthogonal mean shift to one measure, then compare the ambient-space $SW_2^2(\mu^d, \nu^d)$ against the lower and upper bounds in Proposition A.22. As the residual energy grows, the exact $(k/d)$ scaling no longer holds, and the ambient-space value moves away from the exact-subspace prediction in a controlled way.

| $\sigma$ | Residual energy | Empirical $SW_2^2(\mu^d, \nu^d)$ | Lower bound | Upper bound |
|---|---|---|---|---|
| 0.00 | 0.000 | 0.040 | 0.040 | 0.040 |
| 0.05 | 0.249 | 0.045 | 0.044 | 0.523 |
| 0.10 | 1.000 | 0.050 | 0.047 | 1.968 |
| 0.20 | 3.997 | 0.046 | 0.044 | 7.719 |
| 0.40 | 16.061 | 0.067 | 0.052 | 30.889 |
| 0.80 | 64.131 | 0.122 | 0.045 | 123.176 |

**Table 1:** Approximate-subspace illustration for Proposition A.22. The exact $(k/d)$ prediction is recovered at zero residual energy, and the ambient-space $SW_2^2$ moves away from it as the orthogonal residual increases.

# B    Additional Details for the Numerical Experiments

## B.1    Gradient Flow

### B.1.1    Background Overview

Let $\mathcal{P}(\mathbb{R}^d)$ denote the space of probability measures on $\mathbb{R}^d$. For $\mu, \nu \in \mathcal{P}(\mathbb{R}^d)$, the gradient flow of the SWD distance in the space of probability measures evolves according to the continuity equation

$$\frac{\partial \mu_t}{\partial t} + \nabla \cdot (v_t \mu_t) = 0, \tag{50}$$

where $\mu_t$ is a time-dependent probability measure and $v_t$ the velocity field $v_t = -\nabla \frac{\delta SW_2^2(\mu_t, \nu)}{\delta \mu_t}$. This describes the transport of measure $\mu_t$ in the Wasserstein space $\mathcal{P}_2(\mathbb{R}^d)$, commonly referred to as **Wasserstein Gradient Flows** (WGF, Ambrosio et al. (2008))

For numerical simulation in practice, one discretizes this dynamic using a particle approximation. We let $\{x_t^i\}_{i=1}^N$ denote a system of $N$ particles evolving according to the following system of ODEs:

$$\frac{dx_i^t}{dt} = -\nabla_{x_i} SW_2^2(\mu_t^N, \nu), \tag{51}$$

where $\mu_t^N = \frac{1}{L}\sum_{i=1}^L \delta_{x_i^t}$ is the empirical measure based on the particle positions $x_i^t$.

These WGF particle-based approaches preserve key features of continuous systems and have been widely adopted, especially machine learning applications (Peyré et al. (2019)).

### B.1.2    Experiments

**On classic synthetic datasets**

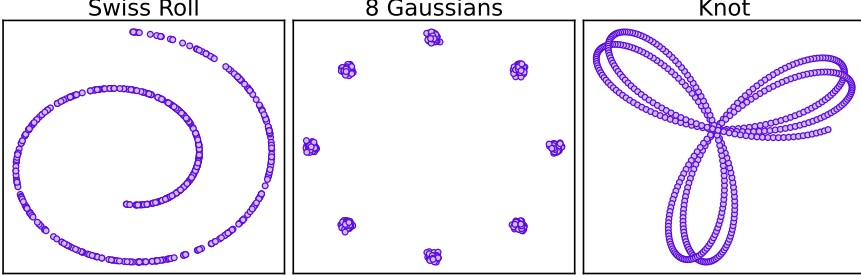

**Figure 7:** Classic synthetic 2D datasets (shown) embedded in spaces of different target dimensions.

**Table 2:** Quantitative comparison of the best final converged $W_2(\downarrow)$ and runtime ($\downarrow$) between different variants for Gradient Flow with (embedded) classic synthetic datasets.

| Met. | Swiss | | | 8 Gauss. | | | Knot | | | RT(s)↓ |
|---|---|---|---|---|---|---|---|---|---|---|
| | $d=2$ | $d=50$ | $d=100$ | $d=2$ | $d=50$ | $d=100$ | $d=2$ | $d=50$ | $d=100$ | |
| SWD | $0.0001^{\pm 0.0000}$ | $0.0004^{\pm 0.0000}$ | $0.0004^{\pm 0.0000}$ | $0.0002^{\pm 0.0000}$ | $0.0002^{\pm 0.0001}$ | $0.0006^{\pm 0.0001}$ | $0.0002^{\pm 0.0000}$ | $0.0004^{\pm 0.0000}$ | $0.0004^{\pm 0.0000}$ | $8.62^{\pm 0.04}$ |
| MaxSW | $0.0000^{\pm 0.0000}$ | $0.0219^{\pm 0.0051}$ | $0.0342^{\pm 0.0022}$ | $0.0005^{\pm 0.0000}$ | $0.0171^{\pm 0.0004}$ | $0.0385^{\pm 0.0006}$ | $0.0005^{\pm 0.0000}$ | $0.0246^{\pm 0.0009}$ | $0.0303^{\pm 0.0009}$ | $74.02^{\pm 1.61}$ |
| DSW | $0.0002^{\pm 0.0001}$ | $0.0004^{\pm 0.0000}$ | $0.0004^{\pm 0.0000}$ | $0.0002^{\pm 0.0001}$ | $0.0004^{\pm 0.0001}$ | $0.0006^{\pm 0.0001}$ | $0.0003^{\pm 0.0000}$ | $0.0004^{\pm 0.0001}$ | $0.0004^{\pm 0.0000}$ | $162.25^{\pm 0.20}$ |
| MaxKSW | $0.0002^{\pm 0.0000}$ | $0.0124^{\pm 0.0082}$ | $0.0122^{\pm 0.0010}$ | $0.0002^{\pm 0.0000}$ | $0.0154^{\pm 0.0001}$ | $0.0216^{\pm 0.0007}$ | $0.0002^{\pm 0.0000}$ | $0.0165^{\pm 0.0048}$ | $0.0171^{\pm 0.0048}$ | $125.23^{\pm 0.54}$ |
| iMSW | $0.0001^{\pm 0.0000}$ | $0.0021^{\pm 0.0001}$ | $0.0050^{\pm 0.0001}$ | $0.0001^{\pm 0.0000}$ | $0.0021^{\pm 0.0001}$ | $0.0059^{\pm 0.0001}$ | $0.0002^{\pm 0.0000}$ | $0.0034^{\pm 0.0001}$ | $0.0054^{\pm 0.0001}$ | $74.45^{\pm 0.03}$ |
| viMSW | $0.0002^{\pm 0.0001}$ | $0.0003^{\pm 0.0000}$ | $0.0005^{\pm 0.0000}$ | $0.0003^{\pm 0.0001}$ | $0.0003^{\pm 0.0001}$ | $0.0008^{\pm 0.0000}$ | $0.0003^{\pm 0.0001}$ | $0.0005^{\pm 0.0000}$ | $0.0005^{\pm 0.0000}$ | $255.76^{\pm 0.28}$ |
| oMSW | $0.0001^{\pm 0.0000}$ | $0.0002^{\pm 0.0000}$ | $0.0005^{\pm 0.0000}$ | $0.0002^{\pm 0.0001}$ | $0.0002^{\pm 0.0000}$ | $0.0006^{\pm 0.0000}$ | $0.0002^{\pm 0.0001}$ | $0.0004^{\pm 0.0000}$ | $0.0004^{\pm 0.0000}$ | $16.55^{\pm 0.01}$ |
| rMSW | $0.0002^{\pm 0.0000}$ | $0.0003^{\pm 0.0000}$ | $0.0005^{\pm 0.0000}$ | $0.0003^{\pm 0.0001}$ | $0.0003^{\pm 0.0000}$ | $0.0008^{\pm 0.0000}$ | $0.0003^{\pm 0.0001}$ | $0.0006^{\pm 0.0000}$ | $0.0005^{\pm 0.0000}$ | $179.70^{\pm 1.08}$ |
| EBSW | $0.0002^{\pm 0.0001}$ | $0.0002^{\pm 0.0000}$ | $0.0005^{\pm 0.0000}$ | $0.0001^{\pm 0.0000}$ | $0.0002^{\pm 0.0000}$ | $0.0006^{\pm 0.0000}$ | $0.0003^{\pm 0.0001}$ | $0.0004^{\pm 0.0000}$ | $0.0002^{\pm 0.0000}$ | $9.66^{\pm 1.15}$ |
| RPSW | $0.0001^{\pm 0.0000}$ | $0.0001^{\pm 0.0000}$ | $0.0004^{\pm 0.0000}$ | $0.0002^{\pm 0.0000}$ | $0.0001^{\pm 0.0000}$ | $0.0010^{\pm 0.0000}$ | $0.0002^{\pm 0.0000}$ | $0.0001^{\pm 0.0000}$ | $0.0004^{\pm 0.0000}$ | $19.47^{\pm 0.03}$ |
| EBRPSW | $0.0007^{\pm 0.0002}$ | $0.0002^{\pm 0.0001}$ | $0.0003^{\pm 0.0000}$ | $0.0002^{\pm 0.0001}$ | $0.0002^{\pm 0.0001}$ | $0.0006^{\pm 0.0000}$ | $0.0002^{\pm 0.0001}$ | $0.0004^{\pm 0.0000}$ | $0.0002^{\pm 0.0000}$ | $20.30^{\pm 0.05}$ |

**On the MNIST and CelebA images**

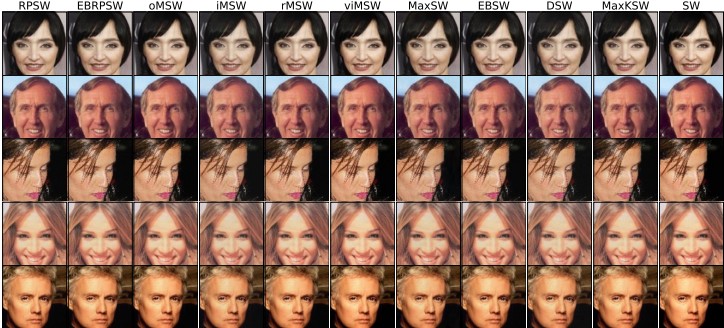

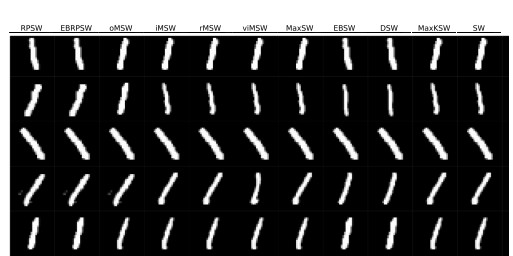

**Figure 8:** Gradient Flow visualization for images from the MNIST dataset (left) and the CelebA dataset (right).

| Method | MNIST (s) ↓ | CelebA (s)↓ |
|---|---|---|
| DSW | $12500.00^{\pm0.00}$ | $126054.85^{\pm745.89}$ |
| EBSW | $686.18^{\pm45.31}$ | $6694.50^{\pm148.08}$ |
| RPSW | $800.36^{\pm5.97}$ | $6038.05^{\pm86.32}$ |
| EBRPSW | $699.33^{\pm9.70}$ | $3171.20^{\pm582.31}$ |
| oMSW | $482.29^{\pm8.46}$ | $3808.34^{\pm475.66}$ |
| iMSW | $1359.97^{\pm10.19}$ | $3601.99^{\pm11.01}$ |
| rMSW | $1115.98^{\pm150.49}$ | $100358.26^{\pm1002.95}$ |
| viMSW | $4161.11^{\pm16.05}$ | $96007.74^{\pm937.50}$ |
| MaxSW | $7231.97^{\pm70.28}$ | $9780.51^{\pm485.71}$ |
| MaxKSW | $6891.43^{\pm35.52}$ | $65560.25^{\pm332.10}$ |
| SWD | $441.41^{\pm36.85}$ | $3335.51^{\pm76.52}$ |

**Table 3:** Runtime comparison for all methods in the MNIST/CelebA setups

## B.2  Color Transfer

**Table 4:** Quantitative comparison of the best final converged $W_2$ ↓ and runtime ↓ between different variants for Color Transfer.

| Method | Best $W_2$ ↓ (LR) | | | Runtime(s) ↓ |
|---|---|---|---|---|
| | Set 1 | Set 2 | Set 3 | |
| SWD | $0.01^{\pm0.00}$ (1e-1) | $0.01^{\pm0.00}$ (8e-1) | $0.00^{\pm0.00}$ (1e0) | $8.62^{\pm0.04}$ |
| MaxSW | $0.03^{\pm0.00}$ (1e-3) | $0.03^{\pm0.00}$ (3e-4) | $0.03^{\pm0.00}$ (3e-4) | $74.02^{\pm1.61}$ |
| DSW | $0.03^{\pm0.00}$ (1e-3) | $0.03^{\pm0.00}$ (1e-3) | $0.03^{\pm0.00}$ (8e-4) | $162.25^{\pm0.20}$ |
| MaxKSW | $0.03^{\pm0.00}$ (5e-4) | $0.03^{\pm0.00}$ (5e-4) | $0.03^{\pm0.00}$ (5e-4) | $125.23^{\pm0.54}$ |
| iMSW | $0.03^{\pm0.00}$ (1e-3) | $0.03^{\pm0.00}$ (1e-4) | $0.03^{\pm0.00}$ (1e-3) | $74.45^{\pm0.03}$ |
| viMSW | $0.03^{\pm0.00}$ (1e-3) | $0.03^{\pm0.00}$ (1e-4) | $0.03^{\pm0.00}$ (1e-3) | $255.76^{\pm0.28}$ |
| oMSW | $0.03^{\pm0.00}$ (1e-3) | $0.03^{\pm0.00}$ (1e-3) | $0.03^{\pm0.00}$ (1e-3) | $16.55^{\pm0.01}$ |
| rMSW | $0.03^{\pm0.00}$ (1e-3) | $0.03^{\pm0.00}$ (1e-3) | $0.03^{\pm0.00}$ (1e-3) | $179.70^{\pm1.08}$ |
| EBSW | $0.03^{\pm0.00}$ (1e-3) | $0.03^{\pm0.00}$ (1e-3) | $0.03^{\pm0.00}$ (1e-3) | $9.66^{\pm1.15}$ |
| RPSW | $0.10^{\pm0.00}$ (3e-3) | $0.10^{\pm0.00}$ (1e-2) | $0.10^{\pm0.09}$ (1e-3) | $19.47^{\pm0.03}$ |
| EBRPSW | $0.03^{\pm0.00}$ (8e-4) | $0.03^{\pm0.00}$ (5e-4) | $0.03^{\pm0.00}$ (5e-4) | $20.30^{\pm0.05}$ |

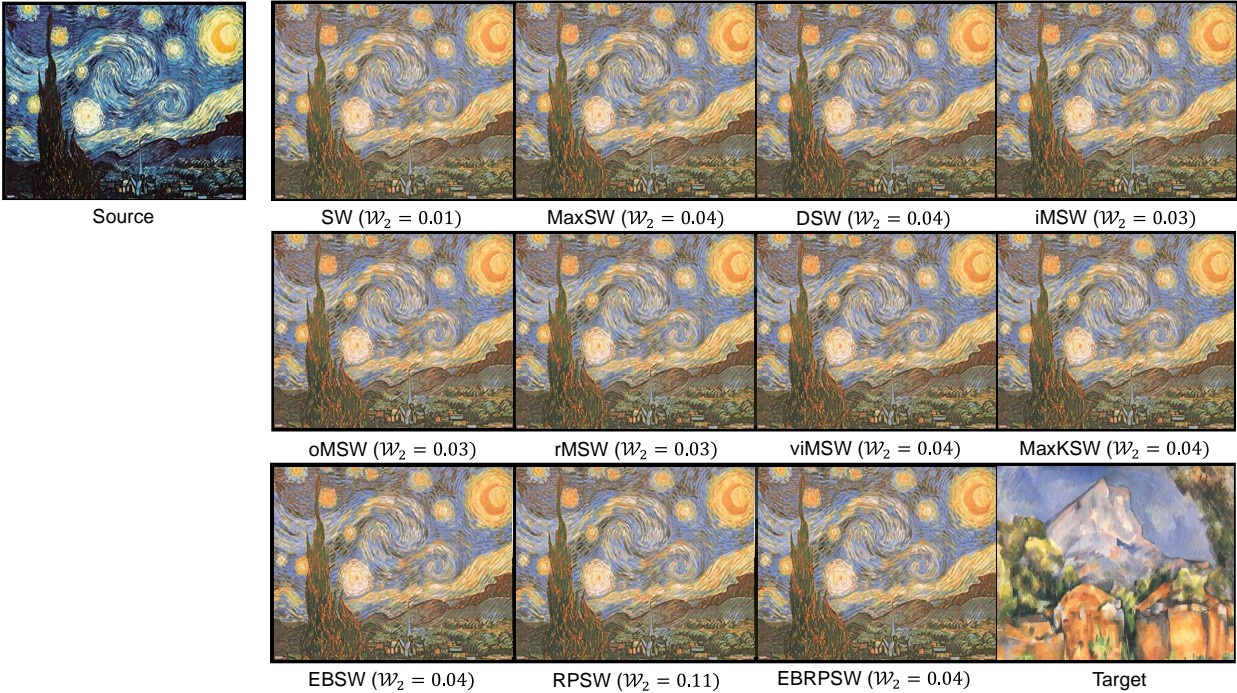

**Figure 9:** Additional color transfer visualization

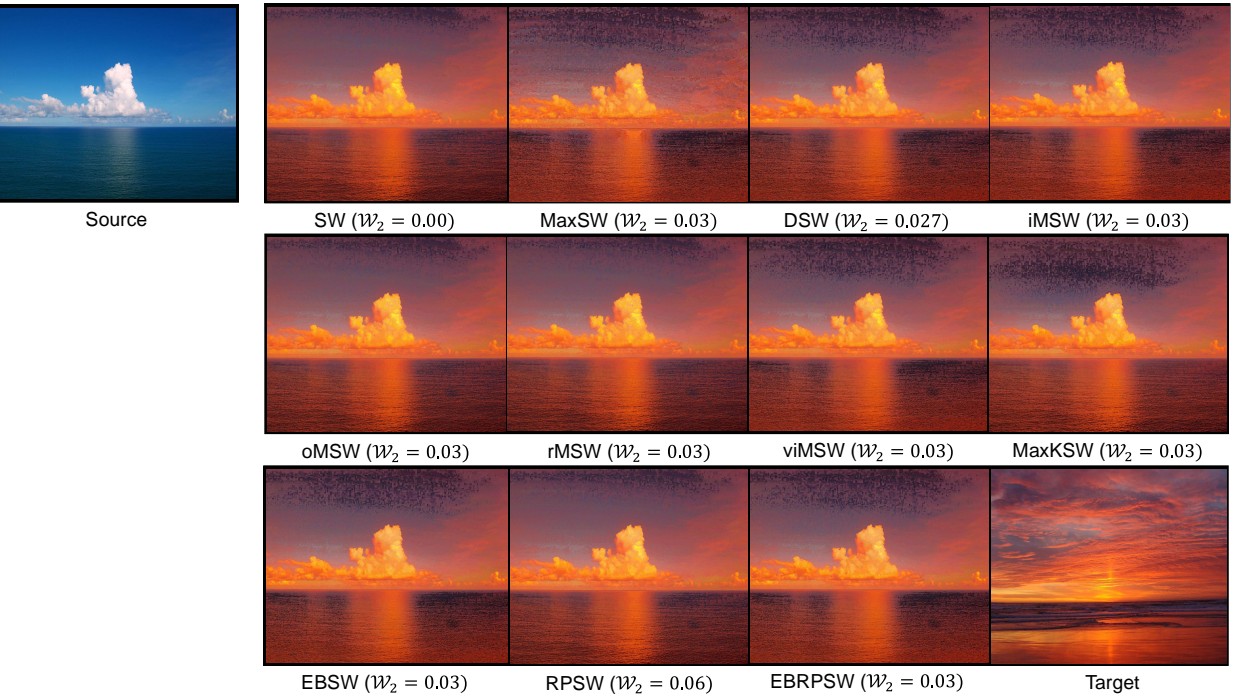

**Figure 10:** Additional color transfer visualization

### B.3 Deep Generative Modeling

In this section, we provide additional details on the model architecture and numerical results for the deep generative modeling tasks. We present both qualitative and quantitative results, including the $W_2$ metric, to evaluate the model's performance across different SWD variants. The experiments were conducted using the FFHQ dataset.

**Model architecture:**

For unconditional generation:

$$G(z) = \text{FC}_d \circ \text{LeakyReLU}_{0.2} \circ \text{BN} \circ \text{FC}_{1024}$$
$$\circ \text{LeakyReLU}_{0.2} \circ \text{BN} \circ \text{FC}_{512}$$
$$\circ \text{LeakyReLU}_{0.2} \circ \text{BN} \circ \text{FC}_{256}(z), \quad z \in \mathbb{R}^8, G(z) \in \mathbb{R}^{512}$$

For unpaired translation:

$$T(z) = z + \text{FC}_d \circ \text{LeakyReLU}_{0.2} \circ \text{BN} \circ \text{FC}_{1024}$$
$$\circ \text{LeakyReLU}_{0.2} \circ \text{BN} \circ \text{FC}_{1024}(z), \quad z \in \mathbb{R}^{512}$$

$$(52)$$

**Table 5:** Quantitative comparison between different variants for Deep Generative Modeling 5.4.

| Method | Unconditional Gen. | | | Unpaired Translation | | | |
|---|---|---|---|---|---|---|---|
| | $W_2 \downarrow$ | (LR) | RT(s) $\downarrow$ | M2F: $W_2 \downarrow$ (LR) | RT(s)$\downarrow$ | A2C: $W_2 \downarrow$ (LR) | RT(s)$\downarrow$ |
| SWD | $14.67^{\pm.01}$ (1e0) | | $26.69^{\pm.23}$ | $14.15^{\pm.02}$ (5e-1) | $25.47^{\pm.09}$ | $14.58^{\pm.03}$ (1e0) | $27.94^{\pm.14}$ |
| MaxSW | $13.38^{\pm.17}$ (1e-2) | | $95.83^{\pm.24}$ | $14.01^{\pm.02}$ (5e-3) | $102.68^{\pm.03}$ | $14.52^{\pm.02}$ (3e-3) | $103.29^{\pm2.98}$ |
| DSW | $14.35^{\pm.06}$ (8e-3) | | $197.70^{\pm.14}$ | $14.11^{\pm.02}$ (5e-3) | $198.42^{\pm.38}$ | $14.60^{\pm.04}$ (5e-3) | $198.08^{\pm.32}$ |
| MaxKSW | $15.22^{\pm.03}$ (1e-2) | | $53.38^{\pm3.16}$ | $14.20^{\pm.01}$ (5e-2) | $45.90^{\pm.04}$ | $14.65^{\pm.02}$ (5e-2) | $45.73^{\pm.08}$ |
| iMSW | $14.27^{\pm.01}$ (1e-3) | | $90.27^{\pm.23}$ | $14.06^{\pm.01}$ (3e-3) | $92.70^{\pm.12}$ | $14.59^{\pm.01}$ (1e-3) | $92.65^{\pm.17}$ |
| viMSW | $15.16^{\pm.03}$ (1e-3) | | $49.26^{\pm.05}$ | $14.09^{\pm.01}$ (3e-3) | $271.10^{\pm.42}$ | $14.57^{\pm.01}$ (3e-3) | $271.09^{\pm.84}$ |
| oMSW | $14.81^{\pm.04}$ (5e-2) | | $29.34^{\pm.18}$ | $14.12^{\pm.01}$ (8e-2) | $31.39^{\pm.05}$ | $14.58^{\pm.03}$ (1e-2) | $31.23^{\pm.04}$ |
| rMSW | $14.80^{\pm.07}$ (5e-2) | | $193.77^{\pm.63}$ | $14.16^{\pm.01}$ (8e-2) | $195.07^{\pm.12}$ | $14.60^{\pm.02}$ (8e-3) | $195.81^{\pm.27}$ |
| EBSW | $16.68^{\pm.19}$ (3e-3) | | $22.16^{\pm.07}$ | $14.71^{\pm.02}$ (1e-2) | $26.67^{\pm.00}$ | $15.09^{\pm.04}$ (8e-4) | $26.68^{\pm.02}$ |
| RPSW | $14.46^{\pm.06}$ (3e-2) | | $33.35^{\pm.07}$ | $14.14^{\pm.00}$ (1e-1) | $37.11^{\pm.10}$ | $14.60^{\pm.01}$ (1e-2) | $36.99^{\pm.12}$ |
| EBRPSW | $16.90^{\pm.22}$ (3e-3) | | $34.40^{\pm.07}$ | $14.69^{\pm.02}$ (1e-2) | $38.15^{\pm.05}$ | $15.10^{\pm.05}$ (1e-3) | $38.14^{\pm.11}$ |

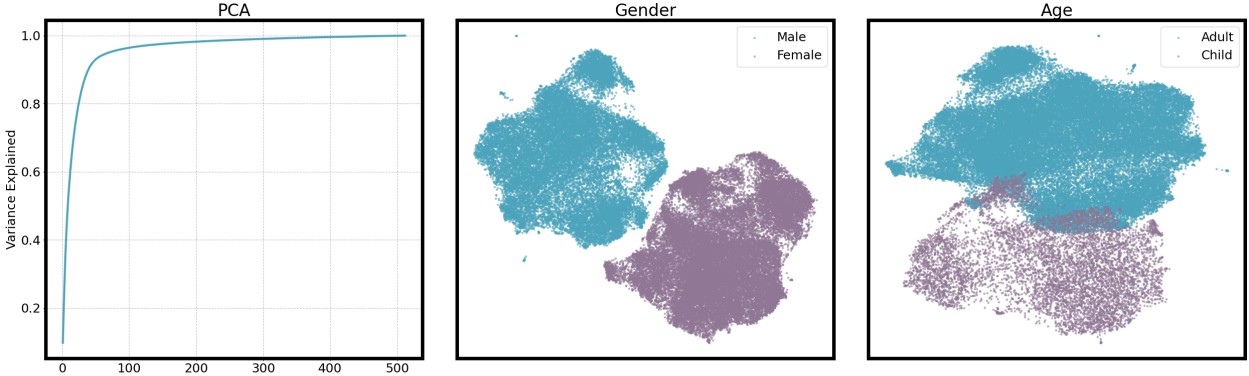

**(a) Left:** Cumulative Explaning Variance plot for the FFHQ latents. **Middle/Right:** UMAP visualization of the Gender and Age splits.

| FFHQ Subset | Train size | Test size |
|---|---|---|
| Adults ($\geq 18$) | 48786 | 8104 |
| Children ($< 10$) | 8345 | 1405 |
| Male | 26732 | 4351 |
| Female | 32816 | 5572 |

**(b)** Subset size.

**Figure 11:** FFHQ dataset (Karras et al. (2019))

## B.4 Detailed numerical results

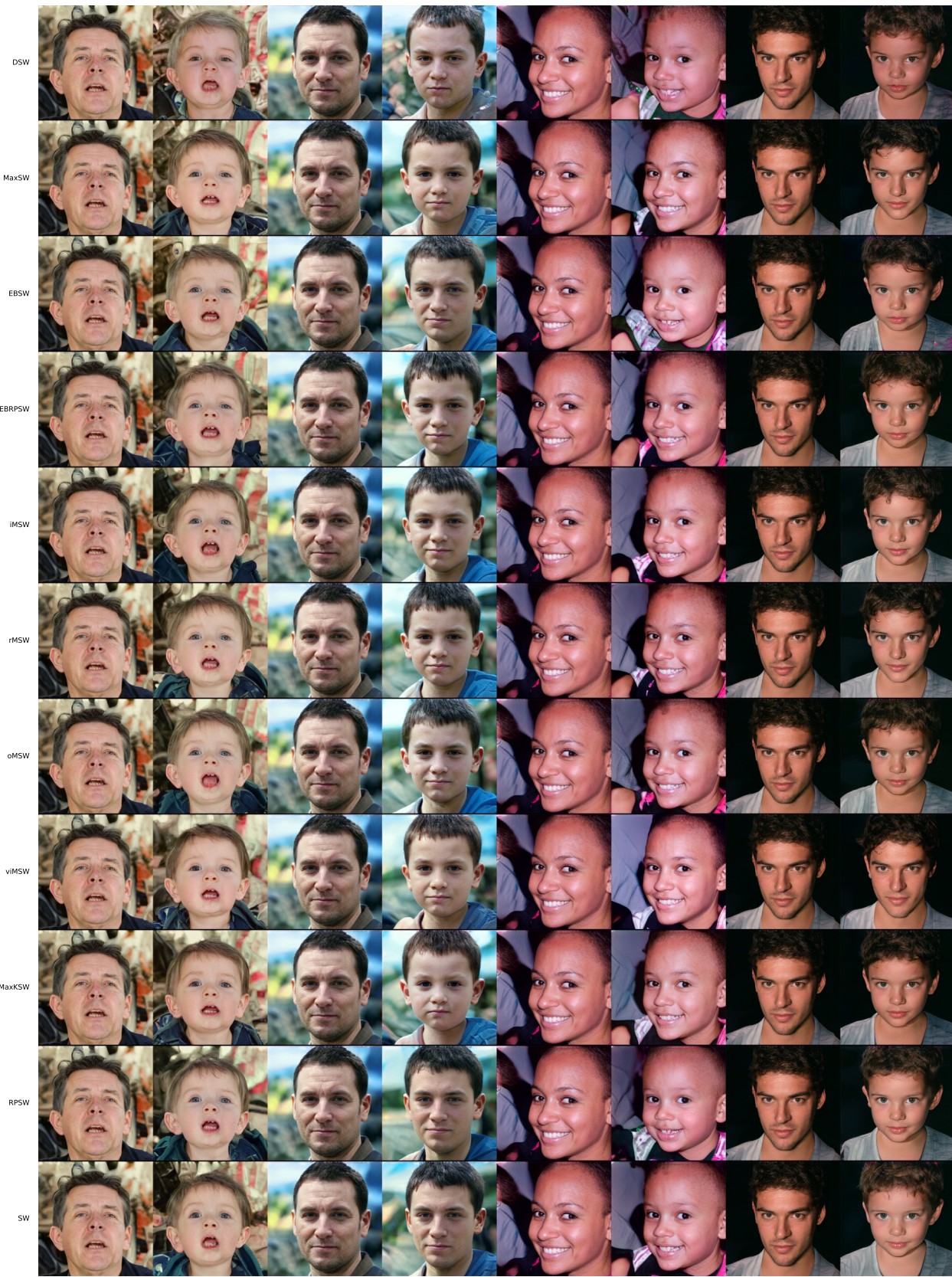

**Figure 12:** Visualizations for the A2C translation task (using the model with the lowest $W_2$ for each method).

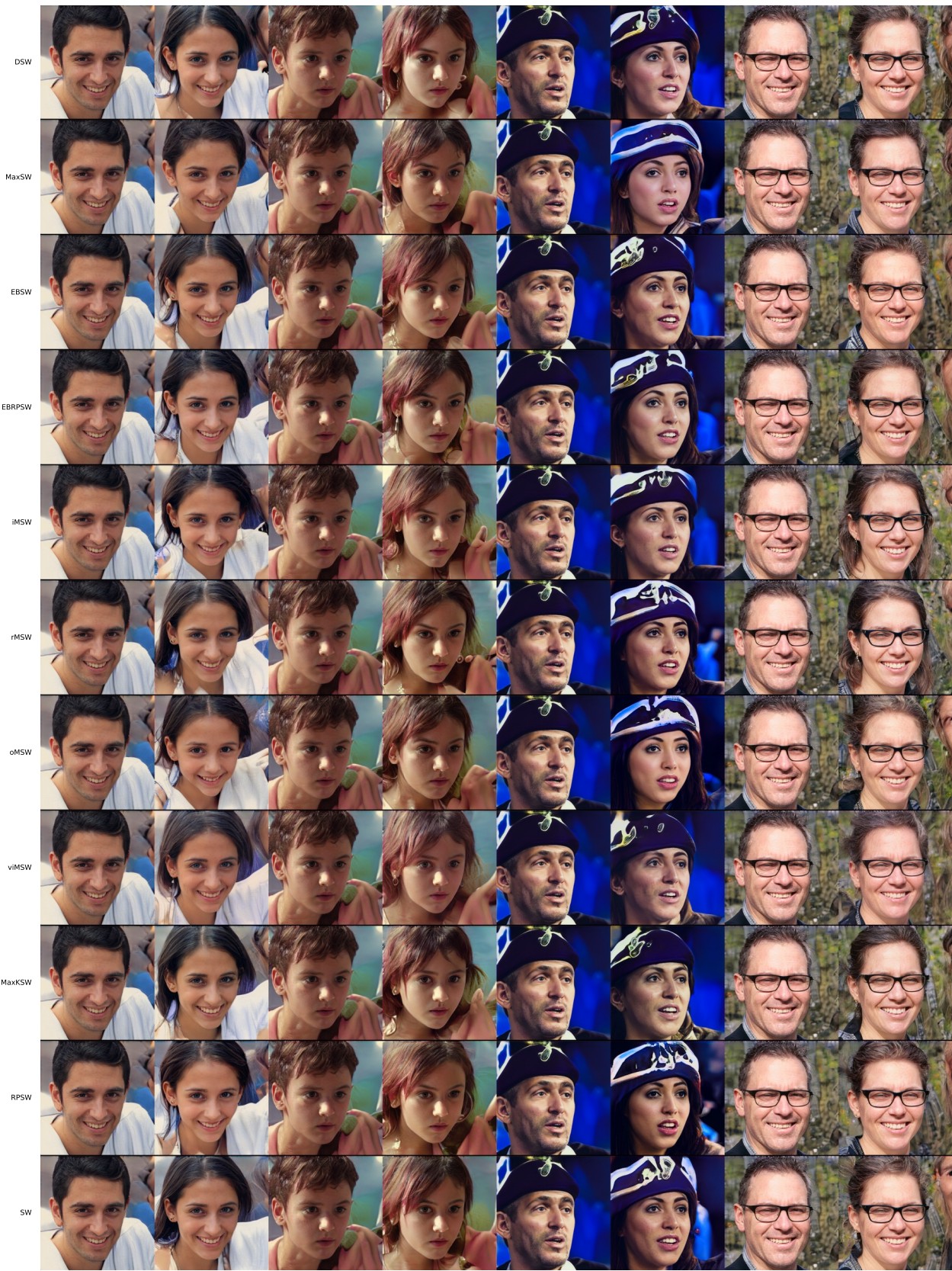

**Figure 13:** Visualizations for the M2F translation task (using the model with the lowest $W_2$ for each method).

| LR | SW | MaxSW | DSW | MaxKSW | iMSW | viMSW | oMSW | rMSW | EBSW | RPSW | EBRPSW |
|---|---|---|---|---|---|---|---|---|---|---|---|
| 100 | 2.3681±0.3508 | 0.5442±0.0195 | 1.8232±1.2533 | 1.2128±0.0685 | 3.8247±0.6643 | 3.1542±0.3530 | 1.2212±0.0565 | 1.8336±0.2363 | 2.9758±0.0588 | 2.7564±0.1211 | 3.0755±0.1628 |
| 80 | 1.7578±0.4630 | 0.5171±0.0098 | 0.9759±0.5058 | 1.3729±0.0088 | 2.3046±0.5586 | 1.8896±0.9034 | 1.3554±0.1909 | 1.8872±0.4225 | 2.4539±0.0358 | 2.1423±0.0344 | 2.2756±0.0813 |
| 50 | 0.8872±0.3438 | 1.8463±0.9464 | 0.9317±0.5872 | 1.6624±0.0214 | 0.6639±0.1253 | 1.2846±0.0949 | 1.4466±0.1613 | 1.4180±0.1092 | 1.4591±0.0569 | 1.1420±0.0967 | 1.4347±0.1352 |
| 30 | 0.6367±0.2107 | 0.9107±0.3422 | 0.5040±0.2023 | 0.9698±0.0011 | 0.7317±0.2280 | 1.2114±0.0310 | 0.7519±0.2114 | 0.9445±0.2815 | 1.1241±0.3024 | 0.7251±0.0681 | 0.9207±0.2912 |
| 10 | 0.2152±0.0336 | 0.4321±0.0154 | 0.2632±0.1356 | 0.1576±0.0045 | 0.2419±0.0488 | 0.1563±0.0275 | 0.1995±0.0240 | 0.2076±0.0414 | 0.3243±0.0648 | 0.2212±0.0339 | 0.3707±0.0406 |
| 8 | 0.2267±0.0385 | 0.3502±0.0581 | 0.1917±0.0742 | 0.2063±0.0025 | 0.2307±0.0693 | 0.2052±0.0742 | 0.1746±0.0061 | 0.1845±0.0750 | 0.1792±0.1090 | 0.1613±0.0202 | 0.3136±0.0649 |
| 5 | 0.1355±0.0168 | 0.2353±0.0056 | 0.1060±0.0565 | 0.1113±0.0150 | 0.1258±0.0381 | 0.0990±0.0052 | 0.1097±0.0109 | 0.1307±0.0414 | 0.1225±0.0701 | 0.1417±0.0255 | 0.2239±0.0023 |
| 3 | 0.0612±0.0111 | 0.1480±0.0087 | 0.0575±0.0198 | 0.0793±0.0143 | 0.0774±0.0036 | 0.0644±0.0219 | 0.0599±0.0023 | 0.1042±0.0094 | 0.1042±0.0027 | 0.0392±0.0146 | 0.1115±0.0030 |
| 1 | 0.0286±0.0079 | 0.0715±0.0020 | 0.0221±0.0119 | 0.0473±0.0016 | 0.0224±0.0038 | 0.0268±0.0093 | 0.0208±0.0004 | 0.0245±0.0041 | 0.0210±0.0021 | 0.0176±0.0076 | 0.0216±0.0022 |
| $8 \times 10^{-1}$ | 0.0182±0.0067 | 0.0631±0.0046 | 0.0165±0.0049 | 0.0429±0.0012 | 0.0197±0.0031 | 0.0267±0.0039 | 0.0181±0.0007 | 0.0205±0.0026 | 0.0178±0.0048 | 0.0233±0.0054 | 0.0151±0.0021 |
| $5 \times 10^{-1}$ | 0.0144±0.0058 | 0.0492±0.0012 | 0.0100±0.0046 | 0.0353±0.0012 | 0.0123±0.0008 | 0.0128±0.0066 | 0.0118±0.0009 | 0.0121±0.0051 | 0.0109±0.0001 | 0.0117±0.0049 | 0.0091±0.0014 |
| $3 \times 10^{-1}$ | 0.0063±0.0025 | 0.0365±0.0014 | 0.0040±0.0026 | 0.0238±0.0014 | 0.0079±0.0014 | 0.0078±0.0012 | 0.0063±0.0003 | 0.0062±0.0020 | 0.0051±0.0010 | 0.0083±0.0029 | 0.0052±0.0019 |
| $1 \times 10^{-1}$ | 0.0014±0.0005 | 0.0099±0.0001 | 0.0023±0.0003 | 0.0041±0.0001 | 0.0019±0.0001 | 0.0023±0.0008 | 0.0019±0.0002 | 0.0025±0.0010 | 0.0019±0.0008 | 0.0015±0.0007 | 0.0025±0.0005 |
| $8 \times 10^{-2}$ | 0.0019±0.0005 | 0.0070±0.0005 | 0.0016±0.0008 | 0.0022±0.0000 | 0.0017±0.0001 | 0.0029±0.0003 | 0.0016±0.0002 | 0.0017±0.0008 | 0.0023±0.0002 | 0.0020±0.0003 | 0.0012±0.0001 |
| $5 \times 10^{-2}$ | 0.0013±0.0005 | 0.0036±0.0006 | 0.0016±0.0001 | 0.0013±0.0002 | 0.0015±0.0002 | 0.0011±0.0001 | 0.0009±0.0002 | 0.0014±0.0008 | 0.0012±0.0002 | 0.0013±0.0003 | 0.0012±0.0002 |
| $3 \times 10^{-2}$ | 0.0005±0.0001 | 0.0014±0.0001 | 0.0007±0.0004 | 0.0006±0.0002 | 0.0006±0.0002 | 0.0007±0.0004 | 0.0007±0.0000 | 0.0006±0.0001 | 0.0004±0.0003 | 0.0009±0.0001 | 0.0007±0.0002 |
| $1 \times 10^{-2}$ | 0.0002±0.0000 | 0.0005±0.0000 | 0.0003±0.0000 | 0.0002±0.0000 | 0.0002±0.0001 | 0.0003±0.0001 | 0.0002±0.0001 | 0.0003±0.0001 | 0.0003±0.0001 | 0.0002±0.0000 | 0.0002±0.0001 |
| $8 \times 10^{-3}$ | 0.0002±0.0001 | 0.0004±0.0000 | 0.0002±0.0001 | 0.0002±0.0000 | 0.0002±0.0000 | 0.0002±0.0000 | 0.0001±0.0001 | 0.0003±0.0000 | 0.0002±0.0000 | 0.0002±0.0001 | 0.0002±0.0001 |
| $5 \times 10^{-3}$ | 0.0046±0.0063 | 0.0145±0.0083 | 0.0001±0.0000 | 0.0056±0.0045 | 0.0176±0.0004 | 0.0001±0.0000 | 0.0108±0.0023 | 0.0001±0.0000 | 0.0001±0.0000 | 0.0139±0.0109 | 0.0040±0.0045 |
| $3 \times 10^{-3}$ | 0.0744±0.0031 | 0.0547±0.0049 | 0.0584±0.0035 | 0.0675±0.0070 | 0.0652±0.0027 | 0.0578±0.0032 | 0.0693±0.0050 | 0.0571±0.0098 | 0.0490±0.0053 | 0.0720±0.0118 | 0.0430±0.0022 |
| $1 \times 10^{-3}$ | 0.4724±0.0245 | 0.4962±0.0489 | 0.3954±0.0311 | 0.4584±0.0498 | 0.4294±0.0182 | 0.3809±0.0095 | 0.4875±0.0300 | 0.4454±0.0413 | 0.4598±0.0035 | 0.4550±0.0257 | 0.4569±0.0212 |
| $8 \times 10^{-4}$ | 0.5596±0.0618 | 0.5197±0.0412 | 0.5045±0.0217 | 0.5379±0.0168 | 0.5398±0.0372 | 0.5325±0.0290 | 0.5704±0.0030 | 0.4655±0.0322 | 0.5102±0.0100 | 0.5269±0.0288 | 0.5842±0.0629 |
| $5 \times 10^{-4}$ | 0.6497±0.0230 | 0.6422±0.0430 | 0.5670±0.0304 | 0.6304±0.0142 | 0.6935±0.0272 | 0.5945±0.0148 | 0.6689±0.0122 | 0.6006±0.0077 | 0.6022±0.0140 | 0.6490±0.0217 | 0.6022±0.0103 |
| $3 \times 10^{-4}$ | 0.7475±0.0404 | 0.7277±0.0425 | 0.7061±0.0107 | 0.7534±0.0072 | 0.7213±0.0118 | 0.6642±0.0273 | 0.7026±0.0120 | 0.6609±0.0067 | 0.7100±0.0204 | 0.7137±0.0168 | 0.7379±0.0714 |
| $1 \times 10^{-4}$ | 0.8183±0.0169 | 0.8544±0.0166 | 0.7573±0.0481 | 0.7824±0.0234 | 0.7953±0.0529 | 0.7755±0.0233 | 0.8008±0.0592 | 0.7258±0.0020 | 0.8298±0.0434 | 0.8379±0.0426 | 0.8268±0.0378 |
| $8 \times 10^{-5}$ | 0.8061±0.0603 | 0.8473±0.0326 | 0.7661±0.0200 | 0.8084±0.0445 | 0.8328±0.0587 | 0.7158±0.0079 | 0.8242±0.0773 | 0.7232±0.0083 | 0.8122±0.0442 | 0.8063±0.0226 | 0.7910±0.0183 |
| $5 \times 10^{-5}$ | 0.8857±0.0193 | 0.7754±0.0065 | 0.7568±0.0355 | 0.8341±0.0581 | 0.7777±0.0060 | 0.7548±0.0466 | 0.7539±0.0168 | 0.7539±0.0457 | 0.8688±0.0086 | 0.8262±0.0200 | 0.8144±0.0535 |
| $3 \times 10^{-5}$ | 0.8197±0.0106 | 0.8329±0.0499 | 0.7885±0.0524 | 0.8745±0.0079 | 0.8467±0.0079 | 0.7556±0.0234 | 0.8440±0.0403 | 0.7488±0.0444 | 0.8240±0.0296 | 0.8314±0.0190 | 0.8560±0.0302 |
| $1 \times 10^{-5}$ | 0.8319±0.0335 | 0.8280±0.0646 | 0.7332±0.0098 | 0.8436±0.0053 | 0.8606±0.0325 | 0.7585±0.0207 | 0.8248±0.0125 | 0.8057±0.0343 | 0.8514±0.0213 | 0.8401±0.0153 | 0.8115±0.0153 |
| $8 \times 10^{-6}$ | 0.7844±0.0210 | 0.8536±0.0094 | 0.7587±0.0282 | 0.7918±0.0159 | 0.8102±0.0273 | 0.7989±0.0125 | 0.7989±0.0217 | 0.8103±0.0076 | 0.8061±0.0339 | 0.8325±0.0247 | 0.8034±0.0514 |
| $5 \times 10^{-6}$ | 0.8850±0.0634 | 0.8597±0.0217 | 0.7783±0.0245 | 0.8346±0.0281 | 0.8473±0.0400 | 0.8492±0.0526 | 0.8505±0.0519 | 0.8617±0.0261 | 0.8431±0.0268 | 0.8187±0.0314 | 0.8304±0.0387 |
| $3 \times 10^{-6}$ | 0.8508±0.0199 | 0.8491±0.0034 | 0.7657±0.0149 | 0.8293±0.0317 | 0.8062±0.0639 | 0.8466±0.0131 | 0.8505±0.0494 | 0.8266±0.0428 | 0.8254±0.0214 | 0.8292±0.0522 | 0.8453±0.0027 |
| $1 \times 10^{-6}$ | 0.8197±0.0568 | 0.8836±0.0216 | 0.8065±0.0506 | 0.8372±0.0384 | 0.8665±0.0376 | 0.8543±0.0218 | 0.8868±0.0447 | 0.8319±0.0460 | 0.8134±0.0318 | 0.8402±0.0557 | 0.8057±0.0303 |

**Table 6:** Numerical results for Gradient Flow with the Knot dataset ($d = 2$)

| LR | SW | MaxSW | DSW | MaxKSW | iMSW | viMSW | oMSW | rMSW | EBSW | RPSW | EBRPSW |
|---|---|---|---|---|---|---|---|---|---|---|---|
| 100 | 2.7116±0.6616 | 0.6408±0.0079 | 3.2280±0.5094 | 1.3326±0.0411 | 2.2337±0.8727 | 2.5000±0.8335 | 1.3862±0.2603 | 2.5671±0.8620 | 2.8073±0.1416 | 2.6577±0.0282 | 2.7618±0.1329 |
| 80 | 2.1367±0.6087 | 0.5710±0.0041 | 1.0384±0.3640 | 1.3753±0.1106 | 1.7140±0.1703 | 1.8681±0.6268 | 1.5932±0.1041 | 2.5953±0.6861 | 2.3348±0.0828 | 2.2001±0.0631 | 2.2090±0.1610 |
| 50 | 0.6877±0.0723 | 1.4925±0.8598 | 1.4048±0.1910 | 1.5581±0.0122 | 1.1232±0.1551 | 0.8977±0.2941 | 1.2540±0.0568 | 1.4821±0.1792 | 1.4024±0.1124 | 1.1371±0.0850 | 1.4613±0.0400 |
| 30 | 0.5389±0.2452 | 0.8486±0.0496 | 0.8765±0.1254 | 0.9592±0.0019 | 0.8816±0.0522 | 0.6488±0.3089 | 0.8072±0.0214 | 0.9189±0.1387 | 0.9178±0.3697 | 0.6777±0.0658 | 0.9473±0.1249 |
| 10 | 0.1721±0.0402 | 0.4116±0.0909 | 0.1965±0.0915 | 0.1744±0.0095 | 0.2179±0.1150 | 0.3502±0.0193 | 0.1614±0.0207 | 0.1752±0.0400 | 0.3453±0.0457 | 0.1866±0.1160 | 0.3806±0.0842 |
| 8 | 0.1892±0.0560 | 0.4166±0.0334 | 0.2441±0.0545 | 0.2037±0.0043 | 0.1355±0.0865 | 0.1707±0.0453 | 0.1800±0.0074 | 0.2334±0.0587 | 0.2948±0.1139 | 0.1311±0.0399 | 0.2549±0.1194 |
| 5 | 0.0907±0.0390 | 0.2481±0.0296 | 0.1453±0.0272 | 0.1270±0.0113 | 0.0916±0.0514 | 0.1663±0.0083 | 0.0956±0.0204 | 0.0990±0.0220 | 0.1724±0.0669 | 0.1171±0.0207 | 0.1779±0.0662 |
| 3 | 0.0738±0.0312 | 0.1655±0.0088 | 0.0793±0.0224 | 0.1074±0.0030 | 0.0767±0.0161 | 0.0645±0.0172 | 0.0645±0.0015 | 0.0593±0.0474 | 0.0931±0.0083 | 0.0656±0.0181 | 0.0988±0.0052 |
| 1 | 0.0167±0.0020 | 0.0788±0.0019 | 0.0264±0.0057 | 0.0489±0.0069 | 0.0205±0.0067 | 0.0167±0.0011 | 0.0191±0.0021 | 0.0148±0.0076 | 0.0234±0.0030 | 0.0230±0.0114 | 0.0208±0.0013 |
| $8 \times 10^{-1}$ | 0.0225±0.0040 | 0.0648±0.0041 | 0.0244±0.0036 | 0.0404±0.0040 | 0.0166±0.0036 | 0.0094±0.0042 | 0.0159±0.0007 | 0.0277±0.0018 | 0.0176±0.0034 | 0.0178±0.0031 | 0.0189±0.0066 |
| $5 \times 10^{-1}$ | 0.0090±0.0039 | 0.0482±0.0017 | 0.0147±0.0016 | 0.0251±0.0038 | 0.0078±0.0023 | 0.0125±0.0050 | 0.0106±0.0006 | 0.0099±0.0044 | 0.0095±0.0022 | 0.0101±0.0049 | 0.0086±0.0035 |
| $3 \times 10^{-1}$ | 0.0057±0.0014 | 0.0405±0.0012 | 0.0134±0.0078 | 0.0131±0.0008 | 0.0060±0.0010 | 0.0092±0.0013 | 0.0066±0.0003 | 0.0039±0.0021 | 0.0062±0.0012 | 0.0047±0.0011 | 0.0055±0.0015 |
| $1 \times 10^{-1}$ | 0.0018±0.0004 | 0.0138±0.0012 | 0.0019±0.0005 | 0.0031±0.0003 | 0.0078±0.0088 | 0.0021±0.0011 | 0.0020±0.0003 | 0.0013±0.0003 | 0.0018±0.0004 | 0.0020±0.0003 | 0.0019±0.0002 |
| $8 \times 10^{-2}$ | 0.0018±0.0004 | 0.0076±0.0007 | 0.0025±0.0001 | 0.0020±0.0002 | 0.0088±0.0099 | 0.0018±0.0002 | 0.0018±0.0001 | 0.0020±0.0001 | 0.0020±0.0002 | 0.0016±0.0005 | 0.0015±0.0004 |
| $5 \times 10^{-2}$ | 0.0010±0.0001 | 0.0038±0.0004 | 0.0017±0.0005 | 0.0016±0.0003 | 0.0015±0.0004 | 0.0009±0.0004 | 0.0010±0.0000 | 0.0010±0.0001 | 0.0009±0.0001 | 0.0010±0.0001 | 0.0007±0.0002 |
| $3 \times 10^{-2}$ | 0.0007±0.0004 | 0.0020±0.0003 | 0.0081±0.0104 | 0.0008±0.0000 | 0.0008±0.0001 | 0.0009±0.0002 | 0.0005±0.0001 | 0.0005±0.0001 | 0.0031±0.0030 | 0.0005±0.0001 | 0.0007±0.0002 |
| $1 \times 10^{-2}$ | 0.0002±0.0001 | 0.0004±0.0000 | 0.0003±0.0001 | 0.0002±0.0000 | 0.0001±0.0000 | 0.0071±0.0098 | 0.0001±0.0000 | 0.0002±0.0000 | 0.0002±0.0001 | 0.0002±0.0000 | 0.0003±0.0001 |
| $8 \times 10^{-3}$ | 0.0002±0.0001 | 0.0000±0.0000 | 0.0002±0.0001 | 0.0002±0.0001 | 0.0002±0.0000 | 0.0002±0.0001 | 0.0001±0.0001 | 0.0002±0.0001 | 0.0003±0.0001 | 0.0001±0.0001 | 0.0028±0.0037 |
| $5 \times 10^{-3}$ | 0.0001±0.0000 | 0.0113±0.0080 | 0.0023±0.0031 | 0.0006±0.0007 | 0.0074±0.0104 | 0.0166±0.0130 | 0.0014±0.0018 | 0.0012±0.0016 | 0.0018±0.0024 | 0.0001±0.0000 | 0.0101±0.0102 |
| $3 \times 10^{-3}$ | 0.0462±0.0025 | 0.0576±0.0029 | 0.0463±0.0037 | 0.0524±0.0014 | 0.0520±0.0014 | 0.0573±0.0045 | 0.0440±0.0008 | 0.0443±0.0063 | 0.0388±0.0036 | 0.0537±0.0011 | 0.0461±0.0020 |
| $1 \times 10^{-3}$ | 0.4562±0.0128 | 0.4431±0.0294 | 0.4015±0.0061 | 0.4037±0.0474 | 0.4144±0.0245 | 0.4502±0.0148 | 0.4378±0.0449 | 0.3841±0.0371 | 0.4561±0.0102 | 0.4135±0.0113 | 0.4053±0.0156 |
| $8 \times 10^{-4}$ | 0.5542±0.0443 | 0.5135±0.0036 | 0.4889±0.0376 | 0.5049±0.0621 | 0.4779±0.0132 | 0.4414±0.0209 | 0.5153±0.0144 | 0.4962±0.0564 | 0.5377±0.0322 | 0.5181±0.0430 | 0.4708±0.0153 |
| $5 \times 10^{-4}$ | 0.6267±0.0115 | 0.6978±0.0145 | 0.6018±0.0523 | 0.5962±0.0185 | 0.6326±0.0183 | 0.6380±0.0633 | 0.5696±0.0220 | 0.6054±0.0356 | 0.5978±0.0170 | 0.6307±0.0299 | 0.6033±0.0117 |
| $3 \times 10^{-4}$ | 0.6768±0.0396 | 0.7378±0.0884 | 0.6948±0.0818 | 0.6501±0.0650 | 0.7257±0.0189 | 0.6678±0.0099 | 0.7080±0.0066 | 0.6992±0.0195 | 0.6705±0.0717 | 0.7056±0.0626 | 0.6892±0.0496 |
| $1 \times 10^{-4}$ | 0.8010±0.0609 | 0.8032±0.0078 | 0.7756±0.0116 | 0.8050±0.0386 | 0.7569±0.0699 | 0.7831±0.0213 | 0.7689±0.0297 | 0.7545±0.0447 | 0.7353±0.0105 | 0.8001±0.0219 | 0.7609±0.0142 |
| $8 \times 10^{-5}$ | 0.7912±0.0136 | 0.8318±0.0096 | 0.7750±0.0291 | 0.7513±0.0187 | 0.7757±0.0432 | 0.7533±0.0463 | 0.7707±0.0304 | 0.7996±0.0452 | 0.7798±0.0275 | 0.8173±0.0162 | 0.7751±0.0186 |
| $5 \times 10^{-5}$ | 0.7931±0.0177 | 0.8609±0.0384 | 0.7861±0.0270 | 0.7883±0.0147 | 0.7286±0.0337 | 0.7751±0.0454 | 0.7664±0.0712 | 0.7524±0.0513 | 0.7911±0.0444 | 0.7857±0.0189 | 0.7777±0.0174 |
| $3 \times 10^{-5}$ | 0.7866±0.0157 | 0.8463±0.0257 | 0.8185±0.0083 | 0.8197±0.0474 | 0.8045±0.0219 | 0.7678±0.0227 | 0.8187±0.0423 | 0.7756±0.0405 | 0.8055±0.0188 | 0.7912±0.0439 | 0.7578±0.0359 |
| $1 \times 10^{-5}$ | 0.7728±0.0195 | 0.8348±0.0121 | 0.8120±0.0121 | 0.7983±0.0219 | 0.7745±0.0238 | 0.7827±0.0089 | 0.8054±0.0241 | 0.8240±0.0307 | 0.8172±0.0239 | 0.7912±0.0288 | 0.7982±0.0576 |
| $8 \times 10^{-6}$ | 0.7996±0.0623 | 0.8318±0.0096 | 0.7549±0.0266 | 0.8504±0.0217 | 0.8288±0.0260 | 0.8287±0.0296 | 0.8424±0.0903 | 0.8012±0.0562 | 0.7854±0.0054 | 0.8035±0.0243 | 0.7890±0.0228 |
| $5 \times 10^{-6}$ | 0.8437±0.0315 | 0.8609±0.0384 | 0.8357±0.0442 | 0.8353±0.0145 | 0.7918±0.0215 | 0.7905±0.0248 | 0.7750±0.0088 | 0.8025±0.0390 | 0.8363±0.0332 | 0.7982±0.0254 | 0.8205±0.0078 |
| $3 \times 10^{-6}$ | 0.7935±0.0268 | 0.8463±0.0257 | 0.8084±0.0411 | 0.8457±0.0174 | 0.7783±0.0195 | 0.7867±0.0324 | 0.8515±0.0514 | 0.8093±0.0442 | 0.8079±0.0208 | 0.8266±0.0051 | 0.8228±0.0168 |
| $1 \times 10^{-6}$ | 0.8332±0.0122 | 0.8348±0.0121 | 0.8042±0.0282 | 0.8091±0.0190 | 0.7770±0.0248 | 0.7748±0.0164 | 0.7932±0.0170 | 0.8373±0.0212 | 0.8203±0.0194 | 0.8121±0.0115 | 0.8057±0.0246 |

**Table 7:** Numerical results for Gradient Flow with the Swiss dataset ($d = 2$)

| LR | SW | MaxSW | DSW | MaxKSW | iMSW | viMSW | oMSW | rMSW | EBSW | RPSW | EBRPSW |
|---|---|---|---|---|---|---|---|---|---|---|---|
| 100 | 1.8754±1.2183 | 3.6719±1.6669 | 2.2167±1.3910 | 1.1766±0.0901 | 2.0996±0.5058 | 2.3956±0.7993 | 3.0252±0.1706 | 2.6436±1.0882 | 3.0252±0.1706 | 2.6426±0.0669 | 2.8145±0.1491 |
| 80 | 1.6682±0.9727 | 1.8203±0.9088 | 2.2781±0.4102 | 1.1566±0.0318 | 2.0056±0.7764 | 1.8520±0.5824 | 2.2930±0.0903 | 1.8364±1.0702 | 2.2930±0.0903 | 2.0552±0.1070 | 2.3738±0.0394 |
| 50 | 0.7122±0.1076 | 1.5041±0.2527 | 0.7793±0.4214 | 0.9272±0.0260 | 1.1239±0.3555 | 1.3310±0.4241 | 1.7172±0.1023 | 1.4756±0.3155 | 1.7172±0.1023 | 1.1673±0.1447 | 1.4905±0.0416 |
| 30 | 0.6045±0.3404 | 1.0704±0.0587 | 0.6909±0.3766 | 0.7364±0.0218 | 0.7664±0.0791 | 0.4753±0.1999 | 1.0015±0.0625 | 0.8430±0.2975 | 1.0015±0.0625 | 0.6435±0.3303 | 0.9820±0.1278 |
| 10 | 0.1544±0.0464 | 0.4782±0.0479 | 0.2348±0.0887 | 0.1933±0.0312 | 0.1795±0.0326 | 0.3404±0.0127 | 0.2363±0.0632 | 0.2242±0.0245 | 0.2363±0.0632 | 0.2729±0.0176 | 0.3536±0.0355 |
| 8 | 0.2014±0.0251 | 0.3462±0.0410 | 0.2254±0.0901 | 0.1854±0.0010 | 0.1386±0.0738 | 0.1293±0.0563 | 0.3805±0.0112 | 0.1638±0.0318 | 0.3805±0.0112 | 0.2027±0.0718 | 0.0854±0.0351 |
| 5 | 0.0995±0.0458 | 0.2004±0.0136 | 0.0772±0.0469 | 0.0998±0.0078 | 0.1063±0.0017 | 0.1894±0.0047 | 0.0780±0.0013 | 0.2197±0.0233 | 0.0780±0.0013 | 0.1016±0.0034 | 0.1692±0.0604 |
| 3 | 0.0868±0.0193 | 0.1177±0.0058 | 0.0640±0.0260 | 0.0613±0.0033 | 0.0885±0.0218 | 0.4035±0.0251 | 0.1012±0.0058 | 0.4207±0.0243 | 0.1012±0.0058 | 0.0676±0.0091 | 0.0958±0.0013 |
| 1 | 0.0230±0.0086 | 0.0436±0.0010 | 0.0245±0.0084 | 0.0230±0.0008 | 0.0183±0.0091 | 0.7765±0.0445 | 0.0187±0.0014 | 0.8306±0.0391 | 0.0171±0.0014 | 0.0250±0.0009 | 0.0209±0.0019 |
| $8\times10^{-1}$ | 0.0246±0.0058 | 0.0342±0.0012 | 0.0168±0.0050 | 0.0189±0.0005 | 0.0171±0.0011 | 0.8516±0.0207 | 0.0139±0.0012 | 0.8743±0.0473 | 0.0177±0.0050 | 0.0217±0.0016 | 0.0151±0.0036 |
| $5\times10^{-1}$ | 0.0091±0.0027 | 0.0233±0.0006 | 0.0103±0.0042 | 0.0146±0.0006 | 0.0096±0.0027 | 0.9355±0.0290 | 0.0090±0.0007 | 0.9269±0.0258 | 0.0092±0.0024 | 0.0142±0.0033 | 0.0099±0.0037 |
| $3\times10^{-1}$ | 0.0070±0.0006 | 0.0158±0.0001 | 0.0064±0.0027 | 0.0119±0.0005 | 0.0078±0.0019 | 0.9861±0.0259 | 0.0062±0.0002 | 1.0561±0.0192 | 0.0041±0.0019 | 0.0073±0.0002 | 0.0059±0.0017 |
| $1\times10^{-1}$ | 0.0020±0.0002 | 0.0097±0.0002 | 0.0018±0.0010 | 0.0059±0.0009 | 0.0025±0.0007 | 1.0240±0.0293 | 0.0017±0.0001 | 1.0638±0.0370 | 0.0015±0.0005 | 0.0027±0.0008 | 0.0021±0.0007 |
| $8\times10^{-2}$ | 0.0019±0.0003 | 0.0076±0.0002 | 0.0021±0.0003 | 0.0040±0.0002 | 0.0020±0.0001 | 1.0657±0.0399 | 0.0017±0.0001 | 1.0979±0.0350 | 0.0018±0.0003 | 0.0019±0.0003 | 0.0013±0.0001 |
| $5\times10^{-2}$ | 0.0010±0.0003 | 0.0061±0.0000 | 0.0006±0.0002 | 0.0031±0.0002 | 0.0013±0.0002 | 1.0926±0.0206 | 0.0010±0.0001 | 1.1166±0.0107 | 0.0012±0.0001 | 0.0014±0.0005 | 0.0006±0.0004 |
| $3\times10^{-2}$ | 0.0008±0.0002 | 0.0031±0.0003 | 0.0010±0.0001 | 0.0018±0.0002 | 0.0008±0.0002 | 1.1155±0.0033 | 0.0006±0.0001 | 1.1021±0.0320 | 0.0009±0.0001 | 0.0007±0.0002 | 0.0007±0.0002 |
| $1\times10^{-2}$ | 0.0002±0.0001 | 0.1316±0.0190 | 0.0002±0.0001 | 0.0384±0.0497 | 0.0002±0.0000 | 0.0003±0.0001 | 0.0002±0.0000 | 0.0003±0.0313 | 0.0001±0.0000 | 0.0002±0.0000 | 0.0009±0.0001 |
| $8\times10^{-3}$ | 0.0007±0.0009 | 0.1939±0.0405 | 0.0001±0.0001 | 0.1640±0.0036 | 0.0002±0.0000 | 0.0002±0.0000 | 0.0002±0.0000 | 0.0002±0.0211 | 0.1260±0.0075 | 0.0014±0.0009 | 0.1106±0.0398 |
| $5\times10^{-3}$ | 0.1965±0.0290 | 0.3497±0.0149 | 0.2026±0.0270 | 0.3047±0.0217 | 0.2563±0.0217 | 0.1894±0.0047 | 0.2271±0.0173 | 0.2197±0.0233 | 0.3223±0.0112 | 0.1987±0.0204 | 0.2964±0.0164 |
| $3\times10^{-3}$ | 0.4310±0.0204 | 0.4287±0.0222 | 0.4433±0.0227 | 0.4484±0.0108 | 0.4080±0.0145 | 0.4035±0.0181 | 0.4203±0.0266 | 0.4207±0.0243 | 0.4304±0.0058 | 0.4490±0.0226 | 0.4122±0.0402 |
| $1\times10^{-3}$ | 0.8561±0.0290 | 0.8367±0.0442 | 0.7788±0.0083 | 0.8168±0.0181 | 0.7574±0.0171 | 0.7765±0.0445 | 0.8061±0.0452 | 0.8306±0.0391 | 0.7096±0.0249 | 0.8049±0.0369 | 0.7294±0.0128 |
| $8\times10^{-4}$ | 0.8471±0.0388 | 0.8914±0.0049 | 0.8349±0.0178 | 0.8478±0.0113 | 0.8654±0.0206 | 0.8516±0.0207 | 0.8791±0.0156 | 0.8743±0.0473 | 0.8134±0.0245 | 0.8864±0.0115 | 0.7763±0.0230 |
| $5\times10^{-4}$ | 0.9638±0.0207 | 0.9201±0.0121 | 0.9686±0.0407 | 0.9992±0.0064 | 0.9837±0.0163 | 0.9355±0.0290 | 0.9339±0.0409 | 0.9269±0.0258 | 0.9093±0.0198 | 0.9818±0.0177 | 0.9462±0.0092 |
| $3\times10^{-4}$ | 1.0198±0.0345 | 0.9876±0.0126 | 0.9971±0.0412 | 0.9834±0.0111 | 1.0509±0.0250 | 0.9861±0.0259 | 1.0586±0.0182 | 1.0561±0.0192 | 0.9803±0.0350 | 0.9542±0.0045 | 0.9542±0.0045 |
| $1\times10^{-4}$ | 1.0491±0.0545 | 1.0148±0.0181 | 1.0628±0.0423 | 1.0362±0.0304 | 0.9967±0.0192 | 1.0240±0.0293 | 1.0540±0.0237 | 1.0638±0.0370 | 1.0043±0.0284 | 1.0486±0.0266 | 1.0699±0.0781 |
| $8\times10^{-5}$ | 1.1368±0.0152 | 1.0651±0.0629 | 1.0756±0.0461 | 1.0883±0.0655 | 1.0514±0.0400 | 1.0657±0.0206 | 1.1419±0.0481 | 1.0979±0.0350 | 1.1419±0.0481 | 1.0805±0.0667 | 1.0124±0.0534 |
| $5\times10^{-5}$ | 1.1340±0.0280 | 0.9856±0.0165 | 0.9856±0.0501 | 1.1182±0.0655 | 1.0685±0.0534 | 1.0926±0.0206 | 1.0581±0.0062 | 1.1166±0.0107 | 1.0581±0.0062 | 1.1368±0.0152 | 1.1027±0.0240 |
| $3\times10^{-5}$ | 1.1742±0.0390 | 1.1101±0.0456 | 1.1101±0.0523 | 1.1371±0.0335 | 1.1516±0.0201 | 1.1155±0.0033 | 1.0978±0.0461 | 1.1021±0.0320 | 1.0978±0.0461 | 1.1340±0.0280 | 1.1002±0.0187 |
| $1\times10^{-5}$ | 1.1502±0.0119 | 1.1172±0.0257 | 1.1101±0.0553 | 1.0822±0.0282 | 1.1155±0.0244 | 1.1149±0.0311 | 1.1149±0.0311 | 1.0607±0.0759 | 1.1149±0.0311 | 1.1742±0.0390 | 1.1040±0.0386 |
| $8\times10^{-6}$ | 1.1234±0.0241 | 1.1757±0.0295 | 1.1211±0.0590 | 1.1268±0.0387 | 1.1363±0.0206 | 1.1419±0.0481 | 1.1419±0.0481 | 1.1154±0.0252 | 1.1419±0.0481 | 1.1368±0.0152 | 1.1397±0.0270 |
| $5\times10^{-6}$ | 1.0738±0.0273 | 1.0898±0.0584 | 1.1256±0.0493 | 1.1105±0.0139 | 1.1288±0.0081 | 1.1205±0.0187 | 1.0581±0.0062 | 1.1064±0.0194 | 1.0581±0.0062 | 1.1340±0.0280 | 1.1027±0.0240 |
| $3\times10^{-6}$ | 1.1109±0.0355 | 1.1249±0.0357 | 1.0902±0.0320 | 1.1371±0.0335 | 1.1040±0.0153 | 1.1130±0.0274 | 1.0978±0.0461 | 1.0785±0.0528 | 1.0978±0.0461 | 1.1742±0.0390 | 1.1002±0.0187 |
| $1\times10^{-6}$ | 1.1502±0.0119 | 1.1172±0.0257 | 1.1101±0.0553 | 1.0822±0.0282 | 1.1097±0.0482 | 1.1694±0.0198 | 1.1149±0.0311 | 1.0951±0.0297 | 1.1149±0.0311 | 1.1284±0.0389 | 1.1040±0.0386 |

**Table 8:** Numerical results for Gradient Flow with the 8 Gaussians dataset ($d = 2$)

| LR | SW | MaxSW | DSW | MaxKSW | iMSW | viMSW | oMSW | rMSW | EBSW | RPSW | EBRPSW |
|---|---|---|---|---|---|---|---|---|---|---|---|
| 100 | 0.7900±0.0704 | 4.4368±0.0966 | 3.0902±0.1026 | 2.9073±0.3947 | 2.2985±0.1220 | 1.2282±0.0578 | 0.8969±0.0500 | 1.2429±0.1167 | 10.5129±0.2240 | 1.2812±0.0165 | 11.0450±0.8609 |
| 80 | 0.6047±0.0850 | 3.4799±0.0544 | 2.4792±0.2602 | 2.0292±0.2711 | 1.7091±0.0921 | 0.9662±0.0284 | 0.6897±0.0442 | 0.9003±0.0392 | 8.5668±0.3944 | 1.0411±0.0586 | 7.8665±0.8317 |
| 50 | 0.3459±0.0234 | 2.7423±0.5278 | 1.0806±0.3335 | 1.7169±0.1028 | 1.0745±0.0742 | 0.5512±0.0653 | 0.3995±0.0698 | 0.5736±0.0181 | 4.3396±0.3767 | 0.6150±0.0368 | 4.0199±0.0448 |
| 30 | 0.2229±0.0178 | 1.8470±0.1703 | 0.8920±0.0499 | 1.5803±0.1051 | 0.7473±0.0193 | 0.3307±0.0363 | 0.2335±0.0071 | 0.3464±0.0307 | 2.3719±0.0339 | 0.3622±0.0363 | 2.7110±0.3379 |
| 10 | 0.0819±0.0056 | 1.2519±0.0267 | 0.1853±0.0391 | 0.9043±0.0135 | 0.5119±0.0033 | 0.1162±0.0097 | 0.0825±0.0035 | 0.1102±0.0206 | 1.0896±0.0531 | 0.1307±0.0169 | 1.0587±0.0079 |
| 8 | 0.0592±0.0014 | 1.0781±0.0115 | 0.1659±0.0566 | 0.7843±0.0256 | 0.4259±0.0210 | 0.0821±0.0011 | 0.0590±0.0082 | 0.0835±0.0041 | 1.0515±0.0428 | 0.1071±0.0068 | 0.9913±0.0529 |
| 5 | 0.0402±0.0026 | 0.7808±0.0114 | 0.0734±0.0121 | 0.5545±0.0083 | 0.3104±0.0177 | 0.0649±0.0070 | 0.0410±0.0007 | 0.0560±0.0027 | 0.0412±0.0027 | 0.0693±0.0138 | 0.0429±0.0037 |
| 3 | 0.0234±0.0026 | 0.5466±0.0103 | 0.0374±0.0089 | 0.4019±0.0028 | 0.2041±0.0022 | 0.0386±0.0027 | 0.0265±0.0017 | 0.0348±0.0009 | 0.0258±0.0015 | 0.0430±0.0029 | 0.0217±0.0019 |
| 1 | 0.0073±0.0002 | 0.2788±0.0092 | 0.0125±0.0025 | 0.1792±0.0007 | 0.0912±0.0016 | 0.0113±0.0013 | 0.0070±0.0009 | 0.0115±0.0005 | 0.0081±0.0008 | 0.0144±0.0009 | 0.0089±0.0009 |
| $8\times10^{-1}$ | 0.0067±0.0007 | 0.2448±0.0066 | 0.0099±0.0007 | 0.1514±0.0017 | 0.0805±0.0016 | 0.0082±0.0013 | 0.0063±0.0003 | 0.0094±0.0010 | 0.0063±0.0006 | 0.0117±0.0003 | 0.0062±0.0004 |
| $5\times10^{-1}$ | 0.0038±0.0004 | 0.1762±0.0045 | 0.0064±0.0006 | 0.1104±0.0031 | 0.0573±0.0011 | 0.0057±0.0003 | 0.0036±0.0003 | 0.0057±0.0003 | 0.0042±0.0002 | 0.0074±0.0006 | 0.0043±0.0003 |
| $3\times10^{-1}$ | 0.0026±0.0002 | 0.1282±0.0012 | 0.0036±0.0001 | 0.0788±0.0018 | 0.0381±0.0004 | 0.0034±0.0001 | 0.0023±0.0001 | 0.0035±0.0001 | 0.0022±0.0003 | 0.0041±0.0003 | 0.0023±0.0001 |
| $1\times10^{-1}$ | 0.0008±0.0000 | 0.0597±0.0009 | 0.0015±0.0000 | 0.0342±0.0005 | 0.0107±0.0002 | 0.0012±0.0001 | 0.0008±0.0000 | 0.0012±0.0001 | 0.0008±0.0001 | 0.0015±0.0001 | 0.0008±0.0000 |
| $8\times10^{-2}$ | 0.0006±0.0000 | 0.0499±0.0003 | 0.0010±0.0000 | 0.0282±0.0004 | 0.0086±0.0000 | 0.0010±0.0000 | 0.0006±0.0000 | 0.0009±0.0000 | 0.0006±0.0000 | 0.0012±0.0001 | 0.0006±0.0000 |
| $5\times10^{-2}$ | 0.0004±0.0000 | 0.0342±0.0003 | 0.0007±0.0000 | 0.0171±0.0003 | 0.0054±0.0000 | 0.0006±0.0000 | 0.0003±0.0000 | 0.0006±0.0000 | 0.0004±0.0000 | 0.0008±0.0000 | 0.0004±0.0000 |
| $3\times10^{-2}$ | 0.0002±0.0000 | 0.0272±0.0072 | 0.0004±0.0000 | 0.0080±0.0000 | 0.0031±0.0000 | 0.0004±0.0000 | 0.0002±0.0000 | 0.0003±0.0000 | 0.0003±0.0000 | 0.0005±0.0001 | 0.0002±0.0000 |
| $1\times10^{-2}$ | 0.1699±0.0262 | 0.0530±0.0060 | 0.0395±0.0020 | 0.0639±0.0039 | 0.1088±0.0063 | 0.1371±0.0141 | 0.1715±0.0089 | 0.1507±0.0100 | 0.0535±0.0008 | 0.0056±0.0077 | 0.0641±0.0027 |
| $8\times10^{-3}$ | 0.2678±0.0292 | 0.0706±0.0028 | 0.0564±0.0037 | 0.0922±0.0046 | 0.1784±0.0108 | 0.2101±0.0076 | 0.2381±0.0090 | 0.2667±0.0288 | 0.0968±0.0027 | 0.0010±0.0012 | 0.0925±0.0011 |
| $5\times10^{-3}$ | 0.4264±0.0227 | 0.1373±0.0006 | 0.1307±0.0060 | 0.2096±0.0176 | 0.3224±0.0325 | 0.3775±0.0155 | 0.4109±0.0132 | 0.3907±0.0216 | 0.1715±0.0034 | 0.0770±0.0092 | 0.1853±0.0080 |
| $3\times10^{-3}$ | 0.5863±0.0435 | 0.2441±0.0004 | 0.2907±0.0178 | 0.3481±0.0071 | 0.4836±0.0280 | 0.5586±0.0159 | 0.5553±0.0304 | 0.5572±0.0043 | 0.2927±0.0148 | 0.2330±0.0223 | 0.3111±0.0292 |
| $1\times10^{-3}$ | 0.7987±0.0118 | 0.7585±0.0052 | 0.7517±0.0287 | 0.7607±0.0420 | 0.7778±0.0164 | 0.7404±0.0106 | 0.7858±0.0109 | 0.7806±0.0178 | 0.7751±0.0066 | 0.7427±0.0242 | 0.7704±0.0552 |
| $8\times10^{-4}$ | 0.7785±0.0360 | 0.7701±0.0215 | 0.7445±0.0253 | 0.7571±0.0562 | 0.7751±0.0262 | 0.7760±0.0339 | 0.7826±0.0458 | 0.7852±0.0142 | 0.7615±0.0246 | 0.7313±0.0053 | 0.7520±0.0261 |
| $5\times10^{-4}$ | 0.7931±0.0415 | 0.7580±0.0470 | 0.7146±0.0403 | 0.7738±0.0153 | 0.7709±0.0115 | 0.7612±0.0216 | 0.7837±0.0457 | 0.8190±0.0229 | 0.7759±0.0414 | 0.7704±0.0147 | 0.7323±0.0210 |
| $3\times10^{-4}$ | 0.7726±0.0087 | 0.7806±0.0640 | 0.7752±0.0142 | 0.7582±0.0184 | 0.7892±0.0316 | 0.7936±0.0294 | 0.7779±0.0213 | 0.7843±0.0390 | 0.7901±0.0189 | 0.7528±0.0385 | 0.7730±0.0263 |
| $1\times10^{-4}$ | 0.7724±0.0251 | 0.7373±0.0223 | 0.7380±0.0580 | 0.7984±0.0495 | 0.7581±0.0290 | 0.8025±0.0279 | 0.7750±0.0102 | 0.7991±0.0153 | 0.7713±0.0014 | 0.7643±0.0285 | 0.7380±0.0197 |
| $8\times10^{-5}$ | 0.7802±0.0147 | 0.7643±0.0223 | 0.7921±0.0231 | 0.7104±0.0534 | 0.7957±0.0327 | 0.7725±0.0192 | 0.7747±0.0062 | 0.7413±0.0228 | 0.7748±0.0297 | 0.7585±0.0187 | 0.7867±0.0122 |
| $5\times10^{-5}$ | 0.7682±0.0403 | 0.7974±0.0291 | 0.8113±0.0685 | 0.7807±0.0192 | 0.7793±0.0381 | 0.8181±0.0332 | 0.8045±0.0346 | 0.7838±0.0321 | 0.7908±0.0161 | 0.7631±0.0420 | 0.7142±0.0102 |
| $3\times10^{-5}$ | 0.7980±0.0513 | 0.7476±0.0426 | 0.7775±0.0425 | 0.7561±0.0282 | 0.7576±0.0305 | 0.7579±0.0212 | 0.7556±0.0193 | 0.7656±0.0462 | 0.7890±0.0293 | 0.8213±0.0123 | 0.7827±0.0255 |
| $1\times10^{-5}$ | 0.7860±0.0132 | 0.7859±0.0271 | 0.7955±0.0397 | 0.7627±0.0285 | 0.7747±0.0025 | 0.7576±0.0395 | 0.7869±0.0353 | 0.7682±0.0255 | 0.7753±0.0191 | 0.7573±0.0311 | 0.7831±0.0199 |
| $8\times10^{-6}$ | 0.7860±0.0132 | 0.7859±0.0271 | 0.7955±0.0397 | 0.7627±0.0285 | 0.7747±0.0025 | 0.7576±0.0395 | 0.7869±0.0353 | 0.7682±0.0255 | 0.7753±0.0191 | 0.7573±0.0311 | 0.7831±0.0199 |
| $5\times10^{-6}$ | 0.7980±0.0513 | 0.7476±0.0426 | 0.7775±0.0282 | 0.7561±0.0282 | 0.7576±0.0305 | 0.7579±0.0212 | 0.7556±0.0193 | 0.7656±0.0462 | 0.7890±0.0293 | 0.8213±0.0123 | 0.7827±0.0255 |
| $3\times10^{-6}$ | 0.7980±0.0513 | 0.7476±0.0426 | 0.7775±0.0425 | 0.7561±0.0282 | 0.7579±0.0305 | 0.7579±0.0212 | 0.7556±0.0193 | 0.7656±0.0462 | 0.7890±0.0293 | 0.8213±0.0123 | 0.7827±0.0255 |
| $1\times10^{-6}$ | 0.7860±0.0132 | 0.7859±0.0271 | 0.7955±0.0397 | 0.7627±0.0285 | 0.7747±0.0025 | 0.7576±0.0395 | 0.7869±0.0353 | 0.7682±0.0255 | 0.7753±0.0191 | 0.7573±0.0311 | 0.7831±0.0199 |

**Table 9:** Numerical results for Gradient Flow with the Knot dataset ($d = 50$)

| LR | SW | MaxSW | DSW | MaxKSW | iMSW | viMSW | oMSW | rMSW | EBSW | RPSW | EBRPSW |
|---|---|---|---|---|---|---|---|---|---|---|---|
| 100 | 0.7547±0.0159 | 5.2763±0.0581 | 2.7308±0.2356 | 3.2047±0.1553 | 2.2315±0.0152 | 1.1926±0.1009 | 0.7655±0.0725 | 1.0225±0.0259 | 8.2665±0.6763 | 1.2582±0.1625 | 8.2852±0.4012 |
| 80 | 0.7061±0.0588 | 4.4078±0.8741 | 2.1265±0.5707 | 2.6758±0.2432 | 1.5470±0.0262 | 0.9951±0.0213 | 0.6729±0.0501 | 0.8837±0.0816 | 6.7290±0.0731 | 1.1278±0.1272 | 6.6422±0.3327 |
| 50 | 0.4126±0.0458 | 3.1516±0.4662 | 1.5598±0.1044 | 2.6188±0.1909 | 1.0823±0.0751 | 0.6173±0.0160 | 0.4052±0.0346 | 0.5515±0.0068 | 4.5307±0.1181 | 0.6737±0.0310 | 4.1734±0.1881 |
| 30 | 0.2164±0.0158 | 2.5630±0.0389 | 0.8967±0.0765 | 2.0890±0.0750 | 1.2381±0.0289 | 0.3187±0.0404 | 0.2560±0.0072 | 0.3502±0.0163 | 2.8793±0.0917 | 0.4439±0.0344 | 2.8941±0.1512 |
| 10 | 0.0798±0.0005 | 1.4811±0.0137 | 0.1845±0.0386 | 1.1085±0.0246 | 0.6727±0.0053 | 0.1076±0.0092 | 0.0733±0.0010 | 0.1136±0.0098 | 1.6626±0.0213 | 0.1442±0.0078 | 1.5850±0.0564 |
| 8 | 0.0701±0.0045 | 1.2975±0.0386 | 0.1219±0.0369 | 0.9595±0.0109 | 0.5404±0.0106 | 0.0885±0.0099 | 0.0584±0.0057 | 0.0884±0.0097 | 1.3687±0.0730 | 0.1110±0.0185 | 1.4399±0.0461 |
| 5 | 0.0387±0.0020 | 1.0414±0.0267 | 0.0892±0.0231 | 0.7089±0.0243 | 0.3520±0.0050 | 0.0595±0.0041 | 0.0373±0.0030 | 0.0508±0.0028 | 0.0395±0.0016 | 0.0723±0.0041 | 0.0426±0.0054 |
| 3 | 0.0253±0.0009 | 0.7394±0.0176 | 0.0374±0.0112 | 0.4655±0.0047 | 0.1601±0.0228 | 0.0301±0.0014 | 0.0235±0.0011 | 0.0314±0.0018 | 0.0226±0.0020 | 0.0438±0.0020 | 0.0239±0.0020 |
| 1 | 0.0079±0.0003 | 0.2967±0.0070 | 0.0146±0.0022 | 0.1618±0.0059 | 0.0558±0.0016 | 0.0112±0.0014 | 0.0076±0.0004 | 0.0120±0.0013 | 0.0078±0.0009 | 0.0150±0.0005 | 0.0089±0.0003 |
| $8 \times 10^{-1}$ | 0.0065±0.0005 | 0.2347±0.0065 | 0.0106±0.0018 | 0.0932±0.0034 | 0.0469±0.0011 | 0.0089±0.0002 | 0.0059±0.0003 | 0.0091±0.0008 | 0.0068±0.0004 | 0.0126±0.0013 | 0.0059±0.0003 |
| $5 \times 10^{-1}$ | 0.0039±0.0002 | 0.1408±0.0040 | 0.0072±0.0003 | 0.0627±0.0017 | 0.0350±0.0005 | 0.0053±0.0004 | 0.0043±0.0003 | 0.0056±0.0006 | 0.0038±0.0002 | 0.0081±0.0007 | 0.0039±0.0003 |
| $3 \times 10^{-1}$ | 0.0025±0.0001 | 0.0740±0.0014 | 0.0038±0.0004 | 0.0448±0.0006 | 0.0256±0.0004 | 0.0033±0.0003 | 0.0024±0.0003 | 0.0035±0.0002 | 0.0024±0.0004 | 0.0052±0.0002 | 0.0024±0.0002 |
| $1 \times 10^{-1}$ | 0.0009±0.0001 | 0.0338±0.0004 | 0.0014±0.0001 | 0.0250±0.0001 | 0.0123±0.0000 | 0.0013±0.0001 | 0.0008±0.0001 | 0.0011±0.0002 | 0.0008±0.0002 | 0.0017±0.0001 | 0.0009±0.0001 |
| $8 \times 10^{-2}$ | 0.0006±0.0001 | 0.0303±0.0009 | 0.0012±0.0002 | 0.0216±0.0004 | 0.0102±0.0004 | 0.0008±0.0001 | 0.0006±0.0000 | 0.0010±0.0000 | 0.0007±0.0000 | 0.0013±0.0001 | 0.0006±0.0000 |
| $5 \times 10^{-2}$ | 0.0004±0.0000 | 0.0246±0.0009 | 0.0007±0.0001 | 0.0154±0.0001 | 0.0065±0.0001 | 0.0006±0.0001 | 0.0004±0.0001 | 0.0006±0.0000 | 0.0004±0.0000 | 0.0009±0.0001 | 0.0004±0.0000 |
| $3 \times 10^{-2}$ | 0.1207±0.0565 | 0.0171±0.0004 | 0.0004±0.0001 | 0.1128±0.0265 | 0.1479±0.0204 | 0.0808±0.0513 | 0.0953±0.0630 | 0.1041±0.0235 | 0.0002±0.0000 | 0.0005±0.0000 | 0.0002±0.0000 |
| $1 \times 10^{-2}$ | 0.5471±0.0314 | 0.3074±0.0063 | 0.2104±0.0210 | 0.3963±0.0099 | 0.4644±0.0191 | 0.5370±0.0153 | 0.5546±0.0361 | 0.5783±0.0261 | 0.3110±0.0059 | 0.1862±0.0244 | 0.3056±0.0195 |
| $8 \times 10^{-3}$ | 0.6607±0.0252 | 0.3471±0.0367 | 0.2931±0.0161 | 0.4406±0.0058 | 0.5575±0.0223 | 0.6040±0.0050 | 0.6462±0.0206 | 0.6438±0.0218 | 0.3705±0.0090 | 0.2789±0.0171 | 0.3714±0.0098 |
| $5 \times 10^{-3}$ | 0.8309±0.0754 | 0.4564±0.0277 | 0.4910±0.0170 | 0.5249±0.0144 | 0.6980±0.0091 | 0.7606±0.0329 | 0.7976±0.0299 | 0.8461±0.0325 | 0.5150±0.0186 | 0.4749±0.0196 | 0.4967±0.0190 |
| $3 \times 10^{-3}$ | 0.9346±0.0160 | 0.5658±0.0233 | 0.6897±0.0284 | 0.6755±0.0063 | 0.7964±0.0128 | 0.8806±0.0164 | 0.9133±0.0362 | 0.9116±0.0411 | 0.6939±0.0332 | 0.6552±0.0181 | 0.7147±0.0048 |
| $1 \times 10^{-3}$ | 1.0732±0.0251 | 0.8076±0.0251 | 0.9769±0.0156 | 0.9318±0.0178 | 1.0055±0.0251 | 1.0421±0.0599 | 1.0351±0.0384 | 0.9116±0.0010 | 0.9582±0.0105 | 0.9311±0.0332 | 0.9782±0.0298 |
| $8 \times 10^{-4}$ | 1.0660±0.0099 | 0.8543±0.0055 | 0.9574±0.0476 | 0.9526±0.0065 | 1.0176±0.0510 | 1.0811±0.0094 | 1.0274±0.0524 | 1.0162±0.0435 | 1.0022±0.0106 | 1.0049±0.0174 | 0.9958±0.0305 |
| $5 \times 10^{-4}$ | 1.0721±0.0250 | 0.9709±0.0153 | 0.9846±0.0195 | 0.9930±0.0107 | 1.0416±0.0409 | 1.0991±0.0175 | 1.1032±0.0322 | 1.0534±0.0326 | 1.0223±0.0399 | 1.0613±0.0227 | 1.0550±0.0291 |
| $3 \times 10^{-4}$ | 1.0848±0.0382 | 1.0400±0.0263 | 1.0311±0.0734 | 1.0448±0.0222 | 1.0676±0.0230 | 1.0752±0.0319 | 1.0726±0.0208 | 1.1065±0.0231 | 1.0811±0.0291 | 1.0522±0.0462 | 1.0969±0.0233 |
| $1 \times 10^{-4}$ | 1.1030±0.0147 | 1.0385±0.0311 | 1.1083±0.0502 | 1.1365±0.0700 | 1.1079±0.0175 | 1.1241±0.0141 | 1.0896±0.0660 | 1.1286±0.0108 | 1.1013±0.0297 | 1.0749±0.0316 | 1.1096±0.0245 |
| $8 \times 10^{-5}$ | 1.0942±0.0655 | 1.0906±0.0433 | 1.1044±0.0174 | 1.1250±0.0374 | 1.1226±0.0454 | 1.0875±0.0187 | 1.0936±0.0374 | 1.0882±0.0041 | 1.0679±0.0441 | 1.1070±0.0730 | 1.0884±0.0650 |
| $5 \times 10^{-5}$ | 1.1349±0.0215 | 1.0513±0.0381 | 1.0550±0.0217 | 1.1064±0.0151 | 1.1607±0.0343 | 1.1231±0.0358 | 1.0709±0.0248 | 1.1488±0.0098 | 1.0924±0.0221 | 1.0870±0.0166 | 1.0661±0.0188 |
| $3 \times 10^{-5}$ | 1.0990±0.0313 | 1.0985±0.0104 | 1.0675±0.0382 | 1.0855±0.0425 | 1.1012±0.0343 | 1.1434±0.0657 | 1.1014±0.0307 | 1.1131±0.0082 | 1.1001±0.0724 | 1.1312±0.0281 | 1.1283±0.0125 |
| $1 \times 10^{-5}$ | 1.0485±0.0162 | 1.1895±0.0512 | 1.0654±0.0337 | 1.1409±0.0482 | 1.1170±0.0215 | 1.1335±0.0207 | 1.1062±0.0194 | 1.1249±0.0395 | 1.1434±0.0072 | 1.1078±0.0490 | 1.1283±0.0508 |
| $8 \times 10^{-6}$ | 1.1477±0.0352 | 1.1061±0.0364 | 1.0993±0.0411 | 1.1129±0.0117 | 1.1289±0.0148 | 1.0819±0.0535 | 1.1062±0.0531 | 1.0946±0.0192 | 1.1224±0.0029 | 1.0909±0.0091 | 1.0903±0.0345 |
| $5 \times 10^{-6}$ | 1.1039±0.0245 | 1.0986±0.0137 | 1.1386±0.0528 | 1.1225±0.0556 | 1.0472±0.0103 | 1.0801±0.0381 | 1.0740±0.0291 | 1.1074±0.0248 | 1.0940±0.0571 | 1.1170±0.0424 | 1.1369±0.0183 |
| $3 \times 10^{-6}$ | 1.0944±0.0144 | 1.1067±0.0313 | 1.0853±0.0478 | 1.1657±0.0502 | 1.0876±0.0485 | 1.0898±0.0251 | 1.1064±0.0428 | 1.1198±0.0351 | 1.1420±0.0470 | 1.1234±0.0185 | 1.1284±0.0406 |
| $1 \times 10^{-6}$ | 1.0992±0.0299 | 1.0725±0.0256 | 1.1476±0.0510 | 1.1265±0.0194 | 1.1256±0.0139 | 1.1005±0.0335 | 1.0683±0.0558 | 1.0980±0.0285 | 1.1159±0.0161 | 1.0996±0.0567 | 1.0926±0.0170 |

**Table 10:** Numerical results for Gradient Flow with the 8 Gaussians dataset ($d = 50$)

| LR | SW | MaxSW | DSW | MaxKSW | iMSW | viMSW | oMSW | rMSW | EBSW | RPSW | EBRPSW |
|---|---|---|---|---|---|---|---|---|---|---|---|
| 100 | 0.7762±0.1023 | 4.1470±0.0352 | 2.9805±0.0642 | 2.5629±0.4443 | 2.0662±0.1062 | 1.2985±0.1430 | 0.7226±0.0545 | 1.0492±0.0837 | 8.9485±0.8584 | 1.2748±0.0332 | 8.9753±0.7660 |
| 80 | 0.6647±0.0359 | 3.2381±0.0367 | 2.2192±0.2220 | 2.1054±0.0841 | 1.6892±0.1181 | 0.9497±0.0987 | 0.5743±0.0987 | 0.9389±0.0556 | 7.5420±0.9074 | 1.1116±0.0787 | 7.5470±0.5091 |
| 50 | 0.3936±0.0427 | 2.3955±0.5774 | 1.4745±0.1405 | 1.5428±0.0838 | 1.1088±0.0616 | 0.5304±0.0832 | 0.3697±0.0686 | 0.6062±0.0722 | 4.2090±0.0792 | 0.6496±0.0087 | 4.4983±0.4854 |
| 30 | 0.2592±0.0305 | 1.7158±0.0576 | 0.9065±0.0385 | 1.3463±0.0646 | 0.7396±0.0385 | 0.3412±0.0455 | 0.2242±0.0245 | 0.3475±0.0257 | 2.6767±0.0900 | 0.3805±0.0476 | 2.5791±0.0469 |
| 10 | 0.0843±0.0069 | 1.2179±0.0184 | 0.2198±0.0271 | 0.8623±0.0304 | 0.5016±0.0059 | 0.1124±0.0121 | 0.0718±0.0044 | 0.1114±0.0134 | 1.0026±0.0296 | 0.1323±0.0029 | 0.9923±0.0272 |
| 8 | 0.0563±0.0017 | 1.0353±0.0213 | 0.2052±0.0283 | 0.7273±0.0190 | 0.4534±0.0015 | 0.0943±0.0109 | 0.0583±0.0041 | 0.0887±0.0075 | 0.9543±0.0348 | 0.1118±0.0083 | 0.9426±0.0210 |
| 5 | 0.0399±0.0046 | 0.7122±0.0174 | 0.0919±0.0103 | 0.5057±0.0135 | 0.3198±0.0128 | 0.0546±0.0033 | 0.0387±0.0038 | 0.0542±0.0050 | 0.0414±0.0033 | 0.0714±0.0053 | 0.9407±0.0013 |
| 3 | 0.0298±0.0048 | 0.5158±0.0225 | 0.3866±0.0029 | 0.3866±0.0041 | 0.2153±0.0097 | 0.0334±0.0026 | 0.0256±0.0007 | 0.0322±0.0026 | 0.0271±0.0061 | 0.0448±0.0043 | 0.0280±0.0066 |
| 1 | 0.0084±0.0009 | 0.2754±0.0034 | 0.0120±0.0024 | 0.1864±0.0049 | 0.1010±0.0016 | 0.0105±0.0007 | 0.0072±0.0003 | 0.0113±0.0020 | 0.0082±0.0001 | 0.0150±0.0013 | 0.0076±0.0006 |
| $8 \times 10^{-1}$ | 0.0064±0.0006 | 0.2443±0.0031 | 0.0164±0.0065 | 0.1644±0.0012 | 0.0856±0.0006 | 0.0103±0.0005 | 0.0059±0.0009 | 0.0092±0.0001 | 0.0066±0.0007 | 0.0128±0.0010 | 0.0122±0.0084 |
| $5 \times 10^{-1}$ | 0.0039±0.0003 | 0.1865±0.0033 | 0.0061±0.0005 | 0.1229±0.0014 | 0.0555±0.0006 | 0.0058±0.0005 | 0.0044±0.0003 | 0.0056±0.0004 | 0.0042±0.0003 | 0.0075±0.0009 | 0.0040±0.0002 |
| $3 \times 10^{-1}$ | 0.0023±0.0000 | 0.1347±0.0006 | 0.0035±0.0003 | 0.0823±0.0012 | 0.0367±0.0031 | 0.0038±0.0001 | 0.0022±0.0002 | 0.0032±0.0002 | 0.0022±0.0003 | 0.0041±0.0004 | 0.0024±0.0002 |
| $1 \times 10^{-1}$ | 0.0008±0.0001 | 0.0598±0.0005 | 0.0013±0.0000 | 0.0301±0.0001 | 0.0080±0.0001 | 0.0011±0.0002 | 0.0073±0.0092 | 0.0011±0.0000 | 0.0009±0.0001 | 0.0015±0.0000 | 0.0008±0.0000 |
| $8 \times 10^{-2}$ | 0.0006±0.0001 | 0.0479±0.0024 | 0.0011±0.0001 | 0.0232±0.0001 | 0.0059±0.0001 | 0.0010±0.0001 | 0.0006±0.0000 | 0.0009±0.0001 | 0.0007±0.0001 | 0.0011±0.0001 | 0.0007±0.0000 |
| $5 \times 10^{-2}$ | 0.0004±0.0000 | 0.0358±0.0047 | 0.0007±0.0000 | 0.0165±0.0000 | 0.0034±0.0001 | 0.0005±0.0000 | 0.0004±0.0000 | 0.0006±0.0000 | 0.0004±0.0000 | 0.0008±0.0000 | 0.0004±0.0000 |
| $3 \times 10^{-2}$ | 0.0030±0.0040 | 0.0210±0.0051 | 0.0004±0.0000 | 0.0124±0.0000 | 0.0021±0.0001 | 0.0003±0.0000 | 0.0002±0.0000 | 0.0003±0.0000 | 0.0002±0.0000 | 0.0004±0.0000 | 0.0073±0.0100 |
| $1 \times 10^{-2}$ | 0.1316±0.0033 | 0.0501±0.0031 | 0.0302±0.0003 | 0.0598±0.0009 | 0.0906±0.0095 | 0.1090±0.0164 | 0.1320±0.0154 | 0.1244±0.0033 | 0.0472±0.0033 | 0.0001±0.0000 | 0.0546±0.0011 |
| $8 \times 10^{-3}$ | 0.2177±0.0424 | 0.0575±0.0036 | 0.0526±0.0021 | 0.0824±0.0072 | 0.1306±0.0105 | 0.1858±0.0147 | 0.2047±0.0427 | 0.2182±0.0078 | 0.0760±0.0004 | 0.0111±0.0155 | 0.0743±0.0014 |
| $5 \times 10^{-3}$ | 0.3952±0.0247 | 0.1087±0.0024 | 0.1025±0.0047 | 0.1622±0.0081 | 0.3173±0.0275 | 0.3845±0.0282 | 0.4071±0.0187 | 0.4151±0.0329 | 0.1511±0.0098 | 0.0622±0.0067 | 0.1411±0.0066 |
| $3 \times 10^{-3}$ | 0.5586±0.0151 | 0.2257±0.0155 | 0.2471±0.0098 | 0.3464±0.0133 | 0.4916±0.0096 | 0.5395±0.0327 | 0.5485±0.0271 | 0.5645±0.0297 | 0.3053±0.0138 | 0.2302±0.0240 | 0.2800±0.0173 |
| $1 \times 10^{-3}$ | 0.7368±0.0071 | 0.5209±0.0295 | 0.6224±0.0271 | 0.6223±0.0124 | 0.6734±0.0601 | 0.6885±0.0230 | 0.7442±0.0126 | 0.7277±0.0133 | 0.6449±0.0427 | 0.6091±0.0403 | 0.6653±0.0580 |
| $8 \times 10^{-4}$ | 0.7768±0.0225 | 0.6110±0.0293 | 0.6699±0.0478 | 0.6629±0.0105 | 0.7314±0.0226 | 0.7445±0.0266 | 0.7542±0.0457 | 0.7287±0.0117 | 0.6765±0.0522 | 0.6724±0.0188 | 0.6419±0.0300 |
| $5 \times 10^{-4}$ | 0.7490±0.0225 | 0.6869±0.0197 | 0.6954±0.0774 | 0.7295±0.0491 | 0.7891±0.0284 | 0.7690±0.0318 | 0.7695±0.0567 | 0.7668±0.0369 | 0.7451±0.0210 | 0.6965±0.0405 | 0.7216±0.0103 |
| $3 \times 10^{-4}$ | 0.7827±0.0395 | 0.6642±0.0428 | 0.7586±0.0062 | 0.7633±0.0165 | 0.7398±0.0400 | 0.7723±0.0548 | 0.7431±0.0404 | 0.7707±0.0251 | 0.7329±0.0414 | 0.7311±0.0240 | 0.7943±0.0136 |
| $1 \times 10^{-4}$ | 0.8140±0.0313 | 0.7916±0.0414 | 0.7861±0.0208 | 0.7831±0.0390 | 0.8007±0.0424 | 0.8189±0.0443 | 0.7720±0.0456 | 0.8255±0.0249 | 0.7897±0.0402 | 0.7853±0.0086 | 0.7956±0.0433 |
| $8 \times 10^{-5}$ | 0.8251±0.0557 | 0.7742±0.0468 | 0.7564±0.0264 | 0.7817±0.0112 | 0.7730±0.0625 | 0.7962±0.0255 | 0.7882±0.0464 | 0.7486±0.0271 | 0.8021±0.0180 | 0.7926±0.0361 | 0.7754±0.0085 |
| $5 \times 10^{-5}$ | 0.8447±0.0485 | 0.8429±0.0513 | 0.7968±0.0078 | 0.7805±0.0351 | 0.7909±0.0394 | 0.7495±0.0347 | 0.7897±0.0257 | 0.8281±0.0520 | 0.8203±0.0164 | 0.7809±0.0271 | 0.8060±0.0349 |
| $3 \times 10^{-5}$ | 0.8487±0.0216 | 0.8205±0.0185 | 0.8784±0.0763 | 0.8266±0.0327 | 0.7722±0.0272 | 0.7971±0.0129 | 0.8065±0.0415 | 0.7762±0.0201 | 0.7999±0.0260 | 0.7997±0.0335 | 0.8139±0.0303 |
| $1 \times 10^{-5}$ | 0.8185±0.0507 | 0.8229±0.0235 | 0.8475±0.0495 | 0.8096±0.0189 | 0.8023±0.0096 | 0.7732±0.0190 | 0.7889±0.0274 | 0.8114±0.0154 | 0.8616±0.0231 | 0.8170±0.0166 | 0.8138±0.0658 |
| $8 \times 10^{-6}$ | 0.7791±0.0343 | 0.8024±0.0081 | 0.8053±0.0170 | 0.7963±0.0172 | 0.8017±0.0222 | 0.7826±0.0209 | 0.8119±0.0277 | 0.8194±0.0162 | 0.8093±0.0391 | 0.7836±0.0407 | 0.8051±0.0071 |
| $5 \times 10^{-6}$ | 0.8004±0.0436 | 0.8115±0.0065 | 0.7933±0.0083 | 0.8196±0.0535 | 0.8087±0.0554 | 0.8442±0.0464 | 0.8336±0.0498 | 0.7760±0.0264 | 0.7825±0.0188 | 0.8363±0.0375 | 0.8199±0.0169 |
| $3 \times 10^{-6}$ | 0.8095±0.0577 | 0.7820±0.0421 | 0.8062±0.0289 | 0.7960±0.0257 | 0.8325±0.0368 | 0.8312±0.0423 | 0.8435±0.0051 | 0.8024±0.0159 | 0.7826±0.0256 | 0.8178±0.0533 | 0.7688±0.0116 |
| $1 \times 10^{-6}$ | 0.8090±0.0194 | 0.8014±0.0267 | 0.7747±0.0278 | 0.7973±0.0361 | 0.8236±0.0444 | 0.8056±0.0109 | 0.8253±0.0041 | 0.8288±0.0236 | 0.8208±0.0290 | 0.8208±0.0422 | 0.8211±0.0454 |

**Table 11:** Numerical results for Gradient Flow with the Swiss dataset ($d = 50$)

| LR | SW | MaxSW | DSW | MaxKSW | iMSW | viMSW | oMSW | rMSW | EBSW | RPSW | EBRPSW |
|---|---|---|---|---|---|---|---|---|---|---|---|
| 100 | 0.8643±0.0480 | 2.4103±0.0243 | 3.0249±0.0992 | 2.8531±0.3598 | 1.9101±0.0904 | 0.9712±0.0227 | 0.8463±0.0478 | 0.9859±0.0645 | 16.7324±0.3058 | 0.8955±0.0380 | 16.3655±0.8700 |
| 80 | 0.6125±0.0264 | 1.7029±0.0444 | 2.2926±0.1929 | 2.2264±0.1102 | 1.5574±0.1757 | 0.7954±0.0465 | 0.6772±0.0312 | 0.7505±0.0809 | 10.8613±0.1851 | 0.7372±0.0058 | 12.4947±0.8208 |
| 50 | 0.4262±0.0147 | 2.0519±0.0738 | 1.5044±0.0269 | 1.7652±0.2612 | 1.1316±0.0373 | 0.4388±0.0366 | 0.4079±0.0182 | 0.4803±0.0080 | 7.0184±0.8989 | 0.4605±0.0160 | 7.5545±0.5874 |
| 30 | 0.2685±0.0022 | 2.0192±0.1526 | 0.9237±0.0327 | 1.3850±0.1726 | 0.7873±0.0145 | 0.2954±0.0180 | 0.2619±0.0018 | 0.3041±0.0170 | 3.7698±0.2624 | 0.2759±0.0187 | 4.3376±0.2378 |
| 10 | 0.0972±0.0068 | 1.3748±0.0235 | 0.2432±0.0166 | 1.1019±0.0105 | 0.6308±0.0093 | 0.1131±0.0041 | 0.0959±0.0036 | 0.1160±0.0059 | 1.3600±0.0370 | 0.1129±0.0066 | 1.3044±0.0268 |
| 8 | 0.0767±0.0032 | 1.2367±0.0192 | 0.1845±0.0193 | 0.9386±0.0170 | 0.5502±0.0083 | 0.0966±0.0016 | 0.0768±0.0016 | 0.0957±0.0045 | 1.2489±0.0797 | 0.0904±0.0078 | 1.2369±0.0945 |
| 5 | 0.0558±0.0024 | 0.9070±0.0095 | 0.0955±0.0141 | 0.6769±0.0079 | 0.3837±0.0111 | 0.0724±0.0017 | 0.0563±0.0020 | 0.0718±0.0008 | 0.0564±0.0016 | 0.0581±0.0027 | 0.0559±0.0017 |
| 3 | 0.0430±0.0009 | 0.6628±0.0086 | 0.0400±0.0069 | 0.4668±0.0040 | 0.2629±0.0053 | 0.0583±0.0009 | 0.0437±0.0001 | 0.0611±0.0002 | 0.0441±0.0005 | 0.0349±0.0015 | 0.0432±0.0005 |
| 1 | 0.0224±0.0001 | 0.3285±0.0080 | 0.0095±0.0010 | 0.2235±0.0048 | 0.1153±0.0003 | 0.0296±0.0004 | 0.0224±0.0002 | 0.0296±0.0004 | 0.0225±0.0003 | 0.0109±0.0006 | 0.0225±0.0003 |
| $8 \times 10^{-1}$ | 0.0165±0.0001 | 0.2840±0.0054 | 0.0088±0.0004 | 0.1902±0.0019 | 0.1007±0.0003 | 0.0164±0.0001 | 0.0166±0.0002 | 0.0163±0.0002 | 0.0165±0.0001 | 0.0104±0.0004 | 0.0167±0.0002 |
| $5 \times 10^{-1}$ | 0.0080±0.0000 | 0.2212±0.0012 | 0.0057±0.0008 | 0.1405±0.0036 | 0.0740±0.0003 | 0.0046±0.0003 | 0.0080±0.0000 | 0.0047±0.0001 | 0.0080±0.0001 | 0.0067±0.0009 | 0.0080±0.0001 |
| $3 \times 10^{-1}$ | 0.0046±0.0001 | 0.1589±0.0011 | 0.0035±0.0006 | 0.1022±0.0014 | 0.0498±0.0002 | 0.0029±0.0001 | 0.0046±0.0001 | 0.0030±0.0000 | 0.0045±0.0000 | 0.0038±0.0002 | 0.0045±0.0001 |
| $1 \times 10^{-1}$ | 0.0013±0.0002 | 0.0787±0.0024 | 0.0012±0.0001 | 0.0449±0.0002 | 0.0149±0.0001 | 0.0010±0.0001 | 0.0015±0.0000 | 0.0010±0.0000 | 0.0013±0.0002 | 0.0012±0.0000 | 0.0013±0.0002 |
| $8 \times 10^{-2}$ | 0.0011±0.0001 | 0.0650±0.0015 | 0.0010±0.0001 | 0.0410±0.0001 | 0.0117±0.0001 | 0.0008±0.0001 | 0.0012±0.0001 | 0.0008±0.0000 | 0.0012±0.0000 | 0.0010±0.0000 | 0.0012±0.0000 |
| $5 \times 10^{-2}$ | 0.0004±0.0000 | 0.0485±0.0026 | 0.0007±0.0001 | 0.0257±0.0003 | 0.0075±0.0001 | 0.0005±0.0000 | 0.0004±0.0000 | 0.0005±0.0000 | 0.0004±0.0000 | 0.0006±0.0001 | 0.0004±0.0000 |
| $3 \times 10^{-2}$ | 0.0045±0.0060 | 0.0390±0.0057 | 0.0004±0.0000 | 0.0242±0.0062 | 0.0115±0.0104 | 0.0000±0.0000 | 0.0004±0.0000 | 0.0343±0.0043 | 0.0003±0.0000 | 0.0004±0.0000 | 0.0003±0.0000 |
| $1 \times 10^{-2}$ | 0.2804±0.0226 | 0.0720±0.0046 | 0.0502±0.0035 | 0.1019±0.0020 | 0.1841±0.0085 | 0.2933±0.0198 | 0.2789±0.0208 | 0.3458±0.0357 | 0.1314±0.0043 | 0.0346±0.0075 | 0.1413±0.0088 |
| $8 \times 10^{-3}$ | 0.3572±0.0416 | 0.0894±0.0071 | 0.0633±0.0024 | 0.1397±0.0037 | 0.2467±0.0085 | 0.4134±0.0194 | 0.3828±0.0295 | 0.4188±0.0111 | 0.1794±0.0026 | 0.0640±0.0079 | 0.1718±0.0033 |
| $5 \times 10^{-3}$ | 0.7284±0.0214 | 0.1556±0.0114 | 0.1280±0.0096 | 0.2343±0.0145 | 0.4257±0.0203 | 0.5644±0.0408 | 0.5199±0.0729 | 0.5625±0.0292 | 0.2970±0.0092 | 0.1951±0.0367 | 0.3066±0.0145 |
| $3 \times 10^{-3}$ | 0.5855±0.0223 | 0.2609±0.0091 | 0.2891±0.0310 | 0.3839±0.0267 | 0.5528±0.0394 | 0.6505±0.0231 | 0.6237±0.0198 | 0.6863±0.0198 | 0.4115±0.0076 | 0.4189±0.0281 | 0.4130±0.0205 |
| $1 \times 10^{-3}$ | 0.7769±0.0223 | 0.5345±0.0245 | 0.6896±0.0310 | 0.6344±0.0232 | 0.6984±0.0159 | 0.7895±0.0378 | 0.7531±0.0458 | 0.7658±0.0333 | 0.6480±0.0414 | 0.5981±0.0180 | 0.6073±0.0028 |
| $8 \times 10^{-4}$ | 0.7223±0.0554 | 0.5782±0.0436 | 0.6573±0.0227 | 0.6719±0.0370 | 0.6832±0.0032 | 0.8203±0.0277 | 0.7193±0.0335 | 0.7771±0.0393 | 0.6816±0.0188 | 0.6958±0.0269 | 0.6699±0.0186 |
| $5 \times 10^{-4}$ | 0.5284±0.0214 | 0.6497±0.0282 | 0.7132±0.0115 | 0.7223±0.0115 | 0.7746±0.0050 | 0.7589±0.0108 | 0.7435±0.0341 | 0.8034±0.0074 | 0.7169±0.0187 | 0.6618±0.0051 | 0.6806±0.0069 |
| $3 \times 10^{-4}$ | 0.7738±0.0239 | 0.6827±0.0282 | 0.7920±0.0172 | 0.7138±0.0237 | 0.7059±0.0415 | 0.7849±0.0082 | 0.7744±0.0022 | 0.8368±0.0076 | 0.7549±0.0262 | 0.7491±0.0323 | 0.7581±0.0283 |
| $1 \times 10^{-4}$ | 0.7769±0.0223 | 0.7339±0.0433 | 0.8135±0.0696 | 0.7113±0.0250 | 0.7601±0.0404 | 0.8799±0.0589 | 0.7841±0.0240 | 0.8516±0.0175 | 0.7906±0.0125 | 0.7495±0.0050 | 0.7495±0.0843 |
| $8 \times 10^{-5}$ | 0.7693±0.0421 | 0.7755±0.0336 | 0.8598±0.0077 | 0.7652±0.0212 | 0.7680±0.0261 | 0.8133±0.0309 | 0.8206±0.0287 | 0.8416±0.0336 | 0.7758±0.0432 | 0.7305±0.0133 | 0.7520±0.0274 |
| $5 \times 10^{-5}$ | 0.7831±0.0237 | 0.7964±0.0122 | 0.7991±0.0563 | 0.7612±0.0231 | 0.7013±0.0435 | 0.8473±0.0135 | 0.8103±0.0092 | 0.8216±0.0423 | 0.7741±0.0211 | 0.7610±0.0130 | 0.8096±0.0157 |
| $3 \times 10^{-5}$ | 0.7928±0.0350 | 0.7748±0.0484 | 0.8389±0.0115 | 0.7534±0.0318 | 0.7539±0.0087 | 0.8650±0.0217 | 0.7570±0.0352 | 0.8469±0.0187 | 0.7786±0.0070 | 0.7785±0.0239 | 0.7547±0.0152 |
| $1 \times 10^{-5}$ | 0.7588±0.0398 | 0.7469±0.0374 | 0.8472±0.0336 | 0.7930±0.0286 | 0.7672±0.0217 | 0.8318±0.0185 | 0.7799±0.0125 | 0.8329±0.0328 | 0.7835±0.0145 | 0.7804±0.0246 | 0.7885±0.0138 |
| $8 \times 10^{-6}$ | 0.7720±0.0243 | 0.7906±0.0141 | 0.8344±0.0208 | 0.7622±0.0255 | 0.7477±0.0283 | 0.8835±0.0241 | 0.7494±0.0229 | 0.8103±0.0076 | 0.7818±0.0347 | 0.7729±0.0175 | 0.7692±0.0128 |
| $5 \times 10^{-6}$ | 0.7414±0.0346 | 0.7499±0.0495 | 0.8574±0.0395 | 0.8278±0.0333 | 0.7953±0.0533 | 0.8492±0.0526 | 0.7898±0.0170 | 0.8617±0.0261 | 0.7642±0.0455 | 0.7480±0.0184 | 0.7467±0.0161 |
| $3 \times 10^{-6}$ | 0.8039±0.0412 | 0.7761±0.0199 | 0.8475±0.0209 | 0.8175±0.0604 | 0.7411±0.0401 | 0.8466±0.0131 | 0.7385±0.0290 | 0.8266±0.0428 | 0.7709±0.0121 | 0.8049±0.0306 | 0.7930±0.0426 |
| $1 \times 10^{-6}$ | 0.7711±0.0045 | 0.7751±0.0052 | 0.8456±0.0207 | 0.8004±0.0423 | 0.7304±0.0497 | 0.8543±0.0218 | 0.7518±0.0316 | 0.8319±0.0460 | 0.7475±0.0265 | 0.7541±0.0503 | 0.7424±0.0243 |

**Table 12:** Numerical results for Gradient Flow with the Knot dataset ($d = 100$)

| LR | SW | MaxSW | DSW | MaxKSW | iMSW | viMSW | oMSW | rMSW | EBSW | RPSW | EBRPSW |
|---|---|---|---|---|---|---|---|---|---|---|---|
| 100 | 0.8999±0.0210 | 2.7462±0.0473 | 2.8735±0.1213 | 3.2882±0.2012 | 2.0599±0.0665 | 0.9238±0.0107 | 0.8228±0.0550 | 1.0054±0.0460 | 11.1502±0.5452 | 0.9537±0.0065 | 11.5940±0.2757 |
| 80 | 0.6750±0.0370 | 2.2929±0.0425 | 2.3658±0.1707 | 2.8335±0.0114 | 1.6259±0.0782 | 0.7803±0.0317 | 0.6715±0.0205 | 0.8045±0.0388 | 9.5332±0.1474 | 0.7804±0.0251 | 9.4759±0.1591 |
| 50 | 0.3877±0.0209 | 3.0593±0.5077 | 1.5513±0.0461 | 2.5637±0.1616 | 1.1430±0.0018 | 0.4871±0.0066 | 0.4036±0.0349 | 0.4982±0.0160 | 5.9085±0.0718 | 0.5273±0.0293 | 6.1440±0.0923 |
| 30 | 0.2629±0.0154 | 2.4795±0.3492 | 0.9380±0.0379 | 2.3604±0.1002 | 1.3570±0.0652 | 0.2752±0.0008 | 0.2549±0.0194 | 0.2956±0.0055 | 3.7218±0.1138 | 0.3304±0.0195 | 3.6760±0.0457 |
| 10 | 0.0815±0.0022 | 1.6234±0.0147 | 0.2669±0.0078 | 1.2923±0.0119 | 0.8413±0.0116 | 0.0996±0.0031 | 0.0862±0.0043 | 0.0859±0.0026 | 1.8692±0.0718 | 0.1037±0.0045 | 1.9169±0.0229 |
| 8 | 0.0676±0.0019 | 1.5109±0.0179 | 0.1902±0.0150 | 1.1465±0.0356 | 0.6969±0.0184 | 0.0778±0.0033 | 0.0684±0.0045 | 0.0781±0.0031 | 1.8447±0.0816 | 0.0903±0.0086 | 1.8146±0.1152 |
| 5 | 0.0446±0.0021 | 1.1718±0.0410 | 0.0936±0.0274 | 0.8477±0.0221 | 0.4585±0.0021 | 0.0440±0.0019 | 0.0412±0.0005 | 0.0473±0.0021 | 0.0454±0.0038 | 0.0558±0.0023 | 0.0435±0.0026 |
| 3 | 0.0279±0.0011 | 0.9264±0.0262 | 0.0375±0.0020 | 0.6073±0.0108 | 0.2748±0.0066 | 0.0288±0.0008 | 0.0288±0.0013 | 0.0292±0.0011 | 0.0280±0.0014 | 0.0360±0.0014 | 0.0282±0.0009 |
| 1 | 0.0133±0.0003 | 0.4048±0.0096 | 0.0106±0.0007 | 0.2178±0.0017 | 0.0716±0.0009 | 0.0124±0.0006 | 0.0140±0.0002 | 0.0119±0.0005 | 0.0137±0.0010 | 0.0134±0.0002 | 0.0133±0.0004 |
| $8 \times 10^{-1}$ | 0.0120±0.0003 | 0.3242±0.0024 | 0.0091±0.0015 | 0.1738±0.0022 | 0.0590±0.0012 | 0.0104±0.0004 | 0.0125±0.0005 | 0.0105±0.0001 | 0.0124±0.0002 | 0.0100±0.0004 | 0.0128±0.0007 |
| $5 \times 10^{-1}$ | 0.0098±0.0001 | 0.1982±0.0021 | 0.0055±0.0004 | 0.0843±0.0015 | 0.0424±0.0009 | 0.0077±0.0004 | 0.0096±0.0001 | 0.0078±0.0002 | 0.0097±0.0002 | 0.0065±0.0005 | 0.0098±0.0001 |
| $3 \times 10^{-1}$ | 0.0057±0.0000 | 0.0981±0.0066 | 0.0035±0.0001 | 0.0578±0.0012 | 0.0312±0.0008 | 0.0058±0.0001 | 0.0059±0.0000 | 0.0058±0.0000 | 0.0057±0.0001 | 0.0038±0.0001 | 0.0058±0.0001 |
| $1 \times 10^{-1}$ | 0.0016±0.0000 | 0.0439±0.0013 | 0.0013±0.0000 | 0.0290±0.0003 | 0.0159±0.0003 | 0.0010±0.0000 | 0.0015±0.0000 | 0.0009±0.0001 | 0.0015±0.0001 | 0.0013±0.0001 | 0.0016±0.0001 |
| $8 \times 10^{-2}$ | 0.0010±0.0000 | 0.0385±0.0006 | 0.0010±0.0001 | 0.0261±0.0002 | 0.0135±0.0002 | 0.0008±0.0000 | 0.0011±0.0002 | 0.0008±0.0000 | 0.0011±0.0000 | 0.0010±0.0000 | 0.0011±0.0001 |
| $5 \times 10^{-2}$ | 0.0421±0.0445 | 0.0297±0.0008 | 0.0006±0.0001 | 0.0200±0.0001 | 0.0462±0.0330 | 0.0350±0.0487 | 0.0308±0.0428 | 0.0279±0.0351 | 0.0005±0.0000 | 0.0007±0.0000 | 0.0004±0.0000 |
| $3 \times 10^{-2}$ | 0.3033±0.0398 | 0.0219±0.0003 | 0.0004±0.0000 | 0.1169±0.0074 | 0.2579±0.0305 | 0.2547±0.0021 | 0.2924±0.0156 | 0.2703±0.0201 | 0.0600±0.0394 | 0.0004±0.0000 | 0.0803±0.0208 |
| $1 \times 10^{-2}$ | 0.6815±0.0581 | 0.3337±0.0280 | 0.2210±0.0083 | 0.4191±0.0139 | 0.5300±0.0373 | 0.6595±0.0196 | 0.6801±0.0112 | 0.6874±0.0216 | 0.4444±0.0156 | 0.3403±0.0255 | 0.4169±0.0118 |
| $8 \times 10^{-3}$ | 0.7851±0.0479 | 0.3731±0.0236 | 0.3234±0.0314 | 0.4651±0.0083 | 0.6044±0.0072 | 0.6893±0.0299 | 0.7789±0.0269 | 0.7410±0.0204 | 0.4848±0.0111 | 0.4244±0.0203 | 0.5039±0.0169 |
| $5 \times 10^{-3}$ | 0.8508±0.0152 | 0.4733±0.0185 | 0.4979±0.0280 | 0.5620±0.0089 | 0.7247±0.0213 | 0.8281±0.0402 | 0.8774±0.0232 | 0.9089±0.0383 | 0.6009±0.0188 | 0.6378±0.0079 | 0.6152±0.0263 |
| $3 \times 10^{-3}$ | 1.0324±0.0447 | 0.5853±0.0110 | 0.6446±0.0219 | 0.6916±0.0141 | 0.8709±0.0107 | 0.9276±0.0026 | 0.9828±0.0431 | 0.9791±0.0088 | 0.7829±0.0191 | 0.8020±0.0098 | 0.7405±0.0263 |
| $1 \times 10^{-3}$ | 1.0512±0.0197 | 0.8431±0.0326 | 0.8869±0.0561 | 0.9131±0.0176 | 1.0226±0.0193 | 1.0683±0.0350 | 1.0648±0.0060 | 1.0571±0.0341 | 1.0056±0.0210 | 1.0151±0.0527 | 1.0046±0.0041 |
| $8 \times 10^{-4}$ | 1.0779±0.0182 | 0.8908±0.0561 | 0.9720±0.0380 | 0.9676±0.0211 | 1.0130±0.0372 | 1.0758±0.0137 | 1.0793±0.0151 | 1.0885±0.0170 | 1.0231±0.0193 | 1.0303±0.0372 | 1.0333±0.0203 |
| $5 \times 10^{-4}$ | 1.0988±0.0184 | 0.9505±0.0212 | 1.0193±0.0193 | 0.9798±0.0157 | 1.0378±0.0249 | 1.1476±0.0273 | 1.1071±0.0083 | 1.0677±0.0578 | 1.0552±0.0253 | 1.0367±0.0548 | 1.0364±0.0551 |
| $3 \times 10^{-4}$ | 1.1348±0.0345 | 1.0047±0.0080 | 1.0176±0.0270 | 1.0377±0.0328 | 1.0758±0.0090 | 1.1305±0.0379 | 1.1396±0.0675 | 1.1136±0.0423 | 1.0734±0.0079 | 1.0659±0.0355 | 1.0757±0.0183 |
| $1 \times 10^{-4}$ | 1.0925±0.0405 | 1.0953±0.0485 | 1.1185±0.0088 | 1.0751±0.0584 | 1.0750±0.0136 | 1.0832±0.0300 | 1.0511±0.0267 | 1.1315±0.0190 | 1.0838±0.0455 | 1.1096±0.0402 | 1.1157±0.0268 |
| $8 \times 10^{-5}$ | 1.1123±0.0250 | 1.0700±0.0271 | 1.0796±0.0224 | 1.0788±0.0533 | 1.1064±0.0140 | 1.1237±0.0328 | 1.0980±0.0204 | 1.1517±0.0253 | 1.0981±0.0303 | 1.0822±0.0578 | 1.0918±0.0191 |
| $5 \times 10^{-5}$ | 1.1325±0.0183 | 1.0869±0.0071 | 1.1188±0.0054 | 1.1100±0.0342 | 1.1137±0.0182 | 1.0774±0.0188 | 1.1139±0.0363 | 1.1536±0.0548 | 1.1520±0.0330 | 1.0898±0.0257 | 1.0977±0.0389 |
| $3 \times 10^{-5}$ | 1.0957±0.0242 | 1.1209±0.0221 | 1.0884±0.0234 | 1.0532±0.0352 | 1.0918±0.0352 | 1.1116±0.0186 | 1.1636±0.0557 | 1.0753±0.0131 | 1.1512±0.0241 | 1.0965±0.0552 | 1.0595±0.0265 |
| $1 \times 10^{-5}$ | 1.1133±0.0149 | 1.1525±0.0283 | 1.0880±0.0347 | 1.1194±0.0511 | 1.1360±0.0050 | 1.0627±0.0312 | 1.1375±0.0438 | 1.1148±0.0397 | 1.1040±0.0511 | 1.1299±0.0701 | 1.0808±0.0393 |
| $8 \times 10^{-6}$ | 1.1463±0.0507 | 1.1160±0.0364 | 1.1042±0.0119 | 1.1087±0.0458 | 1.1419±0.0305 | 1.0656±0.0384 | 1.1161±0.0382 | 1.1148±0.0381 | 1.1235±0.0733 | 1.1152±0.0240 | 1.1092±0.0640 |
| $5 \times 10^{-6}$ | 1.0819±0.0338 | 1.0975±0.0435 | 1.0985±0.0218 | 1.0971±0.0492 | 1.1398±0.0347 | 1.1297±0.0235 | 1.1038±0.0064 | 1.1181±0.0381 | 1.1171±0.0500 | 1.0907±0.0028 | 1.1375±0.0591 |
| $3 \times 10^{-6}$ | 1.0974±0.0341 | 1.1184±0.0258 | 1.0649±0.0178 | 1.1366±0.0448 | 1.0637±0.0139 | 1.1125±0.0327 | 1.0915±0.0324 | 1.1307±0.0267 | 1.1339±0.0241 | 1.1261±0.0402 | 1.1100±0.0351 |
| $1 \times 10^{-6}$ | 1.1113±0.0431 | 1.0936±0.0302 | 1.1580±0.1150 | 1.0748±0.0342 | 1.1037±0.0867 | 1.0442±0.0314 | 1.0524±0.0236 | 1.0905±0.0209 | 1.1355±0.0364 | 1.1215±0.0309 | 1.0954±0.0074 |

**Table 13:** Numerical results for Gradient Flow with the 8 Gaussians dataset ($d = 100$)

| LR | SW | MaxSW | DSW | MaxKSW | iMSW | viMSW | oMSW | rMSW | EBSW | RPSW | EBRPSW |
|---|---|---|---|---|---|---|---|---|---|---|---|
| 100 | 0.8453±0.0023 | 1.7094±0.0385 | 2.8643±0.2356 | 2.6594±0.1227 | 2.1708±0.0707 | 0.8819±0.0498 | 0.8580±0.0126 | 1.0229±0.0815 | 14.9925±0.9149 | 0.9446±0.0524 | 14.6610±0.9302 |
| 80 | 0.6462±0.0144 | 2.3625±1.6301 | 2.5657±0.0937 | 2.1892±0.1915 | 1.6570±0.0787 | 0.7463±0.0104 | 0.6837±0.0626 | 0.7806±0.0436 | 10.8735±1.6920 | 0.7065±0.0500 | 11.0594±0.9880 |
| 50 | 0.4404±0.0069 | 1.8941±0.0347 | 1.4256±0.0602 | 1.7448±0.0856 | 1.1720±0.0426 | 0.5025±0.0142 | 0.4153±0.0104 | 0.5028±0.0436 | 6.8204±0.2040 | 0.4404±0.0366 | 6.4021±0.4725 |
| 30 | 0.2634±0.0069 | 1.9391±0.0814 | 0.8874±0.0252 | 1.3388±0.0205 | 0.8062±0.0750 | 0.2830±0.0113 | 0.2585±0.0043 | 0.2728±0.0145 | 4.2188±0.4268 | 0.2838±0.0237 | 3.8089±0.3879 |
| 10 | 0.0925±0.0017 | 1.1779±0.1149 | 0.2637±0.0336 | 1.0617±0.0060 | 0.6473±0.0297 | 0.1026±0.0080 | 0.0863±0.0036 | 0.1088±0.0061 | 1.2991±0.0956 | 0.1026±0.0062 | 1.4357±0.1469 |
| 8 | 0.0805±0.0066 | 1.1334±0.0718 | 0.1843±0.0303 | 0.8777±0.0231 | 0.5424±0.0050 | 0.0784±0.0049 | 0.0720±0.0038 | 0.0819±0.0019 | 1.1654±0.0456 | 0.0922±0.0077 | 1.1620±0.0320 |
| 5 | 0.0511±0.0016 | 0.8703±0.0221 | 0.0881±0.0173 | 0.6358±0.0074 | 0.3933±0.0074 | 0.0533±0.0052 | 0.0512±0.0027 | 0.0461±0.0020 | 0.0530±0.0013 | 0.0613±0.0094 | 0.0513±0.0050 |
| 3 | 0.0380±0.0004 | 0.6049±0.0037 | 0.0383±0.0082 | 0.4436±0.0070 | 0.2733±0.0050 | 0.0265±0.0031 | 0.0390±0.0005 | 0.0279±0.0015 | 0.0366±0.0013 | 0.0359±0.0020 | 0.0411±0.0050 |
| 1 | 0.0177±0.0003 | 0.3048±0.0009 | 0.0111±0.0010 | 0.2203±0.0029 | 0.1282±0.0017 | 0.0100±0.0004 | 0.0210±0.0050 | 0.0098±0.0004 | 0.0175±0.0003 | 0.0131±0.0003 | 0.0178±0.0002 |
| $8 \times 10^{-1}$ | 0.0132±0.0002 | 0.2803±0.0002 | 0.0144±0.0094 | 0.2008±0.0039 | 0.1082±0.0018 | 0.0080±0.0005 | 0.0132±0.0001 | 0.0079±0.0002 | 0.0130±0.0003 | 0.0104±0.0008 | 0.0129±0.0001 |
| $5 \times 10^{-1}$ | 0.0070±0.0002 | 0.2261±0.0033 | 0.0052±0.0006 | 0.1506±0.0017 | 0.0731±0.0010 | 0.0046±0.0001 | 0.0071±0.0001 | 0.0049±0.0002 | 0.0071±0.0002 | 0.0062±0.0006 | 0.0070±0.0001 |
| $3 \times 10^{-1}$ | 0.0035±0.0001 | 0.1716±0.0022 | 0.0036±0.0003 | 0.1090±0.0017 | 0.0521±0.0018 | 0.0097±0.0098 | 0.0035±0.0000 | 0.0029±0.0003 | 0.0037±0.0000 | 0.0037±0.0003 | 0.0035±0.0001 |
| $1 \times 10^{-1}$ | 0.0009±0.0000 | 0.0774±0.0003 | 0.0012±0.0000 | 0.0426±0.0005 | 0.0124±0.0001 | 0.0010±0.0000 | 0.0010±0.0000 | 0.0081±0.0101 | 0.0010±0.0000 | 0.0013±0.0001 | 0.0009±0.0001 |
| $8 \times 10^{-2}$ | 0.0007±0.0000 | 0.0675±0.0002 | 0.0007±0.0000 | 0.0357±0.0027 | 0.0093±0.0003 | 0.0008±0.0000 | 0.0007±0.0000 | 0.0008±0.0000 | 0.0007±0.0000 | 0.0010±0.0000 | 0.0007±0.0000 |
| $5 \times 10^{-2}$ | 0.0004±0.0000 | 0.0507±0.0039 | 0.0007±0.0001 | 0.0209±0.0006 | 0.0050±0.0001 | 0.0005±0.0000 | 0.0005±0.0000 | 0.0005±0.0000 | 0.0005±0.0000 | 0.0007±0.0000 | 0.0004±0.0000 |
| $3 \times 10^{-2}$ | 0.0189±0.0099 | 0.0342±0.0022 | 0.0004±0.0000 | 0.0122±0.0000 | 0.0129±0.0081 | 0.0116±0.0092 | 0.0075±0.0102 | 0.0195±0.0063 | 0.0081±0.0111 | 0.0004±0.0000 | 0.0003±0.0000 |
| $1 \times 10^{-2}$ | 0.2747±0.0110 | 0.0596±0.0018 | 0.0432±0.0029 | 0.0831±0.0027 | 0.1521±0.0144 | 0.2559±0.0187 | 0.2458±0.0317 | 0.2713±0.0137 | 0.1024±0.0050 | 0.0238±0.0042 | 0.1028±0.0059 |
| $8 \times 10^{-3}$ | 0.3540±0.0134 | 0.0722±0.0044 | 0.0560±0.0033 | 0.1025±0.0034 | 0.2334±0.0147 | 0.3502±0.0444 | 0.3782±0.0345 | 0.3498±0.0251 | 0.1400±0.0049 | 0.0539±0.0031 | 0.1535±0.0130 |
| $5 \times 10^{-3}$ | 0.5383±0.0144 | 0.1306±0.0044 | 0.1028±0.0079 | 0.1954±0.0088 | 0.4190±0.0172 | 0.4794±0.0464 | 0.5292±0.0509 | 0.5352±0.0263 | 0.2629±0.0107 | 0.1642±0.0141 | 0.2691±0.0017 |
| $3 \times 10^{-3}$ | 0.6450±0.0303 | 0.2361±0.0211 | 0.2410±0.0100 | 0.3577±0.0077 | 0.5810±0.0530 | 0.6048±0.0730 | 0.6123±0.0442 | 0.6584±0.0437 | 0.3916±0.0181 | 0.4047±0.0529 | 0.3907±0.0190 |
| $1 \times 10^{-3}$ | 0.7594±0.0267 | 0.5412±0.0112 | 0.6013±0.0194 | 0.6430±0.0117 | 0.7360±0.0238 | 0.7566±0.0276 | 0.7211±0.0482 | 0.7498±0.0312 | 0.6015±0.0460 | 0.6453±0.0495 | 0.7185±0.0149 |
| $8 \times 10^{-4}$ | 0.7160±0.0087 | 0.5678±0.0262 | 0.6477±0.0109 | 0.6684±0.0222 | 0.7186±0.0174 | 0.7828±0.0445 | 0.7589±0.0165 | 0.7703±0.0604 | 0.6718±0.0382 | 0.6836±0.0211 | 0.7163±0.0128 |
| $5 \times 10^{-4}$ | 0.7653±0.0262 | 0.6971±0.0526 | 0.7226±0.0039 | 0.6803±0.0267 | 0.7617±0.0291 | 0.7912±0.0393 | 0.7891±0.0067 | 0.7953±0.0266 | 0.7830±0.0425 | 0.7378±0.0145 | 0.6751±0.0220 |
| $3 \times 10^{-4}$ | 0.8457±0.0352 | 0.7321±0.0326 | 0.7639±0.0215 | 0.7465±0.0305 | 0.7948±0.0215 | 0.7908±0.0204 | 0.7646±0.0033 | 0.7864±0.0466 | 0.7100±0.0429 | 0.7491±0.0203 | 0.7938±0.0186 |
| $1 \times 10^{-4}$ | 0.8093±0.0227 | 0.7885±0.0195 | 0.7749±0.0105 | 0.7928±0.0380 | 0.8168±0.0158 | 0.8058±0.0158 | 0.8151±0.0481 | 0.8039±0.0251 | 0.7846±0.0221 | 0.7626±0.0589 | 0.8299±0.0233 |
| $8 \times 10^{-5}$ | 0.7991±0.0525 | 0.7841±0.0170 | 0.8303±0.0346 | 0.8011±0.0606 | 0.8028±0.0015 | 0.7600±0.0358 | 0.7743±0.0226 | 0.8303±0.0111 | 0.7847±0.0166 | 0.7977±0.0512 | 0.7738±0.0072 |
| $5 \times 10^{-5}$ | 0.8101±0.0105 | 0.7783±0.0233 | 0.7970±0.0311 | 0.8237±0.0238 | 0.8331±0.0377 | 0.7835±0.0162 | 0.7558±0.0366 | 0.8068±0.0126 | 0.7932±0.0526 | 0.7831±0.0308 | 0.7982±0.0385 |
| $3 \times 10^{-5}$ | 0.8117±0.0263 | 0.7992±0.0453 | 0.8197±0.0457 | 0.7834±0.0271 | 0.7962±0.0431 | 0.8351±0.0116 | 0.7937±0.0126 | 0.7483±0.0244 | 0.7942±0.0143 | 0.8096±0.0368 | 0.8043±0.0252 |
| $1 \times 10^{-5}$ | 0.7595±0.0237 | 0.8169±0.0375 | 0.8115±0.0479 | 0.8115±0.0069 | 0.8231±0.0475 | 0.8176±0.0199 | 0.8142±0.0101 | 0.8302±0.0278 | 0.8018±0.0126 | 0.8152±0.0308 | 0.8371±0.0359 |
| $8 \times 10^{-6}$ | 0.7845±0.0314 | 0.8265±0.0131 | 0.8054±0.0373 | 0.8196±0.0185 | 0.8357±0.0154 | 0.8271±0.0202 | 0.8130±0.0471 | 0.8007±0.0579 | 0.8058±0.0192 | 0.8395±0.0284 | 0.8022±0.0566 |
| $5 \times 10^{-6}$ | 0.8083±0.0300 | 0.8336±0.0266 | 0.8146±0.0384 | 0.8508±0.0368 | 0.7954±0.0439 | 0.8371±0.0248 | 0.8227±0.0695 | 0.8602±0.0203 | 0.8230±0.0152 | 0.8052±0.0382 | 0.8197±0.0247 |
| $3 \times 10^{-6}$ | 0.7842±0.0003 | 0.7975±0.0232 | 0.8045±0.0418 | 0.7585±0.0186 | 0.8474±0.0376 | 0.8087±0.0283 | 0.8302±0.0095 | 0.8099±0.0333 | 0.7705±0.0159 | 0.7857±0.0354 | 0.8152±0.0362 |
| $1 \times 10^{-6}$ | 0.8476±0.0189 | 0.8257±0.0266 | 0.7709±0.0330 | 0.8412±0.0244 | 0.7657±0.0221 | 0.7549±0.0264 | 0.8248±0.0183 | 0.8425±0.0392 | 0.8022±0.0302 | 0.8207±0.0338 | 0.7887±0.0366 |

**Table 14:** Numerical results for Gradient Flow with the Swiss dataset ($d = 100$)

| LR | SW | MaxSW | DSW | MaxKSW | iMSW | viMSW | oMSW | rMSW | EBSW | RPSW | EBRPSW |
|---|---|---|---|---|---|---|---|---|---|---|---|
| 100 | 294.00 ± 0.38 | 287.00 ± 4.65 | 227.00 ± 4.40 | 275.00 ± 7.53 | 286.00 ± 7.55 | 286.00 ± 7.36 | 294.00 ± 0.38 | 201.00 ± 3.34 | 291.00 ± 0.71 | 291.00 ± 0.79 | 291.00 ± 0.52 |
| 80 | 294.00 ± 0.38 | 286.00 ± 4.00 | 227.00 ± 4.49 | 275.00 ± 7.71 | 286.00 ± 7.55 | 286.00 ± 7.36 | 294.00 ± 0.38 | 168.00 ± 4.27 | 289.00 ± 0.56 | 289.00 ± 0.54 | 290.00 ± 0.47 |
| 50 | 294.00 ± 0.38 | 280.00 ± 4.74 | 228.00 ± 5.45 | 276.00 ± 0.85 | 286.00 ± 7.55 | 286.00 ± 7.36 | 294.00 ± 0.38 | 147.00 ± 0.45 | 282.00 ± 0.86 | 282.00 ± 1.65 | 284.00 ± 0.81 |
| 30 | 294.00 ± 0.38 | 263.00 ± 6.38 | 232.00 ± 11.00 | 258.00 ± 1.30 | 286.00 ± 7.50 | 286.00 ± 7.36 | 294.00 ± 0.38 | 162.00 ± 7.25 | 232.00 ± 35.60 | 272.00 ± 8.80 | 268.00 ± 0.79 |
| 10 | 294.00 ± 0.38 | 187.00 ± 15.70 | 179.00 ± 1.86 | 173.00 ± 1.69 | 286.00 ± 7.50 | 286.00 ± 7.36 | 145.00 ± 1.13 | 95.90 ± 0.61 | 161.00 ± 3.78 | 191.00 ± 1.47 | 163.00 ± 1.10 |
| 8 | 294.00 ± 0.38 | 166.00 ± 17.30 | 168.00 ± 3.83 | 151.00 ± 3.05 | 286.00 ± 7.50 | 286.00 ± 7.36 | 122.00 ± 0.20 | 89.70 ± 0.32 | 152.00 ± 0.88 | 169.00 ± 4.85 | 181.00 ± 1.50 |
| 5 | 165.00 ± 5.93 | 123.00 ± 17.90 | 134.00 ± 10.20 | 101.00 ± 4.33 | 286.00 ± 7.50 | 286.00 ± 7.36 | 53.40 ± 0.29 | 50.90 ± 0.31 | 143.00 ± 5.02 | 123.00 ± 4.24 | 133.00 ± 5.45 |
| 3 | 48.40 ± 0.01 | 85.60 ± 15.60 | 88.40 ± 8.86 | 71.40 ± 1.34 | 293.90 ± 0.40 | 293.90 ± 0.40 | 29.60 ± 0.01 | 34.60 ± 1.36 | 85.50 ± 3.18 | 43.30 ± 2.77 | 80.20 ± 6.35 |
| 1 | 0.03 ± 0.02 | 28.60 ± 4.04 | 27.70 ± 3.07 | 22.40 ± 2.20 | 293.90 ± 0.40 | 293.90 ± 0.40 | 11.60 ± 0.01 | 10.10 ± 1.78 | 29.20 ± 1.93 | 23.90 ± 8.57 | 30.00 ± 2.60 |
| $8 \times 10^{-1}$ | 0.04 ± 0.04 | 23.70 ± 7.05 | 25.50 ± 6.84 | 16.60 ± 0.06 | 293.90 ± 0.40 | 293.90 ± 0.40 | 8.59 ± 0.43 | 7.33 ± 0.25 | 27.50 ± 0.30 | 23.70 ± 4.18 | 25.90 ± 0.43 |
| $5 \times 10^{-1}$ | 0.04 ± 0.04 | 13.90 ± 3.30 | 11.50 ± 4.65 | 11.30 ± 0.70 | 293.90 ± 0.40 | 293.90 ± 0.40 | 4.99 ± 0.01 | 3.86 ± 0.16 | 14.00 ± 0.19 | 7.90 ± 2.08 | 15.40 ± 0.36 |
| $3 \times 10^{-1}$ | 0.09 ± 0.07 | 10.30 ± 3.68 | 9.24 ± 0.24 | 6.73 ± 0.14 | 293.90 ± 0.40 | 293.90 ± 0.40 | 2.97 ± 0.00 | 2.50 ± 0.55 | 10.20 ± 0.38 | 5.19 ± 2.25 | 11.20 ± 0.88 |
| $1 \times 10^{-1}$ | 0.03 ± 0.01 | 5.55 ± 1.22 | 2.98 ± 0.39 | 4.33 ± 0.01 | 293.90 ± 0.40 | 184.00 ± 11.00 | 0.97 ± 0.00 | 0.89 ± 0.01 | 4.99 ± 0.22 | 1.69 ± 0.49 | 5.55 ± 0.00 |
| $8 \times 10^{-2}$ | 0.04 ± 0.00 | 5.30 ± 1.26 | 2.90 ± 0.09 | 3.86 ± 0.18 | 293.90 ± 0.40 | 5.07 ± 0.03 | 0.78 ± 0.00 | 0.60 ± 0.02 | 4.17 ± 0.00 | 1.32 ± 0.41 | 4.89 ± 0.01 |
| $5 \times 10^{-2}$ | 0.03 ± 0.01 | 4.85 ± 1.70 | 1.91 ± 0.04 | 3.13 ± 0.01 | 293.90 ± 0.40 | 0.31 ± 0.01 | 0.49 ± 0.00 | 0.55 ± 0.17 | 3.81 ± 0.03 | 0.87 ± 0.28 | 3.85 ± 0.03 |
| $3 \times 10^{-2}$ | 0.06 ± 0.04 | 3.68 ± 1.01 | 1.18 ± 0.06 | 2.74 ± 0.08 | 5.59 ± 0.00 | 0.23 ± 0.00 | 0.29 ± 0.00 | 0.22 ± 0.00 | 3.30 ± 0.00 | 0.56 ± 0.15 | 3.47 ± 0.06 |
| $1 \times 10^{-2}$ | 1.18 ± 0.08 | 2.48 ± 0.48 | 0.25 ± 0.07 | 2.00 ± 0.01 | 0.35 ± 0.00 | 0.13 ± 0.02 | 0.10 ± 0.00 | 0.07 ± 0.00 | 2.45 ± 0.04 | 0.21 ± 0.04 | 2.55 ± 0.01 |
| $8 \times 10^{-3}$ | 1.36 ± 0.10 | 2.26 ± 0.47 | 0.17 ± 0.04 | 1.81 ± 0.02 | 0.25 ± 0.00 | 0.06 ± 0.00 | 0.07 ± 0.00 | 0.06 ± 0.00 | 2.34 ± 0.02 | 0.19 ± 0.03 | 2.41 ± 0.02 |
| $5 \times 10^{-3}$ | 1.73 ± 0.08 | 1.70 ± 0.42 | 0.17 ± 0.00 | 1.30 ± 0.01 | 0.13 ± 0.03 | 0.03 ± 0.00 | 0.05 ± 0.00 | 0.04 ± 0.00 | 2.00 ± 0.03 | 0.13 ± 0.02 | 2.03 ± 0.02 |
| $3 \times 10^{-3}$ | 2.12 ± 0.05 | 1.14 ± 0.40 | 0.11 ± 0.01 | 0.76 ± 0.01 | 0.09 ± 0.00 | 0.03 ± 0.00 | 0.04 ± 0.00 | 0.03 ± 0.00 | 1.48 ± 0.02 | 0.10 ± 0.00 | 1.50 ± 0.01 |
| $1 \times 10^{-3}$ | 3.05 ± 0.03 | 0.15 ± 0.10 | 0.03 ± 0.00 | 0.09 ± 0.05 | 0.03 ± 0.00 | 0.03 ± 0.00 | 0.03 ± 0.00 | 0.03 ± 0.00 | 0.03 ± 0.00 | 0.13 ± 0.02 | 0.10 ± 0.07 |
| $8 \times 10^{-4}$ | 3.34 ± 0.02 | 0.18 ± 0.14 | 0.03 ± 0.00 | 0.04 ± 0.00 | 0.03 ± 0.00 | 0.03 ± 0.00 | 0.06 ± 0.03 | 0.03 ± 0.00 | 0.03 ± 0.00 | 0.13 ± 0.00 | 0.04 ± 0.01 |
| $5 \times 10^{-4}$ | 4.25 ± 0.00 | 0.09 ± 0.06 | 0.03 ± 0.00 | 0.03 ± 0.00 | 0.03 ± 0.00 | 0.03 ± 0.00 | 0.05 ± 0.03 | 0.13 ± 0.10 | 0.07 ± 0.01 | 0.14 ± 0.05 | 0.03 ± 0.01 |
| $3 \times 10^{-4}$ | 6.02 ± 0.04 | 0.03 ± 0.00 | 0.05 ± 0.02 | 0.03 ± 0.00 | 0.03 ± 0.00 | 0.03 ± 0.00 | 0.06 ± 0.03 | 0.03 ± 0.00 | 0.05 ± 0.03 | 0.16 ± 0.01 | 0.08 ± 0.03 |
| $1 \times 10^{-4}$ | 13.91 ± 0.32 | 6.76 ± 4.76 | 0.84 ± 0.07 | 1.19 ± 0.20 | 0.03 ± 0.00 | 0.03 ± 0.00 | 1.38 ± 0.11 | 1.42 ± 0.08 | 1.32 ± 0.11 | 5.20 ± 0.05 | 1.57 ± 0.02 |

**Table 15:** Results for Color Transfer (Set 1).

| LR | SW | MaxSW | DSW | MaxKSW | iMSW | viMSW | oMSW | rMSW | EBSW | RPSW | EBRPSW |
|---|---|---|---|---|---|---|---|---|---|---|---|
| 100 | 271.54 ± 0.22 | 255.40 ± 2.41 | 268.04 ± 0.71 | 266.08 ± 0.17 | 250.17 ± 2.50 | 250.21 ± 2.50 | 269.68 ± 2.46 | 184.92 ± 0.00 | 269.68 ± 2.09 | 267.88 ± 0.27 | 267.89 ± 0.27 |
| 80 | 271.54 ± 0.22 | 267.88 ± 0.27 | 268.93 ± 0.93 | 265.72 ± 0.30 | 250.17 ± 2.50 | 250.21 ± 2.50 | 269.68 ± 2.46 | 183.33 ± 0.08 | 271.54 ± 0.22 | 267.84 ± 0.03 | 267.86 ± 0.03 |
| 50 | 271.54 ± 0.22 | 267.67 ± 0.25 | 270.75 ± 0.57 | 264.57 ± 0.25 | 250.17 ± 2.50 | 250.21 ± 2.50 | 269.68 ± 2.46 | 181.88 ± 0.37 | 271.54 ± 0.22 | 267.48 ± 0.22 | 267.66 ± 0.25 |
| 30 | 206.09 ± 0.32 | 266.45 ± 0.25 | 271.54 ± 0.22 | 263.23 ± 0.32 | 250.21 ± 2.46 | 250.21 ± 2.46 | 269.33 ± 1.63 | 150.88 ± 0.68 | 266.07 ± 0.27 | 265.47 ± 0.10 | 266.50 ± 0.30 |
| 10 | 267.89 ± 0.27 | 231.34 ± 0.18 | 237.76 ± 8.30 | 207.42 ± 14.88 | 250.45 ± 2.28 | 250.45 ± 2.28 | 145.92 ± 0.14 | 110.15 ± 0.05 | 258.87 ± 0.32 | 209.55 ± 8.08 | 188.90 ± 1.52 |
| 8 | 267.89 ± 0.27 | 202.37 ± 0.25 | 208.99 ± 0.19 | 170.61 ± 3.62 | 267.89 ± 0.27 | 267.89 ± 0.27 | 122.14 ± 0.03 | 70.60 ± 0.03 | 241.47 ± 0.33 | 197.62 ± 2.41 | 189.23 ± 7.49 |
| 5 | 267.89 ± 0.27 | 155.92 ± 7.23 | 160.85 ± 2.04 | 104.98 ± 7.47 | 250.96 ± 2.44 | 250.78 ± 2.44 | 63.83 ± 0.05 | 52.42 ± 0.06 | 145.10 ± 23.12 | 109.62 ± 6.34 | 138.06 ± 4.07 |
| 3 | 67.92 ± 3.09 | 81.71 ± 3.84 | 82.30 ± 16.46 | 62.80 ± 2.47 | 249.98 ± 2.76 | 249.94 ± 0.03 | 29.07 ± 0.02 | 26.25 ± 0.45 | 93.20 ± 1.46 | 66.69 ± 3.35 | 89.42 ± 2.56 |
| 1 | 4.82 ± 0.05 | 39.43 ± 1.40 | 17.67 ± 9.12 | 21.70 ± 0.29 | 259.34 ± 0.43 | 259.94 ± 0.03 | 9.43 ± 0.19 | 10.18 ± 1.32 | 30.18 ± 1.27 | 22.54 ± 3.17 | 34.46 ± 2.11 |
| $8 \times 10^{-1}$ | 0.01 ± 0.00 | 31.62 ± 2.06 | 17.77 ± 4.39 | 18.39 ± 0.09 | 267.62 ± 0.01 | 267.62 ± 0.01 | 9.74 ± 0.08 | 7.77 ± 1.12 | 26.50 ± 1.50 | 18.69 ± 0.09 | 25.70 ± 1.05 |
| $5 \times 10^{-1}$ | 0.01 ± 0.00 | 19.61 ± 0.76 | 11.12 ± 0.13 | 15.37 ± 1.67 | 267.89 ± 0.27 | 267.89 ± 0.27 | 5.38 ± 0.33 | 4.35 ± 0.32 | 17.35 ± 0.83 | 12.48 ± 0.86 | 17.65 ± 0.39 |
| $3 \times 10^{-1}$ | 0.03 ± 0.01 | 12.00 ± 0.85 | 1.31 ± 0.01 | 8.26 ± 0.09 | 267.89 ± 0.27 | 267.89 ± 0.27 | 3.31 ± 0.00 | 2.58 ± 0.29 | 10.92 ± 0.30 | 7.50 ± 0.91 | 11.34 ± 0.26 |
| $1 \times 10^{-1}$ | 0.03 ± 0.00 | 6.39 ± 0.52 | 3.40 ± 0.21 | 5.72 ± 0.00 | 267.89 ± 0.27 | 267.89 ± 0.27 | 1.06 ± 0.01 | 0.81 ± 0.04 | 6.36 ± 0.04 | 2.54 ± 0.29 | 7.46 ± 0.09 |
| $8 \times 10^{-2}$ | 0.03 ± 0.00 | 6.62 ± 0.65 | 2.65 ± 0.13 | 3.82 ± 0.02 | 267.89 ± 0.27 | 267.89 ± 0.27 | 0.95 ± 0.01 | 0.70 ± 0.01 | 5.10 ± 0.47 | 1.97 ± 0.23 | 6.87 ± 0.09 |
| $5 \times 10^{-2}$ | 0.33 ± 0.27 | 5.60 ± 0.38 | 2.12 ± 0.10 | 3.28 ± 0.08 | 250.80 ± 2.77 | 251.21 ± 2.88 | 0.59 ± 0.05 | 0.35 ± 0.00 | 5.81 ± 0.00 | 1.28 ± 0.15 | 6.03 ± 0.05 |
| $3 \times 10^{-2}$ | 0.63 ± 0.57 | 4.60 ± 0.46 | 1.18 ± 0.00 | 2.74 ± 0.00 | 3.36 ± 1.76 | 3.19 ± 1.57 | 0.30 ± 0.00 | 0.22 ± 0.01 | 5.41 ± 1.05 | 0.76 ± 0.10 | 5.51 ± 1.02 |
| $1 \times 10^{-2}$ | 2.20 ± 0.51 | 2.71 ± 0.12 | 0.25 ± 0.00 | 2.00 ± 0.00 | 0.34 ± 0.01 | 0.33 ± 0.01 | 0.10 ± 0.00 | 0.08 ± 0.01 | 4.22 ± 0.04 | 0.22 ± 0.02 | 4.25 ± 0.02 |
| $8 \times 10^{-3}$ | 2.57 ± 0.61 | 2.31 ± 0.02 | 0.15 ± 0.01 | 1.83 ± 0.01 | 0.24 ± 0.00 | 0.24 ± 0.00 | 0.08 ± 0.00 | 0.07 ± 0.01 | 3.96 ± 0.01 | 0.21 ± 0.02 | 4.00 ± 0.02 |
| $5 \times 10^{-3}$ | 3.00 ± 0.63 | 1.57 ± 0.06 | 0.10 ± 0.03 | 1.32 ± 0.01 | 0.13 ± 0.00 | 0.13 ± 0.00 | 0.05 ± 0.00 | 0.04 ± 0.00 | 3.28 ± 0.64 | 0.17 ± 0.02 | 3.30 ± 0.64 |
| $3 \times 10^{-3}$ | 4.07 ± 0.98 | 0.92 ± 0.11 | 0.11 ± 0.00 | 0.76 ± 0.00 | 0.09 ± 0.00 | 0.07 ± 0.01 | 0.04 ± 0.00 | 0.03 ± 0.00 | 1.49 ± 0.01 | 0.10 ± 0.00 | 1.50 ± 0.00 |
| $1 \times 10^{-3}$ | 8.48 ± 2.72 | 0.15 ± 0.00 | 0.04 ± 0.01 | 0.10 ± 0.06 | 0.03 ± 0.00 | 0.03 ± 0.00 | 0.03 ± 0.00 | 0.03 ± 0.00 | 0.03 ± 0.00 | 0.13 ± 0.00 | 0.07 ± 0.02 |
| $8 \times 10^{-4}$ | 9.10 ± 2.88 | 0.12 ± 0.03 | 0.04 ± 0.00 | 0.04 ± 0.00 | 0.03 ± 0.00 | 0.03 ± 0.00 | 0.06 ± 0.00 | 0.03 ± 0.00 | 0.03 ± 0.00 | 0.16 ± 0.02 | 0.04 ± 0.00 |
| $5 \times 10^{-4}$ | 11.10 ± 3.42 | 0.06 ± 0.01 | 0.04 ± 0.01 | 0.03 ± 0.00 | 0.03 ± 0.00 | 0.03 ± 0.00 | 0.05 ± 0.00 | 0.07 ± 0.03 | 0.04 ± 0.02 | 0.15 ± 0.00 | 0.03 ± 0.00 |
| $3 \times 10^{-4}$ | 13.97 ± 3.97 | 0.03 ± 0.00 | 0.04 ± 0.00 | 0.03 ± 0.00 | 0.03 ± 0.00 | 0.03 ± 0.00 | 0.05 ± 0.01 | 0.03 ± 0.00 | 0.04 ± 0.01 | 0.15 ± 0.00 | 0.06 ± 0.01 |
| $1 \times 10^{-4}$ | 11.85 ± 0.63 | 4.04 ± 1.36 | 0.86 ± 0.01 | 1.25 ± 0.03 | 0.03 ± 0.00 | 0.03 ± 0.00 | 1.15 ± 0.12 | 1.16 ± 0.13 | 1.32 ± 0.00 | 4.59 ± 0.31 | 1.49 ± 0.04 |

**Table 16:** Results for Color Transfer (Set 2).

| LR | SW | MaxSW | DSW | MaxKSW | iMSW | viMSW | oMSW | rMSW | EBSW | RPSW | EBRPSW |
|---|---|---|---|---|---|---|---|---|---|---|---|
| 100 | 320.55 ± 0.09 | 320.55 ± 0.09 | 279.16 ± 0.05 | 308.63 ± 0.18 | 320.55 ± 0.09 | 320.55 ± 0.09 | 320.55 ± 0.09 | 157.93 ± 0.04 | 320.55 ± 0.09 | 320.55 ± 0.09 | 320.55 ± 0.09 |
| 80 | 320.55 ± 0.09 | 320.55 ± 0.09 | 279.16 ± 0.05 | 307.70 ± 0.20 | 320.55 ± 0.09 | 320.55 ± 0.09 | 320.55 ± 0.09 | 168.12 ± 1.25 | 320.55 ± 0.09 | 320.55 ± 0.09 | 320.55 ± 0.09 |
| 50 | 320.55 ± 0.09 | 320.55 ± 0.09 | 279.16 ± 0.05 | 304.23 ± 0.21 | 320.55 ± 0.09 | 320.55 ± 0.09 | 320.55 ± 0.09 | 124.61 ± 0.79 | 320.55 ± 0.09 | 320.55 ± 0.09 | 320.55 ± 0.09 |
| 30 | 320.55 ± 0.09 | 320.55 ± 0.09 | 279.16 ± 0.05 | 298.63 ± 0.38 | 320.55 ± 0.09 | 320.55 ± 0.09 | 320.45 ± 0.11 | 123.85 ± 10.63 | 149.82 ± 0.07 | 320.52 ± 0.06 | 320.57 ± 0.05 |
| 10 | 320.55 ± 0.09 | 286.84 ± 0.29 | 279.14 ± 0.03 | 202.30 ± 1.04 | 320.17 ± 0.03 | 320.16 ± 0.03 | 183.41 ± 0.22 | 62.75 ± 0.48 | 157.47 ± 28.22 | 266.66 ± 2.21 | 205.07 ± 78.24 |
| 8 | 320.52 ± 0.07 | 251.89 ± 0.26 | 210.92 ± 6.75 | 165.78 ± 0.11 | 311.04 ± 0.20 | 311.02 ± 0.19 | 143.62 ± 0.14 | 91.55 ± 0.01 | 171.06 ± 18.71 | 222.87 ± 15.73 | 165.19 ± 37.58 |
| 5 | 254.50 ± 0.29 | 175.13 ± 4.67 | 161.67 ± 0.60 | 95.66 ± 15.39 | 219.26 ± 19.68 | 232.89 ± 0.29 | 72.48 ± 0.04 | 30.00 ± 1.59 | 123.75 ± 58.87 | 111.06 ± 3.36 | 124.90 ± 7.31 |
| 3 | 59.33 ± 0.07 | 108.52 ± 1.52 | 107.15 ± 8.57 | 57.36 ± 3.19 | 320.55 ± 0.09 | 320.55 ± 0.09 | 34.35 ± 0.22 | 29.95 ± 0.29 | 87.52 ± 10.23 | 66.06 ± 1.43 | 107.12 ± 1.10 |
| 1 | 0.00 ± 0.00 | 35.58 ± 1.42 | 39.61 ± 2.14 | 21.60 ± 0.55 | 320.55 ± 0.09 | 320.55 ± 0.09 | 9.70 ± 0.01 | 10.02 ± 0.49 | 34.00 ± 0.45 | 19.71 ± 3.98 | 32.68 ± 4.24 |
| $8 \times 10^{-1}$ | 0.00 ± 0.00 | 28.51 ± 1.06 | 17.42 ± 0.97 | 17.33 ± 0.41 | 320.55 ± 0.09 | 320.55 ± 0.09 | 7.68 ± 0.01 | 7.67 ± 0.39 | 27.35 ± 1.00 | 18.13 ± 2.59 | 27.86 ± 1.00 |
| $5 \times 10^{-1}$ | 0.00 ± 0.00 | 19.69 ± 1.77 | 12.75 ± 4.32 | 11.51 ± 0.11 | 320.55 ± 0.09 | 320.55 ± 0.09 | 4.80 ± 0.01 | 5.08 ± 0.89 | 16.78 ± 0.14 | 10.86 ± 0.59 | 18.69 ± 1.52 |
| $3 \times 10^{-1}$ | 0.20 ± 0.25 | 12.93 ± 1.49 | 7.88 ± 1.00 | 8.03 ± 0.05 | 320.55 ± 0.09 | 320.55 ± 0.09 | 3.38 ± 0.01 | 2.75 ± 0.22 | 11.10 ± 0.32 | 6.10 ± 0.27 | 11.51 ± 0.58 |
| $1 \times 10^{-1}$ | 0.03 ± 0.00 | 6.24 ± 0.71 | 3.97 ± 0.02 | 5.10 ± 0.24 | 320.15 ± 0.03 | 320.16 ± 0.03 | 0.95 ± 0.00 | 0.79 ± 0.01 | 6.17 ± 0.16 | 1.94 ± 0.23 | 6.96 ± 0.52 |
| $8 \times 10^{-2}$ | 0.20 ± 0.24 | 5.93 ± 0.44 | 2.75 ± 0.40 | 4.72 ± 0.06 | 311.04 ± 0.20 | 311.02 ± 0.19 | 0.77 ± 0.00 | 0.73 ± 0.01 | 5.51 ± 0.21 | 1.59 ± 0.29 | 6.35 ± 0.41 |
| $5 \times 10^{-2}$ | 0.17 ± 0.24 | 4.33 ± 0.74 | 2.02 ± 0.07 | 4.47 ± 0.08 | 219.26 ± 19.68 | 232.89 ± 0.29 | 0.48 ± 0.00 | 0.49 ± 0.06 | 5.03 ± 0.16 | 0.97 ± 0.14 | 5.30 ± 0.10 |
| $3 \times 10^{-2}$ | 0.95 ± 0.01 | 3.68 ± 0.00 | 1.17 ± 0.01 | 4.06 ± 0.08 | 5.99 ± 0.56 | 5.49 ± 0.59 | 0.29 ± 0.00 | 0.24 ± 0.02 | 4.56 ± 0.06 | 0.58 ± 0.05 | 4.78 ± 0.08 |
| $1 \times 10^{-2}$ | 2.16 ± 0.04 | 2.31 ± 0.12 | 0.28 ± 0.05 | 2.93 ± 0.01 | 0.34 ± 0.01 | 0.32 ± 0.01 | 0.10 ± 0.00 | 0.07 ± 0.00 | 3.51 ± 0.05 | 0.28 ± 0.08 | 3.59 ± 0.01 |
| $8 \times 10^{-3}$ | 2.37 ± 0.07 | 2.09 ± 0.12 | 0.22 ± 0.07 | 2.64 ± 0.01 | 0.24 ± 0.01 | 0.23 ± 0.00 | 0.09 ± 0.02 | 0.07 ± 0.01 | 3.32 ± 0.02 | 0.23 ± 0.06 | 3.42 ± 0.01 |
| $5 \times 10^{-3}$ | 2.99 ± 0.02 | 1.61 ± 0.06 | 0.18 ± 0.01 | 1.51 ± 0.01 | 0.13 ± 0.00 | 0.13 ± 0.00 | 0.05 ± 0.00 | 0.04 ± 0.00 | 2.41 ± 0.02 | 0.16 ± 0.05 | 2.44 ± 0.01 |
| $3 \times 10^{-3}$ | 3.24 ± 0.03 | 1.15 ± 0.01 | 0.12 ± 0.01 | 0.93 ± 0.12 | 0.09 ± 0.00 | 0.06 ± 0.00 | 0.04 ± 0.00 | 0.04 ± 0.01 | 1.76 ± 0.01 | 0.10 ± 0.00 | 1.79 ± 0.01 |
| $1 \times 10^{-3}$ | 3.48 ± 0.17 | 0.15 ± 0.00 | 0.05 ± 0.01 | 0.66 ± 0.37 | 0.03 ± 0.00 | 0.10 ± 0.09 | 0.03 ± 0.00 | 0.03 ± 0.00 | 0.03 ± 0.00 | 0.15 ± 0.01 | 0.11 ± 0.11 |
| $8 \times 10^{-4}$ | 3.84 ± 0.35 | 0.12 ± 0.05 | 0.03 ± 0.00 | 0.04 ± 0.00 | 0.08 ± 0.08 | 0.03 ± 0.00 | 0.03 ± 0.00 | 0.03 ± 0.00 | 0.05 ± 0.02 | 0.09 ± 0.05 | 0.05 ± 0.03 |
| $5 \times 10^{-4}$ | 4.93 ± 0.48 | 0.06 ± 0.02 | 0.03 ± 0.00 | 0.03 ± 0.00 | 0.03 ± 0.00 | 0.03 ± 0.00 | 0.05 ± 0.00 | 0.08 ± 0.07 | 0.03 ± 0.00 | 0.11 ± 0.02 | 0.03 ± 0.00 |
| $3 \times 10^{-4}$ | 6.37 ± 0.25 | 0.03 ± 0.00 | 0.08 ± 0.02 | 0.03 ± 0.00 | 0.03 ± 0.00 | 0.03 ± 0.00 | 0.05 ± 0.01 | 0.03 ± 0.00 | 0.07 ± 0.02 | 0.15 ± 0.01 | 0.05 ± 0.02 |
| $1 \times 10^{-4}$ | 13.61 ± 0.21 | 5.55 ± 0.86 | 0.89 ± 0.03 | 6.88 ± 0.15 | 0.70 ± 0.09 | 0.74 ± 0.14 | 11.42 ± 0.11 | 12.17 ± 0.14 | 3.64 ± 0.15 | 7.50 ± 0.16 | 3.68 ± 0.01 |

Table 17: Results for Color Transfer (Set 3).

| LR | SW | MaxSW | DSW | MaxKSW | iMSW | viMSW | oMSW | rMSW | EBSW | RPSW | EBRPSW |
|---|---|---|---|---|---|---|---|---|---|---|---|
| $1$ | $14.17_{\pm0.02}$ | - | - | - | - | - | - | - | - | - | - |
| $8 \times 10^{-1}$ | $14.16_{\pm0.01}$ | - | - | - | - | - | - | - | - | - | - |
| $5 \times 10^{-1}$ | $14.15_{\pm0.02}$ | - | - | - | - | - | - | - | - | - | - |
| $3 \times 10^{-1}$ | $14.16_{\pm0.00}$ | - | - | - | - | - | - | - | - | - | - |
| $1 \times 10^{-1}$ | $14.22_{\pm0.01}$ | - | - | $14.50_{\pm0.03}$ | - | - | $14.13_{\pm0.02}$ | $14.17_{\pm0.02}$ | - | $17.78_{\pm0.18}$ | - |
| $8 \times 10^{-2}$ | $14.26_{\pm0.01}$ | - | $14.78_{\pm0.02}$ | $14.29_{\pm0.01}$ | - | - | $14.12_{\pm0.01}$ | $14.16_{\pm0.01}$ | - | $14.14_{\pm0.00}$ | - |
| $5 \times 10^{-2}$ | $14.33_{\pm0.01}$ | $14.54_{\pm0.02}$ | $14.46_{\pm0.03}$ | $14.20_{\pm0.01}$ | - | - | $14.19_{\pm0.00}$ | $14.18_{\pm0.02}$ | - | $14.15_{\pm0.02}$ | - |
| $3 \times 10^{-2}$ | $14.37_{\pm0.01}$ | $14.26_{\pm0.02}$ | $14.25_{\pm0.02}$ | $14.27_{\pm0.01}$ | - | - | $14.20_{\pm0.01}$ | $14.19_{\pm0.02}$ | - | $14.18_{\pm0.02}$ | - |
| $1 \times 10^{-2}$ | $14.50_{\pm0.01}$ | $14.06_{\pm0.02}$ | $14.12_{\pm0.02}$ | $14.29_{\pm0.02}$ | - | - | $14.17_{\pm0.01}$ | $14.16_{\pm0.01}$ | $14.71_{\pm0.02}$ | $14.18_{\pm0.01}$ | $14.69_{\pm0.02}$ |
| $8 \times 10^{-3}$ | $14.54_{\pm0.01}$ | $14.02_{\pm0.02}$ | $14.11_{\pm0.02}$ | $14.27_{\pm0.01}$ | - | - | $14.16_{\pm0.02}$ | $14.16_{\pm0.01}$ | $14.72_{\pm0.04}$ | $14.16_{\pm0.02}$ | $14.71_{\pm0.02}$ |
| $5 \times 10^{-3}$ | $14.67_{\pm0.00}$ | $14.01_{\pm0.02}$ | $14.11_{\pm0.02}$ | $14.26_{\pm0.03}$ | - | - | $14.16_{\pm0.01}$ | $14.18_{\pm0.01}$ | $14.72_{\pm0.02}$ | $14.17_{\pm0.02}$ | $14.70_{\pm0.04}$ |
| $3 \times 10^{-3}$ | $14.84_{\pm0.01}$ | $14.01_{\pm0.02}$ | $14.16_{\pm0.02}$ | $14.27_{\pm0.02}$ | $14.06_{\pm0.01}$ | $14.09_{\pm0.01}$ | $14.19_{\pm0.01}$ | $14.19_{\pm0.01}$ | $14.77_{\pm0.03}$ | $14.20_{\pm0.01}$ | $14.77_{\pm0.05}$ |
| $1 \times 10^{-3}$ | $15.36_{\pm0.01}$ | $14.04_{\pm0.02}$ | $14.17_{\pm0.02}$ | $14.42_{\pm0.01}$ | $14.12_{\pm0.01}$ | $14.26_{\pm0.00}$ | $14.34_{\pm0.01}$ | $14.37_{\pm0.01}$ | $14.72_{\pm0.02}$ | $14.36_{\pm0.00}$ | $14.73_{\pm0.05}$ |
| $8 \times 10^{-4}$ | $15.47_{\pm0.02}$ | $14.08_{\pm0.01}$ | $14.17_{\pm0.02}$ | $14.47_{\pm0.01}$ | $14.14_{\pm0.02}$ | $14.31_{\pm0.01}$ | $14.35_{\pm0.02}$ | $14.38_{\pm0.01}$ | $14.74_{\pm0.02}$ | $14.38_{\pm0.01}$ | $14.76_{\pm0.02}$ |
| $5 \times 10^{-4}$ | $15.73_{\pm0.02}$ | $14.12_{\pm0.01}$ | $14.22_{\pm0.02}$ | $14.48_{\pm0.01}$ | $14.26_{\pm0.01}$ | $14.34_{\pm0.01}$ | $14.39_{\pm0.01}$ | $14.40_{\pm0.01}$ | $14.85_{\pm0.03}$ | $14.41_{\pm0.01}$ | $14.87_{\pm0.03}$ |
| $3 \times 10^{-4}$ | $16.59_{\pm0.04}$ | $14.75_{\pm0.02}$ | $14.91_{\pm0.04}$ | $14.94_{\pm0.02}$ | $14.81_{\pm0.02}$ | $14.81_{\pm0.02}$ | $14.88_{\pm0.03}$ | $14.88_{\pm0.01}$ | $14.97_{\pm0.02}$ | $14.48_{\pm0.01}$ | $14.99_{\pm0.04}$ |
| $1 \times 10^{-4}$ | $17.36_{\pm0.04}$ | $14.41_{\pm0.02}$ | $14.52_{\pm0.01}$ | $14.70_{\pm0.01}$ | $14.40_{\pm0.01}$ | $14.50_{\pm0.01}$ | $14.70_{\pm0.01}$ | $14.69_{\pm0.00}$ | $15.24_{\pm0.01}$ | $14.73_{\pm0.02}$ | $15.23_{\pm0.02}$ |
| $8 \times 10^{-5}$ | $17.71_{\pm0.11}$ | $14.41_{\pm0.01}$ | $14.53_{\pm0.01}$ | $14.79_{\pm0.01}$ | $14.42_{\pm0.01}$ | $14.53_{\pm0.01}$ | $14.77_{\pm0.01}$ | $14.77_{\pm0.01}$ | $15.15_{\pm0.02}$ | $14.81_{\pm0.01}$ | $15.15_{\pm0.02}$ |
| $5 \times 10^{-5}$ | $18.11_{\pm0.04}$ | $14.45_{\pm0.01}$ | $14.55_{\pm0.01}$ | $14.97_{\pm0.02}$ | $14.47_{\pm0.01}$ | $14.61_{\pm0.01}$ | $14.97_{\pm0.01}$ | $14.96_{\pm0.01}$ | $15.00_{\pm0.01}$ | $14.99_{\pm0.01}$ | $15.01_{\pm0.01}$ |
| $3 \times 10^{-5}$ | $18.49_{\pm0.07}$ | $14.50_{\pm0.01}$ | $14.61_{\pm0.01}$ | $15.22_{\pm0.01}$ | $14.55_{\pm0.01}$ | $14.72_{\pm0.01}$ | $15.20_{\pm0.00}$ | $15.23_{\pm0.01}$ | $14.96_{\pm0.01}$ | $15.57_{\pm0.01}$ | $14.95_{\pm0.02}$ |
| $1 \times 10^{-5}$ | $18.84_{\pm0.06}$ | $14.66_{\pm0.01}$ | $14.75_{\pm0.01}$ | $15.80_{\pm0.00}$ | $14.77_{\pm0.01}$ | $15.09_{\pm0.01}$ | $15.81_{\pm0.02}$ | $15.81_{\pm0.03}$ | $15.39_{\pm0.01}$ | $15.81_{\pm0.02}$ | $15.39_{\pm0.02}$ |
| $8 \times 10^{-6}$ | $18.89_{\pm0.04}$ | $14.70_{\pm0.01}$ | $14.80_{\pm0.01}$ | $15.94_{\pm0.02}$ | $14.84_{\pm0.01}$ | $15.20_{\pm0.01}$ | $15.95_{\pm0.02}$ | $15.95_{\pm0.02}$ | $15.50_{\pm0.02}$ | $15.93_{\pm0.02}$ | $15.50_{\pm0.01}$ |
| $5 \times 10^{-6}$ | $19.08_{\pm0.05}$ | $14.84_{\pm0.01}$ | $14.95_{\pm0.02}$ | $16.34_{\pm0.02}$ | $14.99_{\pm0.01}$ | $15.44_{\pm0.01}$ | $16.35_{\pm0.04}$ | $16.34_{\pm0.04}$ | $15.81_{\pm0.02}$ | $16.30_{\pm0.02}$ | $15.79_{\pm0.01}$ |
| $3 \times 10^{-6}$ | $19.10_{\pm0.08}$ | $15.03_{\pm0.01}$ | $15.20_{\pm0.02}$ | $16.93_{\pm0.05}$ | $15.26_{\pm0.02}$ | $15.77_{\pm0.01}$ | $16.95_{\pm0.03}$ | $16.94_{\pm0.04}$ | $16.25_{\pm0.02}$ | $16.81_{\pm0.03}$ | $16.23_{\pm0.05}$ |
| $1 \times 10^{-6}$ | $19.05_{\pm0.06}$ | $15.76_{\pm0.01}$ | $16.03_{\pm0.01}$ | $18.18_{\pm0.07}$ | $15.96_{\pm0.04}$ | $16.49_{\pm0.03}$ | $18.18_{\pm0.07}$ | $18.15_{\pm0.04}$ | $17.30_{\pm0.03}$ | $18.10_{\pm0.06}$ | $17.30_{\pm0.05}$ |

Table 18: Numerical results for the M2F task.

| LR | SW | MaxSW | DSW | MaxKSW | iMSW | viMSW | oMSW | rMSW | EBSW | RPSW | EBRPSW |
|---|---|---|---|---|---|---|---|---|---|---|---|
| $1$ | $14.58_{\pm0.03}$ | - | - | - | - | - | - | - | - | - | - |
| $8 \times 10^{-1}$ | $14.61_{\pm0.03}$ | - | - | - | - | - | - | - | - | - | - |
| $5 \times 10^{-1}$ | $14.62_{\pm0.03}$ | - | - | - | - | - | - | - | - | - | - |
| $3 \times 10^{-1}$ | $14.62_{\pm0.02}$ | - | - | - | - | - | - | - | - | - | - |
| $1 \times 10^{-1}$ | $14.70_{\pm0.02}$ | - | - | $14.98_{\pm0.02}$ | - | - | $14.62_{\pm0.02}$ | $14.63_{\pm0.01}$ | - | $14.68_{\pm0.13}$ | - |
| $8 \times 10^{-2}$ | $14.73_{\pm0.02}$ | $16.48_{\pm0.00}$ | $15.35_{\pm0.07}$ | $14.81_{\pm0.02}$ | - | - | $14.62_{\pm0.02}$ | $14.63_{\pm0.01}$ | - | $14.65_{\pm0.03}$ | - |
| $5 \times 10^{-2}$ | $14.80_{\pm0.02}$ | $15.17_{\pm0.08}$ | $14.95_{\pm0.02}$ | $14.65_{\pm0.02}$ | - | - | $14.63_{\pm0.01}$ | $14.67_{\pm0.03}$ | - | $14.64_{\pm0.01}$ | - |
| $3 \times 10^{-2}$ | $14.80_{\pm0.03}$ | $14.63_{\pm0.04}$ | $15.44_{\pm0.00}$ | $15.71_{\pm0.00}$ | - | - | $14.65_{\pm0.02}$ | $14.62_{\pm0.02}$ | - | $14.65_{\pm0.03}$ | - |
| $1 \times 10^{-2}$ | $14.95_{\pm0.02}$ | $14.60_{\pm0.02}$ | $14.63_{\pm0.04}$ | $14.69_{\pm0.03}$ | - | - | $14.58_{\pm0.03}$ | $14.62_{\pm0.02}$ | $15.18_{\pm0.04}$ | $14.62_{\pm0.02}$ | $15.16_{\pm0.03}$ |
| $8 \times 10^{-3}$ | $15.01_{\pm0.03}$ | $14.61_{\pm0.02}$ | $14.62_{\pm0.03}$ | $14.70_{\pm0.02}$ | - | - | $14.61_{\pm0.01}$ | $14.60_{\pm0.02}$ | $15.15_{\pm0.03}$ | $14.60_{\pm0.01}$ | $15.15_{\pm0.05}$ |
| $5 \times 10^{-3}$ | $15.09_{\pm0.03}$ | $14.53_{\pm0.03}$ | $14.60_{\pm0.04}$ | $14.69_{\pm0.05}$ | - | - | $14.61_{\pm0.03}$ | $14.63_{\pm0.02}$ | $15.11_{\pm0.04}$ | $14.60_{\pm0.02}$ | $15.16_{\pm0.04}$ |
| $3 \times 10^{-3}$ | $15.28_{\pm0.01}$ | $14.52_{\pm0.02}$ | $14.61_{\pm0.02}$ | $14.70_{\pm0.03}$ | $14.62_{\pm0.04}$ | $14.57_{\pm0.01}$ | $14.65_{\pm0.04}$ | $14.65_{\pm0.04}$ | $15.16_{\pm0.01}$ | $14.64_{\pm0.03}$ | $15.15_{\pm0.05}$ |
| $1 \times 10^{-3}$ | $15.78_{\pm0.01}$ | $14.57_{\pm0.03}$ | $14.65_{\pm0.01}$ | $14.89_{\pm0.01}$ | $14.59_{\pm0.01}$ | $14.72_{\pm0.03}$ | $14.80_{\pm0.02}$ | $14.83_{\pm0.03}$ | $15.13_{\pm0.01}$ | $14.67_{\pm0.02}$ | $15.10_{\pm0.05}$ |
| $8 \times 10^{-4}$ | $15.88_{\pm0.03}$ | $14.59_{\pm0.03}$ | $14.68_{\pm0.02}$ | $14.92_{\pm0.02}$ | $14.64_{\pm0.02}$ | $14.76_{\pm0.01}$ | $14.78_{\pm0.02}$ | $14.81_{\pm0.02}$ | $15.09_{\pm0.04}$ | $14.78_{\pm0.02}$ | $15.12_{\pm0.02}$ |
| $5 \times 10^{-4}$ | $16.17_{\pm0.03}$ | $14.62_{\pm0.02}$ | $14.74_{\pm0.03}$ | $14.92_{\pm0.02}$ | $14.75_{\pm0.01}$ | $14.77_{\pm0.03}$ | $14.82_{\pm0.02}$ | $14.85_{\pm0.02}$ | $15.26_{\pm0.06}$ | $14.81_{\pm0.02}$ | $15.25_{\pm0.02}$ |
| $3 \times 10^{-4}$ | $16.59_{\pm0.04}$ | $14.75_{\pm0.02}$ | $14.91_{\pm0.04}$ | $14.94_{\pm0.02}$ | $14.81_{\pm0.02}$ | $14.81_{\pm0.02}$ | $14.88_{\pm0.03}$ | $14.88_{\pm0.01}$ | $15.50_{\pm0.04}$ | $14.85_{\pm0.02}$ | $15.49_{\pm0.03}$ |
| $1 \times 10^{-4}$ | $18.05_{\pm0.05}$ | $14.89_{\pm0.04}$ | $14.96_{\pm0.02}$ | $15.16_{\pm0.02}$ | $14.81_{\pm0.02}$ | $14.97_{\pm0.02}$ | $15.14_{\pm0.04}$ | $15.13_{\pm0.02}$ | $15.63_{\pm0.04}$ | $14.94_{\pm0.03}$ | $15.66_{\pm0.02}$ |
| $8 \times 10^{-5}$ | $18.26_{\pm0.09}$ | $14.88_{\pm0.03}$ | $14.96_{\pm0.03}$ | $15.20_{\pm0.03}$ | $14.87_{\pm0.01}$ | $14.96_{\pm0.03}$ | $15.19_{\pm0.03}$ | $15.19_{\pm0.02}$ | $15.56_{\pm0.04}$ | $15.17_{\pm0.02}$ | $15.54_{\pm0.03}$ |
| $5 \times 10^{-5}$ | $18.92_{\pm0.06}$ | $14.92_{\pm0.04}$ | $15.03_{\pm0.04}$ | $15.38_{\pm0.03}$ | $14.94_{\pm0.02}$ | $15.05_{\pm0.04}$ | $15.37_{\pm0.03}$ | $15.38_{\pm0.04}$ | $15.43_{\pm0.01}$ | $15.24_{\pm0.01}$ | $15.41_{\pm0.05}$ |
| $3 \times 10^{-5}$ | $19.21_{\pm0.05}$ | $14.95_{\pm0.02}$ | $15.07_{\pm0.01}$ | $15.61_{\pm0.03}$ | $15.00_{\pm0.02}$ | $15.14_{\pm0.01}$ | $15.60_{\pm0.01}$ | $15.59_{\pm0.03}$ | $15.35_{\pm0.01}$ | $15.42_{\pm0.02}$ | $15.39_{\pm0.02}$ |
| $1 \times 10^{-5}$ | $19.64_{\pm0.04}$ | $15.12_{\pm0.01}$ | $15.23_{\pm0.02}$ | $16.21_{\pm0.03}$ | $15.22_{\pm0.04}$ | $15.55_{\pm0.03}$ | $16.23_{\pm0.03}$ | $16.22_{\pm0.03}$ | $15.72_{\pm0.04}$ | $15.64_{\pm0.01}$ | $15.70_{\pm0.04}$ |
| $8 \times 10^{-6}$ | $19.62_{\pm0.05}$ | $15.19_{\pm0.02}$ | $15.27_{\pm0.02}$ | $16.43_{\pm0.02}$ | $15.28_{\pm0.01}$ | $15.61_{\pm0.01}$ | $16.42_{\pm0.03}$ | $16.39_{\pm0.03}$ | $15.91_{\pm0.05}$ | $15.94_{\pm0.02}$ | $15.91_{\pm0.05}$ |
| $5 \times 10^{-6}$ | $19.79_{\pm0.06}$ | $15.32_{\pm0.01}$ | $15.38_{\pm0.02}$ | $16.86_{\pm0.02}$ | $15.47_{\pm0.03}$ | $15.87_{\pm0.03}$ | $16.85_{\pm0.02}$ | $16.85_{\pm0.05}$ | $16.25_{\pm0.03}$ | $16.30_{\pm0.02}$ | $16.26_{\pm0.04}$ |
| $3 \times 10^{-6}$ | $19.85_{\pm0.13}$ | $15.56_{\pm0.02}$ | $15.65_{\pm0.02}$ | $17.57_{\pm0.06}$ | $15.70_{\pm0.02}$ | $16.15_{\pm0.02}$ | $17.60_{\pm0.09}$ | $17.53_{\pm0.07}$ | $16.72_{\pm0.03}$ | $16.81_{\pm0.02}$ | $16.74_{\pm0.04}$ |
| $1 \times 10^{-6}$ | $19.84_{\pm0.10}$ | $16.23_{\pm0.02}$ | $16.46_{\pm0.02}$ | $18.95_{\pm0.06}$ | $16.39_{\pm0.05}$ | $16.93_{\pm0.05}$ | $18.93_{\pm0.08}$ | $18.88_{\pm0.06}$ | $17.92_{\pm0.07}$ | $18.05_{\pm0.09}$ | $18.02_{\pm0.04}$ |

**Table 19:** Numerical results for the A2C translation task.

