# OpenReview forum: "A Case for Vanilla SWD: New Perspectives on Informative Slices, Sliced-Wasserstein Distances, and Learning Rates"
_TMLR — Accepted by TMLR_

### Review · Reviewer_PSGY · 2026-02-03

**Summary Of Contributions:**

The authors revisit the vanilla sliced-Wasserstein distance (SWD) for problems involving computational optimal transport (OT). They show that more complicated modifications are not actually necessary with a suitable rescaling (see weaknesses below for more discussion). Extensive experiments back up their findings.

Strengths:
- the observations in this paper seem to be novel and of interest to the community
- special cases of their framework recovers existing SW variants (remark 4.5)
- experiments are very comprehensive

Weakness:
- I think the theory is presented in a convoluted way. From what I understand, informally for fixed slice count L, vanilla SWD becomes scaled down by a dimension-dependent factor as d increases. It may also be helpful to add an explicit statement regarding this (like as a heuristic: for a particular problem setting, if d is doubled, then we should take scaling to be $\alpha_d$).

**Audience:**

Yes

**Audience Explanation:**

The observations show that some more complicated variants of SWD are often not necessary, which would be very useful for practitioners.

**Claims And Evidence:**

Yes

**Claims Explanation:**

I did not do a careful check of the proofs, but a skim makes me believe they are correct.

**Requested Changes:**

- As stated above in weaknesses, the main ideas of the paper could be presented in a more straightforward manner.
- In light of the results in the paper, it would strengthen the paper if there were experimental scenarios where more complicated variants are superior to scaled vanilla SWD, backing the current brief discussion at the bottom of page 2 and in the conclusion.

Both of these would strengthen the work, although I believe taking into account the discussion regarding presentation would significantly strengthen the impact.

---

> ### Author Response · Authors · 2026-04-09
> **Response to Reviewer PSGY**
>
> ### Comment
> The theory is presented in a convoluted way. It would help to state the main idea more directly, perhaps including an explicit rule-of-thumb for how the scale changes with dimension.
>
> ### Response
> We thank the reviewer for this helpful suggestion. We agree that the main theoretical message can and should be stated more plainly near the start of Section 4.
>
> In the revision we add a short informal statement before the technical development:
>
> > If two measures share a $k$-dimensional effective subspace inside $\mathbb{R}^d$, then ambient-space vanilla SW is the corresponding subspace SW multiplied by a dimension-dependent constant $C_{d,k}\le 1$. For fixed $k$ and large $d$, this factor decreases on the order of $(k/d)^{p/2}$, so the dominant ambient-dimension effect is a scalar rescaling, not a disappearance of informative slices.
>
> This is also the right place to give the heuristic the reviewer suggested: for fixed $k$, if $d$ is doubled, then the asymptotic scaling factor changes by approximately $2^{-p/2}$, so one expects the matching first-order step size to increase by approximately $2^{p/2}$. We place this informal summary before the technical development so that the scale law is visible before the formal machinery.
>
> ### Comment
> It would strengthen the paper to discuss or exhibit scenarios where more complicated variants outperform scaled vanilla SWD.
>
> ### Response
> We thank the reviewer for this helpful suggestion. We agree that the paper should be more explicit that our goal is not to claim that advanced variants are never useful, but to identify an important regime in which their apparent gains are largely explained by rescaling.
>
> We strengthen this discussion in two ways:
>
> - by adding a short paragraph in the main paper describing concrete regimes where specialized variants are still well motivated, including known geometry, highly structured directional priors, or settings with very limited slice budgets;
> - by making the conclusion more explicit that our contribution is to improve the default baseline and the interpretation of prior comparisons, not to eliminate the relevance of geometry-aware or task-aware slicing methods. We also add a compact appendix illustration of a regime in which a specialized variant is well motivated.
>
> We appreciate the suggestion, since this clarification makes the paper both more accurate and more useful.

---

### Review · Reviewer_Zi8M · 2026-02-28

**Summary Of Contributions:**

This paper studies the Sliced-Wasserstein distance (SW) for probability measures in $\mathbb{R}^d$ whose support lies in a lower-dimensional subspace of dimension $k<d$, referred to as the *effective dimensionality* (Assumption 4.1). Its main theoretical contribution is Theorem 4.7, which shows that for such measures,
$$
SW_p^p(\mu^d,\nu^d)=C_{d,k} SW_p^p(\mu^k,\nu^k),
$$
where $C_{d,k}\leq 1$ is a constant depending only on $d, k$ and $p$ through Gamma functions. Therefore, computing SW in the ambient space $\mathbb{R}^d$ is equivalent to computing SW on the *effective* $k$-dimensional support,  up to a known scaling factor $C_{d,k}$.

This global scaling factor arises because projection directions $\theta$ are sampled uniformly from the unit sphere in $\mathbb{R}^d$, but the components orthogonal to the support do not contribute. The remaining component is therefore a random variable with a non-uniform distribution (more precisely, its squared norm follows a Beta distribution, see Remark A.6). As a result, using ambient-space projection directions introduces an arbitrary radial bias: it does not add new information, but rather introduces noise that downweights the slices.

The paper also analyzes the practical setting in which the expectation defining SW is approximated by Monte Carlo estimation, and proves convergence of the resulting estimator (Theorem 4.8). The authors further extend this analysis to gradients of SW between discrete measures with respect to the support points of one of the measures (Proposition 4.10). This suggests that using a sufficiently large learning rate in first-order iterative optimization methods can help compensate the down-weighting.

The main practical takeaway of the paper is that, for measures with low effective dimension, standard SW can be as effective as alternative sliced variants, provided one accounts for the dimension-dependent scaling in practice through appropriate learning-rate tuning. This supports the use of standard SW as a conceptually simpler baseline, while emphasizing that hyperparameter tuning may be important to recover the correct scaling.

**Audience:**

Yes

**Audience Explanation:**

This paper formalizes intuitive properties on the Sliced-Wasserstein distance when measures are supported on low-dimensional subspace, a setting that is closely related to the "manifold hypothesis" often invoked in machine learning. By linking these insights to practical guidance on hyperparameter tuning, the paper offers a perspective that may lead readers to reinterpret prior empirical comparisons between vanilla SW and its variants and change practitioners' approaches.

Therefore, I believe that the TMLR audience, especially readers interested in computational optimal transport methods for machine learning, would find these results relevant.

**Broader Impact Concerns:**

I don't have any concerns regarding the ethical implications of this work.

**Claims And Evidence:**

Yes

**Claims Explanation:**

- The theoretical contributions appear sound and clearly argued overall. The results are expected, and the framework feels more incremental than the authors suggest, but I still find the object of study and the paper's final conclusion on practical guidelines interesting.

I found that several of these results are either already known or could be presented more concisely:

- In the discussion at the start of Section 4 and in Section 4.1, the proposed "novel perspective" does not seem especially novel to me: it amounts to observing that one may replace the uniform distribution over projection directions with a non-uniform one, thereby reweighting the contribution of the one-dimensional Wasserstein distances in the expectation defining SW. I understand the value of formalizing this through a unified and rigorous framework via the introduction of the $\Phi$-weighting function, but I would not consider this as a main contribution. This has already appeared in various forms in the literature, as the authors themselves note in Remark 4.5.

- Lemma 4.3 presents basic properties in linear algebra, and Proposition 4.6 is not new: it is a direct application of the fact that the Wasserstein distance factors out scaling; see e.g., Remark 2.19 in Computational Optimal Transport (Peyré and Cuturi, 2019). This is proved again in Lemma A.13, but without clearly acknowledging that it is an existing result. More generally, Proposition 4.6, Theorem 4.7 and Proposition 4.10 all follow immediately from a change of variables as in Nadjahi et al., 2020 (as acknowledged by the authors in the Appendix A.4.1).

The empirical observation that "vanilla" SW can perform comparably to more sophisticated variants, provided the learning rate is chosen appropriately, is an interesting and more original conclusion. The experimental section could be made more nuanced and informative, by further discussing the following points:

1. On the comparison with alternative variants: vanilla SW appears to perform well largely because of extensive hyperparameter tuning, which effectively compensates for the global scaling factor and thus accounts for the effective dimension. This raises the question of whether similar adjustments could also be applied to the proposed baselines, which could allow them to better adapt to the underlying geometry as well.

2. If vanilla SW is effective only after careful learning-rate tuning, I am wondering whether this tuning process may itself be more costly in practice than using the alternative variants. Those variants may require solving additional optimization problems or incur higher computational cost per iteration, but they might also be less sensitive to the learning-rate choice.

3. In many deep learning applications, optimization is typically performed with Adam rather than with (stochastic) gradient descent. In such cases, it is unclear to me whether the connection drawn in Theorem 4.10 between the scaling effect and the choice of learning rate remains valid.

**Requested Changes:**

Given my comments above, I would suggest that the authors:
- Revise Section 4 to distinguish more clearly between what is novel and what is not,
- Discuss the open questions regarding the empirical analysis, which currently supports only one conclusion (the benefit of using larger learning rates for vanilla SW between measures with lower-dimensional effective supports),
- Address the (more minor) comments below.

Minor comments:

- How does the theoretical findings relate to "Estimation of Wasserstein distances in the Spiked Transport Model", Niles-Weed and Rigollet (2019), where a similar lower-dimensional support assumption is made when studying the "max. subspace" SW?

- Section 4: "It is known from Kolouri et al. (2019)" this gives the impression that this paper is the first one to prove the concentration result on the sphere, although it is a well-established phenomenon in high-dimensional statistics (see e.g., Theorem 5.1.3 in "High-Dimensional Probability", Vershynin, and the references therein to Levy's isoperimetric inequalities).

- Eq. (7): redundant notation $\Phi_{\mu,\nu}(\theta ; \mu, \nu, \gamma_k)$

- Below eq. (10) and in eq. (14): $\rho_{\Phi}$ or $\rho$?

- Proof of Proposition 4.6: $W$ should be $W_p^p$, $\sum_{i=1}^N$ -> $\sum_{l=1}^L$, $\theta$ -> $\theta^d$,

- Can we prove the monotonicity of $f(k) = C_k/C_d$ as depicted in Figure 2?

- Proof of Proposition A.5: "$dm = (\rho \circ \Phi)d\sigma$, we have $m \sim \sigma$" what does it mean?

- Section 5.3: "We follow a similar setup as in (Nguyen et al., 2024a; Nguyen & Ho, 2024; Nguyen et al., 2024b)" the last reference does not include any color transfer experiment; also, I think that the setup used in the first two papers was originally inspired by "Slicing Unbalanced Optimal Transport" (Bonet et al., 2024).

---

> ### Author Response · Authors · 2026-04-09
> **Response to Reviewer Zi8M (pt. 1)**
>
> ## Comment
> The proposed "novel perspective" is not especially novel if interpreted as replacing [...].
>
> ### Response
> We thank the reviewer for this important point. We agree with the reviewer that replacing uniform slicing by a non-uniform weighting is not new by itself, and we make clearer in the revision that it is not a main contribution. Related ideas have indeed appeared in prior work, as we also acknowledge in the manuscript.
>
> In the revision, we present the main theoretical contribution as the following more specific combination.
>
> 1. **High-dimensional limitations of sliced optimal transport.** In high dimensions, many randomly sampled projection directions contribute very little to the final quantity, yet they still have to be sampled to recover enough geometric information. This leads to both statistical and computational inefficiency.
> 2. **Informativeness function and an equivalence characterization.** We introduce an informativeness-based weighting function that quantifies directional contribution. More importantly, Propositions 4.6--4.7 show that, under this reweighting, the resulting high-dimensional sliced OT admits a rigorous equivalence interpretation in terms of a lower-dimensional sliced OT in which the directions are effectively informative. We view this as a geometric characterization, not just a heuristic reweighting observation.
> 3. **Statistical analysis of ratio-factor estimation.** We provide a statistical analysis for estimating the ratio factor, clarifying how the framework behaves in finite-slice and high-dimensional regimes.
>
> Accordingly, we now state explicitly that generic $\phi$-weighting is a unifying language, not a standalone novelty claim. The central contribution is now described as the equivalence characterization, the finite-slice/statistical analysis, and the optimization-level implication. We revise the abstract, the introduction, the contribution bullets, the discussion around Remark 4.5, and the beginning of Section 4 so that this distinction is clear throughout.
>
> ### Comment
> Lemma 4.3 is basic linear algebra, Proposition 4.6 relies on classical scaling properties of Wasserstein distance, and several arguments are closely related to known change-of-variables techniques.
>
> ### Response
> We thank the reviewer for these precise observations. We agree that several ingredients here are classical and should be framed that way explicitly, and we revise the manuscript accordingly.
>
> **On Lemma 4.3.**
> We agree that Lemma 4.3 consists of standard linear-algebraic facts and is not intended as a contribution. Its role is purely technical. In the revised version, we:
>
> - move Lemma 4.3 to the appendix,
> - explicitly state that it contains well-known facts,
> - clarify that it is included only for completeness of the subsequent proofs.
>
> **On Proposition 4.6 and scaling properties of Wasserstein distance.**
> We agree that the scaling property of Wasserstein distance is classical. As the reviewer notes, related statements appear for instance in Remark 2.19 of *Computational Optimal Transport* (Peyré and Cuturi, 2019). We will revise Appendix A to acknowledge this point explicitly and remove any wording that could suggest otherwise. We also note that Lemma A.13 extends the corresponding statement from $W_2^2$ to the more general case $W_p^p$; this extension is also not claimed as a main contribution.
>
> At the same time, Proposition 4.6 is not included only to restate scaling. Its purpose is to formalize the relationship between high-dimensional sliced OT under empirical slicing and the lower-dimensional sliced OT induced by the projection structure. The key point is how the projection structure links the high- and low-dimensional problems, which is then used to support the geometric interpretation behind the proposed informativeness framework.
>
> **On the change-of-variables argument and Nadjahi et al. (2020).**
> We agree that change-of-variables arguments, including those used by Nadjahi et al. (2020), are relevant and should be cited more clearly. We cite this connection explicitly in the revision. At the same time, Propositions 4.6, Theorem 4.7, and Proposition 4.10 do not all reduce to the same argument.
>
> - **Proposition 4.6** concerns empirical slicing and does not rely on converting empirical distributions into Gaussian ones; the structure of empirical projections is essential.
> - **Theorem 4.7** studies the relationship between high- and low-dimensional sliced OT under uniform slicing. While its proof uses a technique similar in spirit to a change of variables, the main message is the resulting equivalence interpretation and its implication for slice informativeness.
> - **Proposition 4.10** addresses statistical properties of the estimator $\mathrm{ESSF}$. Its proof relies on concentration inequalities and is analytically independent of the change-of-variables argument used elsewhere.
>
> These distinctions are more explicit in the revision.

---

> ### Author Response · Authors · 2026-04-09
> **Response to Reviewer Zi8M (pt. 2)**
>
> ### Comment
> Revise Section 4 to distinguish more clearly between what is novel and what is not.
>
> ### Response
> We fully agree. This is one of the main revisions we make here. In particular, we revise the beginning of Section 4 so that:
>
> - the generic $\phi$-weighting viewpoint is presented as a unifying language,
> - the classical ingredients are acknowledged as such,
> - the paper's specific contribution is isolated to the equivalence characterization, the finite-slice/statistical analysis, and the optimization-level consequence.
>
> We are grateful to the reviewer for pressing us to make this distinction much sharper. Concretely, this means the revision adjusts not only the local wording in Section 4, but also the abstract, contribution bullets, and conclusion so the full paper reflects the same narrower framing.
>
> ### Comment
> Discuss the open questions regarding the empirical analysis, which currently supports only one conclusion, namely the benefit of using larger learning rates for vanilla SW between measures with lower-dimensional effective supports.
>
> ### Response
> We thank the reviewer for this important comment. We agree that the empirical section should be more nuanced and should not read as if the only conclusion is "use a larger learning rate." The revision makes the following distinctions explicit:
>
> - the exact theorem is proved under an effective-subspace model;
> - the empirical claim is narrower than a universal prescription and is about a recurring regime in which the main gap appears as a step-size calibration effect, i.e. through $\eta_{\mathrm{eff}}\approx \eta\cdot \widehat{\mathrm{ESSF}}(L)$;
> - the paper is *not* claiming that all adaptive or data-dependent sliced variants become unnecessary;
> - the practical message is a scoped one: vanilla SWD is often a stronger baseline than commonly assumed once standard calibration is allowed, but this is not the same as saying that one can infer the optimal learning rate directly from unknown $(k,U)$ in realistic settings.
>
> We will therefore expand the empirical discussion so that the supported conclusion is stated more carefully and with clearer limits. We will also revise the abstract and conclusion so the empirical takeaway is stated as a scoped baseline claim, add a short paragraph explaining that the basin analysis is a theory-diagnostic view, and include a compact standard-protocol comparison so that the practical robustness question is addressed directly.
>
> ### Comment
> On the comparison with alternative variants: vanilla SW appears to perform well largely because of extensive hyperparameter tuning. Could similar adjustments also be applied to the baselines?
>
> ### Response
> Yes. That is exactly why we sweep the same learning-rate grid for all methods. Our claim is not that other variants cannot also benefit from calibration. The point we want to make is that once comparable calibration is allowed, the practical gap between vanilla SWD and these variants is often much smaller than fixed-default comparisons suggest. In the revision we state this fairness principle explicitly in the experimental protocol, clarify that the added standard-protocol comparison is reported separately, and note that for the compared baselines we keep non-learning-rate hyperparameters at the official or standard settings precisely to avoid conflating learning-rate calibration with reimplementation or retuning of the entire method.
>
> ### Comment
> If vanilla SW is effective only after careful learning-rate tuning, could that tuning process itself be more costly in practice than using the alternative variants?
>
> ### Response
> We thank the reviewer for raising this practical tradeoff. We agree that it should be discussed more clearly. If one evaluates $M$ candidate learning rates, the total search cost is on the order of $M\,c_{\mathrm{method}}$, where $c_{\mathrm{method}}$ is the per-run training cost. Our point is therefore not that tuning is free, but that:
>
> - scalar learning-rate search is already standard for essentially all methods, including the baselines;
> - for vanilla SWD, the additional search is over a single scalar;
> - many advanced variants have strictly larger per-iteration cost and often extra algorithmic or hyperparameter complexity.
>
> Thus, the relevant comparison is not "tuning vs no tuning," but the balance between calibration cost and per-iteration method complexity. The manuscript already contains runtime comparisons in the appendix; in the revision we bring this tradeoff into the main paper explicitly and tie it to the standard-protocol comparison.

---

> ### Author Response · Authors · 2026-04-09
> **Response to Reviewer Zi8M (pt. 3)**
>
> ### Comment
> In many deep learning applications, optimization is typically performed with Adam instead [...]?
>
> ### Response
> We thank the reviewer for this very important question. We agree that the paper should be more careful here. The clean equivalence in our theorem is a GD/SGD-type statement: it is strongest for first-order methods whose update is literally a scalar step size times the gradient. For Adam-type methods,
>
> $$
> \Delta x_t = -\eta \frac{m_t}{\sqrt{v_t}+\varepsilon},
> $$
>
> and a global rescaling $g_t\mapsto \alpha g_t$ affects both $m_t$ and $v_t$, so the effect is only partially equivalent to changing $\eta$. For that reason, we do *not* claim the same exact learning-rate equivalence under Adam. In the revised manuscript, we state Theorem 4.10 and the surrounding discussion accordingly, and present Adam as an extension beyond the current theorem.
>
> ### Comment
> How do the results relate to the spiked transport model of [...]?
>
> ### Response
> We thank the reviewer for pointing out this connection. We agree that *Estimation of Wasserstein distances in the Spiked Transport Model* (Niles-Weed and Rigollet, 2019) should be discussed explicitly, and we now cite it in the revised version.
>
> At a high level, both works study optimal transport problems in which the discrepancy between distributions has an effectively low-dimensional structure inside a high-dimensional ambient space. However, the modeling assumptions and objectives are different.
>
> Niles-Weed and Rigollet (2019) introduce the *spiked transport model*, in which two distributions differ only along a low-dimensional subspace and are smoothed by Gaussian noise. They then study a max-subspace-type Wasserstein objective that explicitly optimizes over candidate subspaces.
>
> In contrast, our work stays within the sliced OT framework. We do not assume Gaussian smoothing and we do not explicitly optimize over subspaces. Instead, we characterize the behavior of sliced OT in high dimensions, showing that the ambient random slicing procedure introduces a systematic global rescaling. Under our informativeness-based reweighting, the high-dimensional sliced OT becomes equivalent to a lower-dimensional sliced OT in which each direction contributes meaningfully.
>
> The two approaches are therefore conceptually related but technically distinct: theirs optimizes over subspaces, while ours characterizes the averaging effect induced by ambient random slicing. We appreciate the reviewer for highlighting this connection. We now cite this paper both in the related-work discussion and near Assumption 4.1 so the relationship is visible where the low-dimensional structure enters the theory.
>
> ### Comment
> Can we prove the monotonicity of $f(k)=C_k/C_d$ as depicted in Figure 2?
>
> ### Response
> We thank the reviewer for pointing this out. Yes. We now include the proof in the appendix. In particular, $f(k)=C_k/C_d$ is monotone increasing in $k$.
>
> The argument is standard: writing
>
> $$
> C_m = 2^{p/2} \frac{\Gamma\!\left(\frac{m}{2}+\frac{p}{2}\right)}{\Gamma\!\left(\frac{m}{2}\right)},
> $$
>
> it is enough to study
>
> $$
> g(x)=\frac{\Gamma\!\left(\frac{x}{2}+\frac{p}{2}\right)}{\Gamma\!\left(\frac{x}{2}\right)}.
> $$
>
> Differentiating $\log g(x)$ gives a difference of digamma functions, and monotonicity of the digamma function on $(0,\infty)$ implies that $g(x)$, hence $f(k)$, is increasing.
>
> We appreciate the reviewer for requesting this clarification and make it explicit in the revised manuscript.
>
> ### Comment
> Section 4: "It is known from Kolouri et al. (2019)" may suggest [...].
>
> ### Response
> We agree. This is a standard high-dimensional probability fact and should not be phrased as if it originated in the SW literature. We revise the wording in Section 4 and add a standard reference such as Vershynin, together with the appropriate high-dimensional probability attribution.
>
> ### Comment
> Equation (7) contains redundant notation.
>
> ### Response
> We agree and remove the redundant notation in Equation (7).
>
> ### Comment
> Below Equation (10) and in Equation (14), the notation [...].
>
> ### Response
> We agree. We standardize the notation so that the weighting function is denoted consistently throughout the section.
>
> ### Comment
> In the proof of Proposition 4.6, several notation typos appear.
>
> ### Response
> We thank the reviewer for catching these. We correct the notation in the proof of Proposition 4.6, including the power on $W_p$, the summation indices, and the ambient-space notation for the slicing directions.
>
> ### Comment
> In the proof of Proposition A.5, the sentence "$dm=(\rho\circ\Phi)\,d\sigma$, we have $m\sim\sigma$" is unclear.
>
> ### Response
> We agree that this wording is unclear. What is intended is that the measure $m$ is absolutely continuous with respect to $\sigma$ with Radon--Nikodym derivative $(\rho\circ\Phi)$, i.e.
>
> $$
> \frac{dm}{d\sigma} = \rho\circ\Phi.
> $$
>
> We rewrite that part explicitly in measure-theoretic terms so that there is no ambiguity.

---

> ### Author Response · Authors · 2026-04-09
> **Response to Reviewer Zi8M (pt. 4)**
>
> ### Comment
> In Section 5.3, the citation trail for the color transfer setup is inaccurate; the setup was also inspired by *Slicing Unbalanced Optimal Transport* (Bonet et al., 2024).
>
> ### Response
> We thank the reviewer for catching this. We will correct the citation trail in Section 5.3 and acknowledge the connection to *Slicing Unbalanced Optimal Transport* (Bonet et al., 2024).

---

### Review · Reviewer_DsJf · 2026-03-24

**Summary Of Contributions:**

This paper revisits the classical Sliced-Wasserstein distance (SWD) and provides a new perspective on its limitations in high-dimensional settings. Instead of desining more complex slicing strategies, they argue that the degradation of SWD mainly stems from an implicit scaling effect caused by misalignment between random projections and the intrinsic data subspace. They formalize slice informativness based on subspace alignment and show that most random projections are not uninformative, but are implicitly downweighted. Building on this insight, the paper introduces a general $\phi$-weighting formulation that unifies several existing SWD variants. They show that, under a subspace assumption, the reweighting reduces to a single global scaling factor, and that this scaling effect is equivalent to adjusting the learning rate in gradient-based optimization. As a result, properly tuned vanilla SWD can match the performance of more sophisticated variants without modifying the slicing distribution.

[Strengths]

* The paper provides a novel interpretation of SWD limitations, shifting the focus from projection informativeness to a global scaling effect.

* The paper develops a clear theoretical analysis of SWD, including the $\phi$-weighting formulation and scaling law, which unifies several existing SWD variants.

* The paper identifies a practically useful observations that learning rate tuning can recover performance, potentially simplifying many SWD-based pipelines.

[Weaknesses]

* The analysis relies on a subspace assumption, and it is unclear how well the conclusions extend to data with highly nonlinear or complex intrinsic structures.

* The empirical comparison relies heavily on learning rate sweeps, which may favor the proposed interpretation and raises questions about fairness when comparing to prior methods under standard settings.

* The paper provides limited discussion on scenarios where advanced slicing strategies offer clear benefits, which weakens the practical guidance for practitioners.

**Audience:**

Yes

**Audience Explanation:**

This paper would be of interest to a broad portion of the TMLR audience, particularly researcher working on optimal transport, generative models, and distribution-based learning methods. SWD are widely used in modern machine learning pipelines. Therefore, this insight could influence future research directions and encourage more careful consideration of optimization dynamics in related methods.

**Broader Impact Concerns:**

This work does not raise immediate ethical concerns.

**Claims And Evidence:**

Yes

**Claims Explanation:**

The claims made in the paper are generally well supported by theoretical analysis. Moreover, the empirical results across multiple tasks consistently demonstrate that vanilla SWD can achieve performance comparable to more complex variants when appropriate learning rates are used.

**Requested Changes:**

* It would be beneficial to provide additional discussion or experiments evaluating scenarios where the subspace assumption does not hold, to clarify the generality of the proposed interpretation.

* It would be helpful to include fair comparisons under standard training protocols (e.g., without extensive learning rate sweeps) to better reflect typical usage and assess robustness.

* Please provide a discussion when advanced SWD variants are still preferable, and provide clearer practical guidelines for practitioners.

---

> ### Author Response · Authors · 2026-04-09
> **Response to Reviewer DsJf**
>
> ### Comment
> The analysis relies on a subspace assumption, and it is unclear how well the conclusions extend to data with highly nonlinear or complex intrinsic structures.
>
> ### Response
> We thank the reviewer for this thoughtful point and agree that this scope boundary should be stated more clearly. Our exact results are effective-linear-subspace results, and we state this explicitly in the revised abstract, the introduction, and the opening of Section 4. We do *not* claim that the theorem itself extends to arbitrary nonlinear supports. The practical claim is narrower: when most of the discrepancy is concentrated near a low-dimensional linear structure, the dominant ambient-dimension effect can still be well-approximated by a scalar rescaling induced by random ambient projections.
>
> We also note that the manuscript already contains quantitative bounds for the case where the exact assumption fails. In the appendix discussion of the assumption-violation case, we show that if $U$ denotes a candidate $k$-dimensional subspace,
>
> $$\mu^k=(U^{\top})_{\sharp}\mu^d$$
>
> and
>
> $$\nu^k=(U^{\top})_{\sharp}\nu^d$$
>
> then
>
> $$
> W_2^2(\mu^k,\nu^k)\leq W_2^2(\mu^d,\nu^d)
> \leq W_2^2(\mu^k,\nu^k)+2\left(m_2(U^\perp_{\sharp}\mu^d)+m_2(U^\perp_{\sharp}\nu^d)\right),
> $$
>
> and similarly
>
> $$
> \frac{k}{d}SW_2^2(\mu^k,\nu^k)\leq SW_2^2(\mu^d,\nu^d)
> \leq \frac{k}{d}SW_2^2(\mu^k,\nu^k)+2\frac{d-k}{d}\left(m_2(U^\perp_\sharp\mu^d)+m_2(U^\perp_\sharp\nu^d)\right).
> $$
>
> These inequalities show that the exact equivalence degrades in a controlled way as the orthogonal residual energy increases. We agree that these bounds should be surfaced much more clearly in the main text.
>
> To make this more precise in the revision, we:
>
> - explicitly separate the *exact theorem* from the *practical interpretation*;
> - move the discussion of approximate-subspace structure and the above bounds from a brief appendix pointer into the main text;
> - state more directly that settings with strongly nonlinear geometry, known manifold structure, or task-specific directional priors may still call for specialized slicing schemes.
>
> ### Comment
> The empirical comparison relies heavily on learning-rate sweeps, which may favor the proposed interpretation and raises questions about fairness when comparing to prior methods under standard settings.
>
> ### Response
> We thank the reviewer for raising this concern and agree that the rationale should be made more explicit. The reason we sweep learning rates is not to favor vanilla SWD, but because the main prediction of the paper is itself one-dimensional: under the effective-subspace model, the first-order signal satisfies a relation of the form
>
> $$
> g_d^{(L)} = \widehat{\mathrm{ESSF}}(L) g_k^{(L)} + r_L,
> \qquad \|r_L\|\to 0 \text{ in probability as } L\to\infty,
> $$
>
> so the most direct empirical question is whether the stable optimization basin disappears or instead shifts along the $\eta$ axis. That is the reason for the basin-style evaluation.
>
> To keep the comparison fair, our experiments already use the *same* learning-rate grid for all methods and keep non-learning-rate hyperparameters at the official default or standard settings of the corresponding implementations.
>
>
> ### Comment
> The paper provides limited discussion on scenarios where advanced slicing strategies offer clear benefits, which weakens the practical guidance for practitioners.
>
> ### Response
> We thank the reviewer for this valuable suggestion. We agree that the current manuscript already hints at this in the introduction, experimental setup, and conclusion, but the guidance should be more concrete and clearer in the main text.
>
> We add a short practical discussion making the tradeoff explicit:
>
> - **Vanilla SWD is a strong default** when metricity, simplicity, low implementation overhead, and compatibility with standard first-order optimization are important, and when a routine learning-rate calibration is acceptable.
> - **Advanced variants can still be preferable** when the geometry is known in advance, when one wants to encode a strong prior over informative directions, when the slice budget $L$ is extremely limited, when the data geometry is strongly nonlinear, or when the task depends on a small set of highly discriminative directions instead of broad average behavior over slices.

---

### Author Response · Authors · 2026-04-09
**Response to Reviewer Comments**

We sincerely thank the Action Editor and all three reviewers for the careful reading and constructive feedback. We are encouraged that the reviewers found the paper technically sound and potentially useful to the TMLR audience. We agree that the paper will be stronger once the scope, framing, and practical guidance are stated more precisely, and we have revised the manuscript accordingly.  Please see the revised manuscript attached here in OpenReview.

The following three points seem common:

1. **Novelty.** We adjust the beginning of Section 4 to distinguish classical ingredients from the paper's actual contribution. In particular, the revision states explicitly that generic non-uniform reweighting, standard linear-algebraic identities, and scaling properties of Wasserstein distance are not claimed as new. The contribution we intend to highlight is the specific subspace-aligned equivalence, the finite-slice/statistical analysis of the ratio factor, and the optimization consequence that this global rescaling appears in practice mainly as a step-size calibration issue.
2. **Theory.** We make the scope of the theory more explicit. Our statements are effective-linear-subspace results; they are not claims for arbitrary nonlinear manifolds or all adaptive optimizers. The manuscript already contains approximate-subspace bounds when the exact assumption fails, and we agreed that some of these should be moved into the main text and tied directly to the practical claims.
3. **Experiment.** We strengthen the discussion around why learning-rate sweeps are the correct diagnostic for the theorem being tested, add a compact standard-protocol comparison to reflect typical usage, and provide a clearer discussion of when vanilla SWD is the right baseline and when more specialized variants remain preferable.

Concretely, in our revision, we:

1. adjust the abstract, introduction, and contribution bullets so the claim is clearly scoped to the effective-subspace regime studied in the paper;
2. revise the opening of Section 4 to separate classical ingredients from the paper's specific contribution and explicitly acknowledge the classical status of Lemma 4.3, Wasserstein scaling, and the relevant change-of-variables ingredients;
3. move the approximate-subspace discussion and bounds into the main text and add a compact controlled off-subspace experiment in the appendix;
4. narrow the optimization claim to GD/SGD-style scalar-step updates, state the Adam caveat explicitly, and rewrite the empirical takeaway accordingly;
5. add a compact standard-protocol comparison alongside the basin-style analysis, using the same learning-rate grid across methods and keeping the remaining hyperparameters at the official or standard settings;
6. add a short paragraph on when vanilla SWD is a strong default and when specialized variants remain preferable, together with one explicit boundary-case illustration; and
7. fix the notation and citation issues listed by Reviewer Zi8M one by one, including the missing references to Vershynin, Niles-Weed and Rigollet, and Bonet et al.

Below we respond to each reviewer point by point.

---

### Decision · Action_Editor_eMq7 · 2026-05-25

**Recommendation:** Accept as is

**Audience:**

Yes

**Audience Explanation:**

SW is important in many areas of machine learning; therefore, the paper should be of interest to the TMLR audience.

**Claims And Evidence:**

Yes

**Claims Explanation:**

The paper revisits the Sliced-Wasserstein distance in high-dimensional settings and provides a new interpretation of its limitations when the data lie on a lower-dimensional subspace. In addition, under a certain framework, the paper derives scaling laws connecting ambient and effective dimensions, and argues that, with appropriate learning-rate tuning, vanilla SW can match the performance of more sophisticated sliced variants while remaining conceptually simpler.

The reviewers appreciated the clarity of the theoretical analysis and the practical relevance of the main observation. They found the paper’s perspective useful for understanding the behavior of SW in high dimensions, and valued the simple practical guidance provided by the results.

The main concerns raised by the reviewers concerned the degree of novelty, the reliance on the subspace assumption, and the experimental comparison. In particular, some reviewers felt that parts of the theoretical development were incremental or followed from known facts, and that the paper should better distinguish its genuinely new contributions from existing results.

However, most of these concerns have been adequately addressed by the authors. Therefore, while some limitations remain, they do not undermine the main contribution of the paper. Most reviewers and I therefore recommend acceptance.

---

> ### Author Response · Authors · 2026-05-28
>
> Thank you for the positive decision, and our sincere thanks to the editor and reviewers again for the insightful comments and shepherding of our manuscript.  We have uploaded the camera-ready version - which also allowed us to fix the author ordering.  Please let us know if anything else is needed.  Thank you again.
>
> (We note the paper still says "Decision pending for TMLR" in OpenReview but perhaps that is normal - thank you again!)